# A Theory-Driven Self-Labeling Refinement Method for Contrastive Representation Learning

**Pan Zhou**[*]    **Caiming Xiong**[*]    **Xiao-Tong Yuan**[†]    **Steven Hoi**[*]
[*] Salesforce Research
[†] Nanjing University of Information Science & Technology
panzhou3@gmail.com    {cxiong, shoi}@salesforce.com    xtyuan@nuist.edu.cn

## Abstract

For an image query, unsupervised contrastive learning labels crops of the same image as positives, and other image crops as negatives. Although intuitive, such a native label assignment strategy cannot reveal the underlying semantic similarity between a query and its positives and negatives, and impairs performance, since some negatives are semantically similar to the query or even share the same semantic class as the query. In this work, we first prove that for contrastive learning, inaccurate label assignment heavily impairs its generalization for semantic instance discrimination, while accurate labels benefit its generalization. Inspired by this theory, we propose a novel self-labeling refinement approach for contrastive learning. It improves the label quality via two complementary modules: (i) self-labeling refinery (SLR) to generate accurate labels and (ii) momentum mixup (MM) to enhance similarity between query and its positive. SLR uses a positive of a query to estimate semantic similarity between a query and its positive and negatives, and combines estimated similarity with vanilla label assignment in contrastive learning to iteratively generate more accurate and informative soft labels. We theoretically show that our SLR can exactly recover the true semantic labels of label-corrupted data, and supervises networks to achieve zero prediction error on classification tasks. MM randomly combines queries and positives to increase semantic similarity between the generated virtual queries and their positives so as to improves label accuracy. Experimental results on CIFAR10, ImageNet, VOC and COCO show the effectiveness of our method.

## 1 Introduction

Self-supervised learning (SSL) is an effective approach to learn features without manual annotations, with great success witnessed to many downstream tasks, e.g. image classification and object detection [1–7]. The methodology of SSL is to construct a pretext task that can obtain data labels via well designing the task itself, and then build a network to learn from these tasks. For instance, by constructing jigsaw puzzle [8], spatial arrangement identification [9], orientation [10], or chromatic channels [11] as a pretext task, SSL learns high-qualified features from the pretext task that can well transfer to downstream tasks. As it gets rid of the manual annotation requirement in supervised deep learning, SSL has been widely attracted increasing researching interests [1, 12].

As a leading approach in SSL, contrastive learning [1, 4, 13–17] constructs a novel instance discrimination pretext task to train a network so that the representations of different crops (augmentations) of the same instance are close, while representations of different instances are far from each other. Specifically, for an image crop query, it randomly augments the same image to obtain a positive, and view other image crops as negatives. Then it constructs a one-hot label over the positive and negatives to pull the query together with its positive and push the query away its negatives in the feature space.

35th Conference on Neural Information Processing Systems (NeurIPS 2021).

**Motivation.** But the one-hot labels in contrastive learning are indeed inaccurate and uninformative because of the following two reasons. *Firstly*, for a query, it could be semantically similar or even more similar to some negatives than its positives. Indeed, some negatives even belong to the same semantic class as the query [18–20]. It holds in practice, as (i) to achieve good performance, one often uses sufficient negatives that are much more than the semantic class number, e.g. tens of thousands of negatives for ImageNet [21] in MoCo [1], unavoidably leading to the issue on negatives; (ii) even for the same image, especially for an image containing different objects which occurs in ImageNet, random augmentations, e.g. crop, provide crops with (slightly) different semantic information, and thus some of the huge negatives could be more similar to query. *Secondly,* samples from different classes also have some similarity which is not characterized in the one-hot labels. For example, given a query cat, it is often more similar to a dog than a car though both dog and car are negatives. Learning from those similarity among samples is also important and can improve the performance. This point is also supported by knowledge distillation or self-training approaches [22, 23], where a teacher model or a currently-training model is used to predict semantic similarity of a sample on different classes for further supervising model training, achieving better performance. Therefore, the one-hot label cannot well reveal the semantic similarity between query and its positives and "negatives", and cannot guarantee the semantically similar samples to close each other, leading to performance degradation.

**Contributions.** In this work, we alleviate the above label issue, and derive some new results and alternatives for contrastive learning. Particularly, we theoretically show that inaccurate labels impair the performance of contrastive learning. Then we propose a self-labeling refinement method to obtain more accurate labels for contrastive learning. Our main contributions are highlighted below.

Our first contribution is proving that the generalization error of MoCo for instance discrimination linearly depends on the discrepancy between the estimated labels (e.g. one-hot labels) in MoCo and the true labels that really reflect semantical similarity between a query and its positives and negatives. Formally, given $n$ training queries $\mathcal{D} = \{\boldsymbol{x}_i\}_{i=1}^n$ with estimated labels $\{\boldsymbol{y}_i\}_{i=1}^n$ (e.g. one-hot labels in MoCo) and ground truth labels $\{\boldsymbol{y}_i^*\}_{i=1}^n$ on their corresponding positives and negatives, the generalization error of MoCo for instance discrimination is lower bounded by $\mathcal{O}\big(\mathbb{E}_{\mathcal{D}}[\|\boldsymbol{y} - \boldsymbol{y}^*\|_2]\big)$ where $\mathbb{E}_{\mathcal{D}}[\|\boldsymbol{y} - \boldsymbol{y}^*\|_2] = \frac{1}{n}\sum_{i=1}^n \|\boldsymbol{y}_i - \boldsymbol{y}_i^*\|_2$, and is upper bounded by $\mathcal{O}\big(\sqrt{\ln(|\mathcal{F}|)/n} + \mathbb{E}_{\mathcal{D}}[\|\boldsymbol{y} - \boldsymbol{y}^*\|_2]\big)$, where $|\mathcal{F}|$ is the covering number of the network hypotheses in MoCo. It means that the more accurate of the estimated labels $\{\boldsymbol{y}_i\}_{i=1}^n$, the better generalization of MoCo for instance discrimination.

Inspired by our theory, we propose a Self-lAbeliNg rEfinement (SANE) method which iteratively employs the network and data themselves to generate more accurate and informative soft labels for contrastive learning. SANE has two complementary modules: (i) *Self-Labeling Refinery (SLR)* to explicitly generate accurate labels, and (ii) *Momentum Mixup (MM)* to increase similarity between query and its positive and implicitly improve label accuracy. Given a query, SLR uses its one positive to estimate semantic similarity between the query and its keys (i.e. its positive and negatives) by computing their feature similarity, since a query and its positive come from the same image and should have close similarity on the same keys. Then SLR linearly combines the estimated similarity of a query with its vanilla one-hot label in contrastive learning to iteratively generate more accurate and informative soft labels. Our strategy is that at the early training stage, one-hot label has heavy combination weight to provide relatively accurate labels; along with more training, the estimated similarity becomes more accurate and informative, and its combination weight becomes larger as it explores useful underlying semantic information between the query and its keys that is missing in the one-hot labels. Besides, we prove that when the semantic labels in the instance discrimination task are corrupted, our SLR can exactly recover the true semantic labels of training data, and networks trained with our SLR can exactly predict the true semantic labels of test samples.

Moreover, we introduce MM for contrastive learning to further reduce the possible label noise and also increase augmentation diversity. Specifically, we randomly combines queries $\{\boldsymbol{x}_i\}_{i=1}^n$ and their positives $\{\widetilde{\boldsymbol{x}}_i\}_{i=1}^n$ as $\boldsymbol{x}_i' = \theta\boldsymbol{x}_i + (1-\theta)\widetilde{\boldsymbol{x}}_k$ and estimate their labels as $\boldsymbol{y}_i' = \theta\bar{\boldsymbol{y}}_i + (1-\theta)\bar{\boldsymbol{y}}_k$, where indexes $i$ and $k$ are randomly selected, $\bar{\boldsymbol{y}}_i$ is the label of both $\boldsymbol{x}_i$ and $\widetilde{\boldsymbol{x}}_i$ estimated by our label refinery, and $\theta \in (0, 1)$ is a random variable. In this way, the component $\widetilde{\boldsymbol{x}}_k$ in the virtual query $\boldsymbol{x}_i'$ directly increases the similarity between the query $\boldsymbol{x}_i'$ and the positive key $\widetilde{\boldsymbol{x}}_k$. So the label weight $(1 - \theta)$ of label $\boldsymbol{y}_i'$ on positive key $\widetilde{\boldsymbol{x}}_i$ to bring $\boldsymbol{x}_i'$ and $\widetilde{\boldsymbol{x}}_k$ together is relatively accurate, as $\boldsymbol{x}_i'$ really contains the semantic information of $\widetilde{\boldsymbol{x}}_k$. Meanwhile, the possible noise at the remaining positions of label $\boldsymbol{y}_i'$ is scaled by $\theta$ and becomes smaller. In this way, MM also improves the label quality.

**Other Related Work.** To estimate similarity between a query and its negatives, Wei *et al.* [20] approximated the similarity by computing cosine similarity between a positive and its negatives, and directly replaced the one-hot label for instance discrimination. Wang *et al.* [12] used similar similarity estimated on weak augmentations to supervise the learning of strong augmentations. In contrast, we respectively estimate the similarities of the query on all contrastive keys ( its positive and negatives) and on only negatives, and linearly combines two estimated similarities with vanilla one-hot label to obtain more accurate and informative label with provable performance guarantee. Learning from noisy label, e.g. [24–26] also uses soft labels generalized by a network to denoise crop labels and supervise representation learning, and often focus on (semi-)supervised learning that differs from our self-supervised learning.

Two relevant works [27, 28] performed vanilla mixup on all query instances to increase data diversity. Differently, our momentum mixup mainly aims to reduce label noise, as it randomly combines one query with one positive (instead of one query) of other instances to increase the similarity between the query and its its positive. Verma *et al.* [29] showed that mixup is a better domain-agnostic noise than Gaussian noise for positive pair construction. But they did not perform mixup on labels, which is contrast to [27, 28] and ours. See more discussion in Sec. 3.2 and empirical comparison in Sec. 4.3.

## 2 Inspiration: A Generalization Analysis of MoCo

In this section, we first briefly review the MoCo [1] method popularly studied for contrastive learning, and then analyze the impact of inaccurate label assignment on its generalization ability.

**Review of MoCo.** The MoCo method contains an online network $f_{\boldsymbol{w}}$ and a target network $g_{\boldsymbol{\xi}}$ receptively parameterized by $\boldsymbol{w}$ and $\boldsymbol{\xi}$. Both $f_{\boldsymbol{w}}$ and $g_{\boldsymbol{\xi}}$ consists of a feature encoder and a projection head (*e.g.* 3-layered MLP). Given a minibatch $\{\boldsymbol{c}_i\}_{i=1}^s$ at each iteration, it first randomly augments each vanilla image $\boldsymbol{c}_i$ into two views $(\boldsymbol{x}_i, \widetilde{\boldsymbol{x}}_i)$ and optimizes the following contrastive loss:

$$\mathcal{L}_{\mathrm{n}}(\boldsymbol{w}) = -\frac{1}{s} \sum\nolimits_{i=1}^s \log \Big( \frac{\sigma(\boldsymbol{x}_i, \widetilde{\boldsymbol{x}}_i)}{\sigma(\boldsymbol{x}_i, \widetilde{\boldsymbol{x}}_i) + \sum_{l=1}^b \sigma(\boldsymbol{x}_i, \boldsymbol{b}_l)} \Big), \tag{1}$$

where $\sigma(\boldsymbol{x}_i, \widetilde{\boldsymbol{x}}_i) = \exp\big(-\frac{\langle f(\boldsymbol{x}_i), g(\widetilde{\boldsymbol{x}}_i)\rangle}{\tau \|f(\boldsymbol{x}_i)\|_2 \cdot \|g(\widetilde{\boldsymbol{x}}_i)\|_2}\big)$ with a temperature $\tau$. The dictionary $\boldsymbol{B} = \{\boldsymbol{b}_i\}_{i=1}^b$ denotes the negative keys of current minibatch queries $\{\boldsymbol{x}_i\}_{i=1}^s$, and is often of huge size to achieve satisfactory performance, e.g. 65,536 in MoCo. In practice, $\boldsymbol{B}$ in MoCo is updated by the minibatch features $\{g(\widetilde{\boldsymbol{x}}_i)\}_{i=1}^s$ in a first-in and first-out order. By fixing $g_{\boldsymbol{\xi}}$ and updating $f_{\boldsymbol{w}}$ in Eqn. (1), MoCo pushes the query $\boldsymbol{x}_i$ away from its negative keys in dictionary $\boldsymbol{B}$ while pulling together its positive key $\widetilde{\boldsymbol{x}}_i$. For $g_{\boldsymbol{\xi}}$, it is updated via exponential moving average, i.e. $\boldsymbol{\xi} = (1-\iota)\boldsymbol{\xi} + \iota\boldsymbol{w}$ with a constant $\iota \in (0, 1)$.

From Eqn. (1), one can observe that MoCo views each image as an individual class and uses one-hot label $\boldsymbol{y} \in \mathbb{R}^{b+1}$ (its nonzero position is at the position of its positive key) to train $f_{\boldsymbol{w}}$. However, as mentioned in Sec. 1, the one-hot labels cannot reveal the semantic similarity between a query $\boldsymbol{x}_i$ and its positive and negatives and thus impair representation learning. In the following, we theoretically analyze the effect of inaccurate labels to the generalization of MoCo for instance discrimination.

**Generalization Analysis.** We focus on analyzing MoCo in the final training stage where the sample (key) distribution in the dictionary $\boldsymbol{B}$ is almost fixed. This simplified setup is reasonable because (i) in the final training stage, the target network $g_{\boldsymbol{\xi}}$ almost does not change due to the very small momentum updating parameter $\iota$ in practice and the oncoming convergence of the online network $f_{\boldsymbol{w}}$; (ii) dictionary is sufficient large to cover different patterns in the dataset. This fixed sample distribution simplifies the analysis, and also provides valuable insights.

Let $\mathcal{D} = \{(\boldsymbol{x}_i, \widetilde{\boldsymbol{x}}_i)\}_{i=1}^n$ denote the training positive pairs in MoCo sampled from an unknown distribution $\mathcal{S}$. Moreover, the query $\boldsymbol{x}_i$ has ground truth soft label $\boldsymbol{y}_i^* \in \mathbb{R}^{b+1}$ over the key set $\boldsymbol{B}_i = \{\widetilde{\boldsymbol{x}}_i \cup \boldsymbol{B}\}$, where the $t$-th entry $\boldsymbol{y}_{it}^*$ measures the semantic similarity between $\boldsymbol{x}_i$ and the $t$-th key $\boldsymbol{b}_t^l$ in $\boldsymbol{B}_i$. In practice, given query $\boldsymbol{x}_i$ and dictionary $\boldsymbol{B}_i$, MoCo estimates an one-hot label of $\boldsymbol{x}_i$ as $\boldsymbol{y}_i \in \mathbb{R}^{b+1}$ whose first entry is one and remaining entries are zero. So $\boldsymbol{y}_i$ ignores the semantic similarity between $\boldsymbol{x}_i$ and keys in $\boldsymbol{B}_i$, and differs from $\boldsymbol{y}_i^*$. Then MoCo minimizes an empirical risk:

$$\widetilde{\mathcal{Q}}(f_{\boldsymbol{w}}) = \frac{1}{n} \sum\nolimits_{i=1}^n \ell(h(f_{\boldsymbol{w}}(\boldsymbol{x}_i), \boldsymbol{B}_i), \boldsymbol{y}_i), \tag{2}$$

where $h(f_{\boldsymbol{w}}(\boldsymbol{x}_i), \boldsymbol{B}_i) = [\sigma(\boldsymbol{x}_i, \widetilde{\boldsymbol{x}}_i), \sigma(\boldsymbol{x}_i, \boldsymbol{b}_1), \cdots, \sigma(\boldsymbol{x}_i, \boldsymbol{b}_b)]$ denotes the predicted class probability, and $\ell(\cdot, \cdot)$ is cross-entropy loss. Ideally, one should sample sufficient pairs $(\boldsymbol{x}_i, \widetilde{\boldsymbol{x}}_i)$ from the distribution $\boldsymbol{\mathcal{S}}$ and use the ground truth label $\boldsymbol{y}_i^*$ of $\boldsymbol{x}_i$ to minimize the population risk:

$$\boldsymbol{\mathcal{Q}}(f_{\boldsymbol{w}}) = \mathbb{E}_{(\boldsymbol{x}_i, \widetilde{\boldsymbol{x}}_i) \sim \boldsymbol{\mathcal{S}}} \left[ \ell(h(f_{\boldsymbol{w}}(\boldsymbol{x}_i), \boldsymbol{B}_i), \boldsymbol{y}_i^*) \right]. \tag{3}$$

Here we assume the ground truth label $\boldsymbol{y}_i^*$ is soft which is indeed more reasonable and stricter than the one-hot label setting especially for contrastive learning [22]. It is because soft label requires the networks to capture the semantic similarity between query and the instances in $\boldsymbol{B}_i$ and bring semantically similar instances together, greatly helping downstream tasks (e.g. classification) where global semantic information is needed, while one-hot label only needs networks to distinguish each instance from others and does not consider the global semantic structures in the data. Actually, the semantic similarity among samples here is also known as "dark knowledge" in knowledge distillation [22] or "Bayes class-probability", and is often used to replace the one-hot label for training network with remarkable performance improvement in many tasks, e.g. classification [30]. As both the data distribution $\boldsymbol{\mathcal{S}}$ and the ground truth labels are unknown, MoCo optimizes the empirical risk $\widetilde{\boldsymbol{\mathcal{Q}}}(f_{\boldsymbol{w}})$ in (2) instead of the population risk $\boldsymbol{\mathcal{Q}}(f_{\boldsymbol{w}})$ in (3). In this way, the optimal network $f_{\boldsymbol{w}}$ by minimizing $\widetilde{\boldsymbol{\mathcal{Q}}}(f_{\boldsymbol{w}})$ differs from that via optimizing $\boldsymbol{\mathcal{Q}}(f_{\boldsymbol{w}})$. It is natural to ask whether $f_{\boldsymbol{w}}$ by minimizing $\widetilde{\boldsymbol{\mathcal{Q}}}(f_{\boldsymbol{w}})$ can well perform instance discrimination task in contrastive learning, i.e. whether $f_{\boldsymbol{w}}$ can capture the semantic similarity $(\boldsymbol{y}_i^*)$ between any test sample $(\boldsymbol{x}_i, \widetilde{\boldsymbol{x}}_i) \sim \boldsymbol{\mathcal{S}}$ and the keys (samples) in $\boldsymbol{B}_i$. To solve this issue, Theorem 1 analyzes the generalization error of $f_{\boldsymbol{w}}$ for instance discrimination. We are interested in the generalization error defined with the true soft labels to measure the semantic similarity learning performance of $f_{\boldsymbol{w}}$ via optimizing $\boldsymbol{\mathcal{Q}}(f_{\boldsymbol{w}})$ on the instance discrimination task which can often better reflect the performance on the downstream tasks.

**Theorem 1.** *Suppose $\ell(h(f_{\boldsymbol{w}}(\boldsymbol{x}), \boldsymbol{B}_{\boldsymbol{x}}), \boldsymbol{y}) \in [a_1, a_2]$, $\ell(\cdot, \boldsymbol{y})$ is $L_y$-Lipschitz w.r.t. $\boldsymbol{y}$. Let $\mathcal{F}$ be a finite class of hypotheses $\ell(h(f_{\boldsymbol{w}}(\boldsymbol{x}), \boldsymbol{B}_{\boldsymbol{x}}), \boldsymbol{y}) : \boldsymbol{\mathcal{X}} \times \boldsymbol{\mathcal{Y}} \to \mathbb{R}$ and $|\mathcal{F}|$ be its covering number under $\|\cdot\|_{\infty}$ metric.*
*(1) Let $\mathbb{E}_{\boldsymbol{\mathcal{D}} \sim \boldsymbol{\mathcal{S}}}[\|\boldsymbol{y} - \boldsymbol{y}^*\|_2] = \mathbb{E}_{\boldsymbol{\mathcal{D}} \sim \boldsymbol{\mathcal{S}}}\left[\frac{1}{n}\sum_{i=1}^n \|\boldsymbol{y}_i - \boldsymbol{y}_i^*\|_2\right]$. For any $\nu \in (0, 1)$, it holds*

$$\left| \boldsymbol{\mathcal{Q}}(f_{\boldsymbol{w}}) - \widetilde{\boldsymbol{\mathcal{Q}}}(f_{\boldsymbol{w}}) \right| \leq L_y \mathbb{E}_{\boldsymbol{\mathcal{D}} \sim \boldsymbol{\mathcal{S}}} \left[ \|\boldsymbol{y} - \boldsymbol{y}^*\|_2 \right]$$
$$+ \sqrt{\frac{2(a_2 - a_1)^2 V_{\boldsymbol{\mathcal{D}}} \ln(2|\mathcal{F}|/\nu)}{n}} + \frac{7(a_2 - a_1)^2 \ln(2|\mathcal{F}|/\nu)}{3(n-1)},$$

*with probability at least $1 - \nu$, where $V_{\boldsymbol{\mathcal{D}}}$ is the variance of $\ell(h(f(\boldsymbol{x}), \boldsymbol{B}_{\boldsymbol{x}}), \boldsymbol{y}^*)$ on the data $\boldsymbol{\mathcal{D}}$.*
*(2) There exists a contrastive classification problem, a class of hypotheses $\ell(h(f_{\boldsymbol{w}}(\boldsymbol{x}), \boldsymbol{B}_{\boldsymbol{x}}), \boldsymbol{y}) : \boldsymbol{\mathcal{X}} \times \boldsymbol{\mathcal{Y}} \to \mathbb{R}$ and a constant $c_0$ such that the generalization error of $f_{\boldsymbol{w}}$ is lower bounded*

$$\left| \boldsymbol{\mathcal{Q}}(f_{\boldsymbol{w}}) - \widetilde{\boldsymbol{\mathcal{Q}}}(f_{\boldsymbol{w}}) \right| \geq c_0 \cdot \mathbb{E}_{\boldsymbol{\mathcal{D}} \sim \boldsymbol{\mathcal{S}}} \left[ \|\boldsymbol{y} - \boldsymbol{y}^*\|_2 \right].$$

See its proof in Appendix B. Theorem 1 shows that for the task of learning semantic similarity between a query and its positive and negatives which is important for downstream tasks (e.g., classification), the generalization error of $f_{\boldsymbol{w}}$ trained with the one-hot labels $\boldsymbol{y}$ is upper bounded by $\mathcal{O}\big(\mathbb{E}_{\boldsymbol{\mathcal{D}} \sim \boldsymbol{\mathcal{S}}}[\|\boldsymbol{y} - \boldsymbol{y}^*\|_2] + \sqrt{V_{\boldsymbol{\mathcal{D}}} \ln(|\mathcal{F}|)/n}\big)$. It means that large training sample number $n$ gives small generalization error, as intuitively, model sees sufficient samples and can generalize better. The loss variance $V_{\boldsymbol{\mathcal{D}}}$ on the dataset $\boldsymbol{\mathcal{D}}$ measures data diversity: the larger data diversity $V_{\boldsymbol{\mathcal{D}}}$, the more challenging to learn a model with good generalization. Here we are particularly interested in the factor $\mathbb{E}_{\boldsymbol{\mathcal{D}} \sim \boldsymbol{\mathcal{S}}}[\|\boldsymbol{y} - \boldsymbol{y}^*\|_2]$ which reveals an important property: the higher accuracy of the training label $\boldsymbol{y}$ to the ground truth label $\boldsymbol{y}^*$, the smaller generalization error. Moreover, Theorem 1 proves that there exists a contrastive classification problem such that the lower bound of generalization error depends on $\mathbb{E}_{\boldsymbol{\mathcal{D}} \sim \boldsymbol{\mathcal{S}}}[\|\boldsymbol{y} - \boldsymbol{y}^*\|_2]$. So the upper bound of generalization error is tight in terms of $\mathbb{E}_{\boldsymbol{\mathcal{D}} \sim \boldsymbol{\mathcal{S}}}[\|\boldsymbol{y} - \boldsymbol{y}^*\|_2]$. Thus, to better capture the underlying semantic similarity between query $\boldsymbol{x}_i$ and samples in dictionary $\boldsymbol{B}_i$ to bring semantically similar samples together and better solve downstream tasks, one should provide accurate label $\boldsymbol{y}_i$ to the soft true label $\boldsymbol{y}_i^*$. In the following, we introduce our solution to estimate more accurate and informative soft labels for contrastive learning.

## 3  Self-Labeling Refinement for Contrastive Learning

Our Self-lAbeliNg rEfinement (SANE) approach for contrastive learning contains (i) *Self-Labeling Refinery* (SLR for short) and (ii) *Momentum Mixup* (MM) which complementarily refine noisy labels

respectively from label estimation and positive pair construction. SLR uses current training model and data to estimate more accurate and informative soft labels, while MM increases similarity between virtual query and its positive, and thus improves label accuracy.

We begin by slightly modifying the instance discrimination task in MoCo. Specifically, for the query $\boldsymbol{x}_i$ in the current minibatch $\{(\boldsymbol{x}_i, \widetilde{\boldsymbol{x}}_i)\}_{i=1}^s$, we maximize its similarity to its positive sample $\widetilde{\boldsymbol{x}}_i$ in the key set $\bar{\boldsymbol{B}} = \{\widetilde{\boldsymbol{x}}_i\}_{i=1}^s \cup \{\boldsymbol{b}_i\}_{i=1}^b$ and minimize its similarity to the remaining samples in $\bar{\boldsymbol{B}}$:

$$\mathcal{L}_c\big(\boldsymbol{w}, \{(\boldsymbol{x}_i, \boldsymbol{y}_i)\}\big) = -\frac{1}{s}\sum_{i=1}^s \sum_{k=1}^{s+b} \boldsymbol{y}_{ik} \log\left(\frac{\sigma(\boldsymbol{x}_i, \bar{\boldsymbol{b}}_k)}{\sum_{l=1}^{s+b} \sigma(\boldsymbol{x}_i, \bar{\boldsymbol{b}}_l)}\right), \tag{4}$$

where $\bar{\boldsymbol{b}}_k$ is the $k$-th sample in $\bar{\boldsymbol{B}}$, and $\boldsymbol{y}_i$ is the one-hot label of query $\boldsymbol{x}_i$ whose $i$-th entry $\boldsymbol{y}_{ii}$ is one. In this way, the labels of current queries $\{\boldsymbol{x}_i\}_{i=1}^s$ are defined on a shared set $\bar{\boldsymbol{B}}$, and can be linearly combined which is key for SLR & MM. Next, we aim to improve the quality of label $\boldsymbol{y}_i$ in (4) below.

### 3.1 Self-Labeling Refinery

**Methodology.** As analyzed in Sec. 1 and 2, the one-hot labels in Eqn. (4) could not well reveal the semantic similarity between $\boldsymbol{x}_i$ and the instance keys in the set $\bar{\boldsymbol{B}}$, and thus impairs good representation learning. To alleviate this issue, we introduce Self-Labeling Refinery (SLR) which employs network and data themselves to generate more accurate and informative labels, and improves the performance of contrastive learning. Specifically, to refine the one-hot label $\boldsymbol{y}_i$ of query $\boldsymbol{x}_i$, SLR uses its positive instance $\widetilde{\boldsymbol{x}}_i$ to estimate the underlying semantic similarity between $\boldsymbol{x}_i$ and instances in $\bar{\boldsymbol{B}} = \{\widetilde{\boldsymbol{x}}_i\}_{i=1}^s \cup \{\boldsymbol{b}_i\}_{i=1}^b$, since $\boldsymbol{x}_i$ and $\widetilde{\boldsymbol{x}}_i$ come from the same image and should have close semantic similarity with instances in $\bar{\boldsymbol{B}}$. Let $\bar{\boldsymbol{b}}_k$ be the $k$-th sample in $\bar{\boldsymbol{B}}$. Then at the $t$-th iteration, SLR first estimates the instance-class probability $\boldsymbol{p}_i^t \in \mathbb{R}^{s+b}$ of $\boldsymbol{x}_i$ on the set $\bar{\boldsymbol{B}}$ whose $k$-th entry is defined as

$$\boldsymbol{p}_{ik}^t = \frac{\sigma^{1/\tau'}(\widetilde{\boldsymbol{x}}_i, \bar{\boldsymbol{b}}_k)}{\sum_{l=1}^{s+b} \sigma^{1/\tau'}(\widetilde{\boldsymbol{x}}_i, \bar{\boldsymbol{b}}_l)}, \quad (\tau' \in (0,1]).$$

The constant $\tau'$ sharpens $\boldsymbol{p}_i^t$ and removes some possible small noise, since smooth labels cannot well distillate their knowledge to a network [31]. Then SLR uses $\boldsymbol{p}_i^t$ to approximate the semantic similarity between $\boldsymbol{x}_i$ and the instances in $\bar{\boldsymbol{B}}$ and employs it as the soft label of $\boldsymbol{x}_i$ for contrastive learning.

However, since $\widetilde{\boldsymbol{x}}_i$ is highly similar to itself, $\boldsymbol{p}_{ii}^t$ could be much larger than others and conceals the similarity of other semantically similar instances in $\bar{\boldsymbol{B}}$. To alleviate this artificial effect, SLR removes $\widetilde{\boldsymbol{x}}_i$ from the set $\bar{\boldsymbol{B}}$ and re-estimates the similarity between $\boldsymbol{x}_i$ and the remaining instances in $\bar{\boldsymbol{B}}$:

$$\boldsymbol{q}_{ik}^t = \frac{\sigma^{1/\tau'}(\widetilde{\boldsymbol{x}}_i, \bar{\boldsymbol{b}}_k)}{\sum_{l=1, l \neq i}^{s+b} \sigma^{1/\tau'}(\widetilde{\boldsymbol{x}}_i, \bar{\boldsymbol{b}}_l)}, \quad \boldsymbol{q}_{ii}^t = 0.$$

Finally, SLR linearly combines the one-hot label $\boldsymbol{y}_i$ and two label estimations, i.e. $\boldsymbol{p}_i$ and $\boldsymbol{q}_i$, to obtain more accurate, robust and informative label $\bar{\boldsymbol{y}}_i^t$ of $\boldsymbol{x}_i$ at the $t$-th iteration:

$$\bar{\boldsymbol{y}}_i^t = (1 - \alpha_t - \beta_t)\boldsymbol{y}_i + \alpha_t \boldsymbol{p}_i^t + \beta_t \boldsymbol{q}_i^t, \tag{5}$$

where $\alpha_t$ and $\beta_t$ are two constants. In our experiments, we set $\alpha_t = \mu \max_k \boldsymbol{p}_{ik}^t / z$ and $\beta_t = \mu \max_k \boldsymbol{q}_{ik}^t / z$, where $z = 1 + \mu \max_k \boldsymbol{p}_{ik}^t + \mu \max_k \boldsymbol{q}_{ik}^t$, the constants 1, $\max_k \boldsymbol{p}_{ik}^t$ and $\max_k \boldsymbol{q}_{ik}^t$ respectively denote the largest confidences of labels $\boldsymbol{y}_i$, $\boldsymbol{p}_i^t$ and $\boldsymbol{q}_i^t$ on a certain class. Here hyperparameter $\mu$ controls the prior confidence of $\boldsymbol{p}^t$ and $\boldsymbol{q}^t$. So SLR only has two parameters $\tau'$ and $\mu$ to tune.

**The Benefit Analysis of Label Refinery.** Now we analyze the performance of our SLR on label-corrupted data. We first describe the dataset. Let $\{\boldsymbol{c}_i\}_{i=1}^K \subset \mathbb{R}^d$ be $K$ vanilla samples belonging to $\bar{K} \leq K$ semantic classes, and $\{(\boldsymbol{x}_i, \boldsymbol{y}_i)\}_{i=1}^n \in \mathbb{R}^d \times \mathbb{R}$ be the random crops of $\{\boldsymbol{c}_i\}_{i=1}^K$. Since in practice, one often cares more the semantic class prediction performance of a model which often directly reflects the performance on the downstream tasks, we assume that the labels $\{\boldsymbol{y}_i\}_{i=1}^n$ denote corrupted semantic-class labels. Accordingly, we will analyze whether SLR can refine the corrupted labels $\{\boldsymbol{y}_i\}_{i=1}^n$ and whether it helps a model learn the essential semantic-class knowledge of $\{\boldsymbol{x}_i\}_{i=1}^n$. Finally, while allowing for multiple classes, we assume the labels are scalars and take values in $[-1, 1]$ interval for simplicity. We formally define our label-corrupted dataset below.

**Definition 1** (($\rho, \varepsilon, \delta$)-corrupted dataset). *Let $\{(\boldsymbol{x}_i, \boldsymbol{y}_i^*)\}_{i=1}^n$ denote the pairs of crops (augmentations) and ground-truth semantic label, where crop $\boldsymbol{x}_i$ generated from the $t$-th sample $\boldsymbol{c}_t$ obeys $\|\boldsymbol{x}_i - \boldsymbol{c}_t\|_2 \leq \varepsilon$ with a constant $\varepsilon$, and $\boldsymbol{y}_i^* \in \{\gamma_t\}_{t=1}^K$ of $\boldsymbol{x}_i$ is the label of $\boldsymbol{c}_t$. Moreover, samples and the crops are normalized, i.e. $\|\boldsymbol{c}_i\|_2 = \|\boldsymbol{x}_k\|_2 = 1 (\forall i, k)$. Each $\boldsymbol{c}_i$ has $n_i$ crops, where $c_l \frac{n}{K} \leq n_i \leq c_u \frac{n}{K}$ with two constants $c_l$ and $c_u$. Besides, different classes are separated with a label separation $\delta$:*

$$|\gamma_i - \gamma_k| \geq \delta, \quad \|\boldsymbol{c}_i - \boldsymbol{c}_k\|_2 \geq 2\varepsilon, \ (\forall i \neq k).$$

*A ($\rho, \varepsilon, \delta$)-corrupted dataset $\{(\boldsymbol{x}_i, \boldsymbol{y}_i)\}_{i=1}^n$ obeys the above conditions but with corrupted label $\{\boldsymbol{y}_i\}_{i=1}^n$. Spefically, for each sample $\boldsymbol{c}_i$, at most $\rho n_i$ augmentations are assigned to wrong labels in $\{\gamma_i\}_{i=1}^K$.*

This data model is rich enough to model realistic data, since different clusters can be assigned to the same label. This definition allows for a fraction $\rho$ of corruptions in each cluster, and can well characterize the realistic data.

Then we study a network of one hidden layer as an example to investigate the label refining performance of our SLR. The network parameterized by $\boldsymbol{W} \in \mathbb{R}^{k \times d}$ and $\boldsymbol{v} \in \mathbb{R}^k$ is defined as

$$\boldsymbol{x} \in \mathbb{R}^d \mapsto f(\boldsymbol{W}, \boldsymbol{x}) = \boldsymbol{v}^\top \phi(\boldsymbol{W}\boldsymbol{x}), \tag{6}$$

where $\phi$ is an activation function. Following [32–34] which analyze convergence of networks or robust learning of network, we fix $\boldsymbol{v}$ to be a unit vector where half the entries are $1/\sqrt{k}$ and other half are $-1/\sqrt{k}$ to simplify exposition. So we only optimize over $\boldsymbol{W}$ that contains most network parameters and will be shown to be sufficient for label refinery. Then given a ($\rho, \varepsilon, \delta$)-corrupted dataset $\{(\boldsymbol{x}_i, \boldsymbol{y}_i)\}_{i=1}^n$, at the $t$-iteration we train the network via minimizing the quadratic loss:

$$\mathcal{L}_t(\boldsymbol{W}) = \frac{1}{2} \sum_{i=1}^n (\bar{\boldsymbol{y}}_i^t - f(\boldsymbol{W}, \boldsymbol{x}_i))^2 = \frac{1}{2} \|\bar{\boldsymbol{y}}^t - f(\boldsymbol{W}, \boldsymbol{X})\|_2^2.$$

Here the label $\bar{\boldsymbol{y}}_i^t$ of sample $\boldsymbol{x}_i$ is estimated by Eqn. (5) in which $\boldsymbol{p}_i^t = f(\boldsymbol{W}_t, \widetilde{\boldsymbol{x}}_i)$ denotes predicted label by using the positive $\widetilde{\boldsymbol{x}}_i$ of $\boldsymbol{x}_i$, i.e. $\|\widetilde{\boldsymbol{x}}_i - \boldsymbol{c}_l\|_2 \leq \varepsilon$ if $\boldsymbol{x}_i$ is augmented from vanilla sample $\boldsymbol{c}_l$. We set $\beta_t = 0$ and $\tau' = 1$ for simplicity, as (i) performing nonlinear mapping on network output greatly increases analysis difficulty; (ii) our refinery (5) is still provably sufficient to refine labels when $\beta_t = 0$ and $\tau' = 1$. Then we update $\boldsymbol{W}$ via gradient descent algorithm with a learning rate $\eta$:

$$\boldsymbol{W}_{t+1} = \boldsymbol{W}_t - \eta \nabla \mathcal{L}_t(\boldsymbol{W}_t). \tag{7}$$

Following most works on network convergence analysis [32–40], we use gradient descent and quadratic loss, since (i) gradient descent is expectation version of stochastic one and often reveals similar convergence behaviors; (ii) one can expect similar results for other losses, e.g. cross entropy, but quadratic loss gives simpler gradient computation. For analysis, we impose mild assumptions on network (6) and our SLR, which are widely used in network analysis [41–46].

**Assumption 1.** *For network (6), assume $\phi$ and its first- and second-order derivatives obey $|\phi(0)|, |\phi'(z)|, |\phi''(z)| \leq \Gamma$ for $\forall z$ and some $\Gamma \geq 1$, the entries of initialization $\boldsymbol{W}_0$ obey i.i.d. $\mathcal{N}(0,1)$.*

**Assumption 2.** *Define network covariance matrix $\Sigma(\boldsymbol{C}) = (\boldsymbol{C}\boldsymbol{C}^\top) \odot \mathbb{E}_{\boldsymbol{u}}[\phi'(\boldsymbol{C}\boldsymbol{u})\phi'(\boldsymbol{C}\boldsymbol{u})^\top]$ where $\boldsymbol{C} = [\boldsymbol{c}_1 \dots \boldsymbol{c}_K]^\top$, $\boldsymbol{u} \sim \mathcal{N}(\boldsymbol{0}, \boldsymbol{I})$, $\odot$ is the elementwise product. Let $\lambda(\boldsymbol{C}) > 0$ be the minimum eigenvalue of $\Sigma(\boldsymbol{C})$. For label refinery, assume $3\sqrt{n} \sum_{t=0}^{t_0-1} |\alpha_t - \alpha_{t+1}| \leq \psi_1 \|f(\boldsymbol{W}_0, \boldsymbol{X}) - \boldsymbol{y}^*\|_2$ and $3\sqrt{n} \sum_{t=0}^{t_0-1} \left(1 - \frac{\eta \alpha^2}{4}\right)^{t_0-t} |\alpha_t - \alpha_{t+1}| \leq \psi_2 \|f(\boldsymbol{W}_0, \boldsymbol{X}) - \boldsymbol{y}^*\|_2^2$, where $t_0 = \frac{c_1 K}{\eta n \lambda(\boldsymbol{C})} \log \left(\frac{\Gamma \sqrt{n \log K}}{(1-\alpha_{max})\rho}\right)$ with three constants $\psi_1$, $\psi_2$ and $c_1$. Here $\alpha_{max}$ is defined as $\alpha_{max} = \max_{1 \leq t \leq t_0} \alpha_t$.*

Assumption 1 is mild, as most differential activation functions, e.g. softplus and sigmoid, satisfy it, and the Gaussian initialization is used in practice. We assume Gaussian variance to be one for notation simplicity, but our technique is applicable to any constant variance. Assumption 2 requires that the discrepancy between $\alpha_t$ and $\alpha_{t+1}$ until some iteration number $t_0$ are bounded, which holds by setting proper $\alpha_t$. Here we assume $\beta_t = 0$ and $\tau' = 1$ for simplicity, as (i) performing nonlinear mapping on network output greatly increases analysis difficulty; (ii) we will show that even though $\beta_t = 0$ and $\tau' = 1$, our refinery (5) is still provably sufficient to refine labels. For $\lambda(\boldsymbol{C})$, many works [34, 41–44] empirically and theoretically show $\lambda(\boldsymbol{C}) > 0$. Based on these assumptions, we state our results in Theorem 2 with constants $c_1 \sim c_6$.

**Theorem 2.** *Assume $\{(\boldsymbol{x}_i, \boldsymbol{y}_i)\}_{i=1}^n$ is a $(\rho, \varepsilon, \delta)$-corrupted dataset with noiseless labels $\{\boldsymbol{y}_i^*\}_{i=1}^n$. Let $\xi = \log\left(\frac{\Gamma\sqrt{n\log K}}{\rho}\right)$. Suppose $\varepsilon$ and the number $k$ of hidden nodes satisfy $\varepsilon \leq c_2 \min(\frac{\lambda(\boldsymbol{C})}{K\Gamma^2\xi^6}, \frac{\rho}{\alpha_{max}})$, $k \geq \frac{c_3 K^2 \Gamma^{10}\xi^6\|\boldsymbol{C}\|^4}{\alpha_{max}^2\lambda(\boldsymbol{C})^4}$. Let $\psi' = 1 + \frac{\psi_1}{2} + \sqrt{\psi_2}$. If step size $\eta \leq \frac{K}{2c_{up}n\Gamma^2\|\boldsymbol{C}\|^2}$, with probability $1\text{–}3/K^{100}\text{–}K\exp(-100d)$, after $t \geq t_0 = \frac{c_4 K}{\eta n\lambda(\boldsymbol{C})}\log\left(\frac{\Gamma\sqrt{n\log K}}{(1-\alpha_{max})\rho}\right)$ iterations, the gradient descent (7) satisfies:*

*(1) By defining $\zeta = 4\rho + c_5\varepsilon\psi' K\Gamma^3\xi\sqrt{\log K}/\lambda(\boldsymbol{C})$ and $\boldsymbol{y}^* = [\boldsymbol{y}_1^*, \cdots, \boldsymbol{y}_n^*]$, the discrepancy between the label $\bar{\boldsymbol{y}}^t$ estimated by our SLR (5) and the true label $\boldsymbol{y}^*$ of the augmentation data $\{\boldsymbol{x}_i\}_{i=1}^n$ is bounded:*

$$\frac{1}{\sqrt{n}}\|\bar{\boldsymbol{y}}^t - \boldsymbol{y}^*\|_2 \leq \frac{1-\alpha_t}{\sqrt{n}}\|\boldsymbol{y} - \boldsymbol{y}^*\|_2 + \alpha_t\zeta.$$

*where $\bar{\boldsymbol{y}}^t = [\bar{\boldsymbol{y}}_1^t, \cdots, \bar{\boldsymbol{y}}_n^t]$. Moreover, if $\rho \leq \frac{\delta}{32}$, $\varepsilon \leq c_6\delta\min\left(\frac{\lambda(\boldsymbol{C})^2}{\psi'\Gamma^5 K^2\xi^3}, \frac{1}{\Gamma\sqrt{d}}\right)$, $1 - \frac{3}{4}\delta \leq \alpha_t$, the estimated label $\bar{\boldsymbol{y}}^t$ predicts true label $\boldsymbol{y}_i^*$ of any crop $\boldsymbol{x}_i$:*

$$\gamma_{k^*} = \boldsymbol{y}_i^* \quad\text{with}\quad k^* = \mathrm{argmin}_{1 \leq k \leq \bar{K}}\,|\bar{\boldsymbol{y}}_i^t - \gamma_k|.$$

*(2) By using the refined label $\bar{\boldsymbol{y}}^t$ in (5) to train network and letting $f(\boldsymbol{W}_t, \boldsymbol{X}) = [f(\boldsymbol{W}_t, \boldsymbol{x}_1), \cdots, f(\boldsymbol{W}_t, \boldsymbol{x}_n)]$, the error of network prediction on $\{\boldsymbol{x}_i\}_{i=1}^n$ is upper bounded*

$$\frac{1}{\sqrt{n}}\|f(\boldsymbol{W}_t, \boldsymbol{X}) - \boldsymbol{y}^*\|_2 \leq \zeta.$$

*If assumptions on $\rho$ and $\varepsilon$ in (1) hold, for vanilla sample $\boldsymbol{c}_k$ ($\forall k = 1\cdots K$), network $f(\boldsymbol{W}_t, \cdot)$ predicts the true semantic label $\gamma_k$ of its any augmentation $\boldsymbol{x}$ that satisfies $\|\boldsymbol{x} - \boldsymbol{c}_k\|_2 \leq \varepsilon$:*

$$\gamma_{k^*} = \gamma_k \quad\text{with}\quad k^* = \mathrm{argmin}_{1 \leq i \leq \bar{K}}\,|f(\boldsymbol{W}_t, \boldsymbol{x}) - \gamma_i|.$$

See its *proof roadmap* and proof in Appendix C.2. The first result in Theorem 2 shows that after training iterations $t_0$, the discrepancy between the label $\bar{\boldsymbol{y}}^t$ estimated by our label refinery (5), i.e. SLR, and ground truth label $\boldsymbol{y}^*$ of cropped training data $\{\boldsymbol{x}_i\}_{i=1}^n$ is upper bounded by $\mathcal{O}\left(\|\boldsymbol{y} - \boldsymbol{y}^*\|_2 + \zeta\right)$. Both factors $\|\boldsymbol{y} - \boldsymbol{y}^*\|_2$ and $\rho$ in the factor $\zeta$ reflect the label error of the provided corrupted label $\boldsymbol{y}$. Another important factor in $\zeta$ is the smallest eigenvalue $\lambda(\boldsymbol{C})$ of network covariance matrix $\boldsymbol{\Sigma}(\boldsymbol{C})$ in Assumption 2. Typically, the performance of a network heavily relies on the data diversity even without label corruption. For instance, if two samples are nearly the same but have different labels, the learning of a network is difficult. $\lambda(\boldsymbol{C})$ can quantify this data diversity, as one can think of $\lambda(\boldsymbol{C})$ as a condition number associated with the network which measures the diversity of the vanilla samples $\{\boldsymbol{c}_i\}_{i=1}^n$. Intuitively, if there are two similar vanilla samples, $\boldsymbol{\Sigma}(\boldsymbol{C})$ is trivially rank deficient and has small minimum eigenvalue, meaning more challenges to distinguish the augmentations $\boldsymbol{x}$ generated from $\boldsymbol{c}_i$. Moreover, when the label corruption ratio $\rho$ and the augmentation distance $\varepsilon$ are small, the label $\bar{\boldsymbol{y}}_i^t$ estimated by our SLR can predict the true semantic label $\boldsymbol{y}_i^*$ for any crop sample $\boldsymbol{x}_i$, and thus can supervises a network to learn the essential semantic-class knowledges from $\{\boldsymbol{x}_i\}_{i=1}^n$.

The second result in Theorem 2 shows that by using the refined label $\bar{\boldsymbol{y}}^t$ in our SLR (5) to train network $f(\boldsymbol{W}, \cdot)$, the error of network prediction on augmentations $\{\boldsymbol{x}_i\}_{i=1}^n$ can be upper bounded by $\zeta$. Similarly, the factor $\rho$ and $\lambda(\boldsymbol{C})$ in $\zeta$ respectively reflect the initial label error and the data diversity, which both reflect the learning difficulty for a model on the augmentation data $\{(\boldsymbol{x}_i, \boldsymbol{y}_i)\}_{i=1}^n$. More importantly, our results also guarantee the test performance of the trained network $f(\boldsymbol{W}_t, \cdot)$. Specifically, when the label corruption ratio $\rho$ and sample augmentation distance $\varepsilon$ are small, for any vanilla sample $\boldsymbol{c}_k$ ($\forall k = 1\cdots K$), the network $f(\boldsymbol{W}_t, \cdot)$ trained by our SLR can exactly predict the true semantic label $\gamma_k$ of its any augmentation $\boldsymbol{x}$ (i.e. $\|\boldsymbol{x} - \boldsymbol{c}_k\|_2 \leq \varepsilon$). These results accord with Theorem 1 that shows the more accurate of training labels, the better generalization of the trained network. These results show the effectiveness of the refined labels by our method.

## 3.2 Momentum Mixup

Now we propose momentum mixup (MM) to further reduce the possible label noise in realistic data and increase the data diversity as well. Similar to vanilla mixup [47], we construct virtual instance as

$$\boldsymbol{x}_i' = \theta\boldsymbol{x}_i + (1-\theta)\widetilde{\boldsymbol{x}}_k, \quad \boldsymbol{y}_i' = \theta\bar{\boldsymbol{y}}_i + (1-\theta)\bar{\boldsymbol{y}}_k, \quad \theta \sim \mathsf{Beta}(\kappa, \kappa) \in [0, 1], \tag{8}$$

where $\widetilde{\boldsymbol{x}}_k$ is randomly sampled from the key set $\{\widetilde{\boldsymbol{x}}_i\}_{i=1}^s$, $\bar{\boldsymbol{y}}_i$ denotes the refined label by Eqn. (5), Beta$(\kappa, \kappa)$ is a beta distribution. Here $\boldsymbol{x}_i$ and $\widetilde{\boldsymbol{x}}_i$ share the same label $\bar{\boldsymbol{y}}_i$ on the set $\bar{\boldsymbol{B}} = \{\widetilde{\boldsymbol{x}}_i\}_{i=1}^s \cup \{\boldsymbol{b}_i\}_{i=1}^b$, as they come from the same instance. We call the mixup (8) as "momentum mixup", since the sample $\widetilde{\boldsymbol{x}}_k$ is fed into the momentum-updated network $g_{\boldsymbol{\xi}}$, and plays a contrastive key for instance discrimination. So MM differs from the vanilla mixup used in [27, 28] where $\widetilde{\boldsymbol{x}}_k$ is replaced with $\boldsymbol{x}_k$ and both are fed into online network $f_{\boldsymbol{w}}$, and enjoys the following advantages. Firstly, MM can improve the accuracy of the label $\boldsymbol{y}'_i$ compared with vanilla mixup. For explanation, assume $\bar{\boldsymbol{y}}_i$ in (8) is one-hot label. Then $\boldsymbol{x}'_i$ has two positive keys $\widetilde{\boldsymbol{x}}_i$ and $\widetilde{\boldsymbol{x}}_k$ in $\bar{\boldsymbol{B}}$ decided by its label $\boldsymbol{y}'_i$. Accordingly, the component $\widetilde{\boldsymbol{x}}_k$ in $\boldsymbol{x}'_i = \theta\boldsymbol{x}_i + (1-\theta)\widetilde{\boldsymbol{x}}_k$ directly increases the similarity between the query $\boldsymbol{x}'_i$ and its positive key $\widetilde{\boldsymbol{x}}_k$ in $\bar{\boldsymbol{B}}$. So the label weight $(1-\theta)$ of label $\boldsymbol{y}'_i$ on the key $\widetilde{\boldsymbol{x}}_k$ to bring $\boldsymbol{x}'_i$ and $\widetilde{\boldsymbol{x}}_k$ together is relatively accurate, as $\boldsymbol{x}'_i$ really contains the semantic information of $\widetilde{\boldsymbol{x}}_k$. Meanwhile, the sum of label weights in $\boldsymbol{y}'_i$ on remaining instance in $\bar{\boldsymbol{B}}\backslash\widetilde{\boldsymbol{x}}_k$ is scaled by $\theta$, which also scales the possible label noise on instances in $\bar{\boldsymbol{B}}\backslash\widetilde{\boldsymbol{x}}_k$ smaller due to $\theta < 1$. By comparison, for vanilla mixup, the label weight $(1-\theta)$ of label $\boldsymbol{y}'_i$ on the key $\widetilde{\boldsymbol{x}}_i$ does not improve label accuracy. It is because the positive pair $\boldsymbol{x}_k$ and $\widetilde{\boldsymbol{x}}_k$ are obtained via random augmentation, e.g. crop, and may not be semantically similar, meaning that the component $\boldsymbol{x}_k$ in $\boldsymbol{x}'_i$ could not increase similarity with $\widetilde{\boldsymbol{x}}_k$. So its label weight $(1-\theta)$ to push $\boldsymbol{x}'_i$ close to the key $\widetilde{\boldsymbol{x}}_k$ is not as accurate as the one in MM. Another advantage of MM is that it allows us to use strong augmentation. As observed in [12], directly using strong augmentation in contrastive learning, e.g. MoCo, leads to performance degradation, since the instance obtained by strong augmentation often heavily differs from the one with weak augmentation. As aforementioned, the component $\widetilde{\boldsymbol{x}}_k$ in $\boldsymbol{x}'_i = \theta\boldsymbol{x}_i + (1-\theta)\widetilde{\boldsymbol{x}}_k$ increases the similarity between the query $\boldsymbol{x}'_i$ and the key $\widetilde{\boldsymbol{x}}_k$ in $\bar{\boldsymbol{B}}$, even though $(\boldsymbol{x}_i, \widetilde{\boldsymbol{x}}_i)$ is obtained via strong augmentation. So MM could reduce the matching difficulty between positive instances.

With all the components in place, we are ready to define our proposed *SANE model* as follows:

$$\mathcal{L}(\boldsymbol{w}) = (1-\lambda)\mathcal{L}_c\big(\boldsymbol{w}, \{(\boldsymbol{x}_i, \boldsymbol{y}_i)\}\big) + \lambda\mathcal{L}_c\big(\boldsymbol{w}, \{(\boldsymbol{x}'_i, \boldsymbol{y}'_i)\}\big), \tag{9}$$

where $\mathcal{L}_c\big(\boldsymbol{w}, \{(\boldsymbol{x}_i, \boldsymbol{y}_i)\}\big)$ defined in Eqn. (4) denotes the vanilla contrastive loss with one-hot label $\boldsymbol{y}_i$, $\mathcal{L}_c\big(\boldsymbol{w}, \{(\boldsymbol{x}'_i, \boldsymbol{y}'_i)\}\big)$ denotes the momentum mixup loss with label $\boldsymbol{y}'_i$ estimated by our self-labeling refinery (5), and $\lambda$ is a constant. Experimental results in Sec. 4 show the effectiveness of both loss terms. See algorithm details in Algorithm 1 of Appendix A.

**Limitation Discussion.** SANE follows MoCo-alike framework and hopes to obtain a more accurate soft label of a query over its positive and negatives for instance discrimination. So one limitation of SANE is that it does not apply to BYOL-alike methods [6] that only pulls positive pair together and does not require any labels. However, momentum mixup in SANE which increases the similarity of positive pair may also benefit BYOL, which is left as our future work to thoroughly test.

**Societal Impact Discussion.** As an unsupervised learning method, SANE could benefit many applications of societal interest where only low-resource labeled data are available, e.g. medical data [48], as SANE can use a mass of unlabeled data for pretraining and few labeled data for fine-tuning. But same to standard contrastive learning, SANE may suffer from learning bias caused by the potential data bias, and may provide bias or worse performance on smaller classes or groups.

## 4 Experiments

Our Pytorch code is available at https://openreview.net/forum?id=P84bifNCpFQ&referrer=%5BAuthor%20Console%5D.

### 4.1 Evaluation Results on CIFAR10 and ImageNet

**Settings.** We use ResNet50 [49] with a 3-layered MLP head for CIFAR10 [50] and ImageNet [21]. We first pretrain SANE, and then train a linear classifier on top of 2048-dimensional frozen features in ResNet50. With dictionary size $4,096$, we pretrain $2,000$ epochs on CIFAR10 instead of $4,000$ epochs of MoCo, BYOL, and i-Mix in [28]. Dictionary size on ImageNet is $65,536$. For linear classifier, we train 200/100 epochs on CIFAR10/ImageNet. See all optimizer settings in

Table 1: Classification accuracy (%).

| CIFAR10 dataset | KNN | linear evaluation |
|---|---|---|
| MoCo v2 [2] | 92.5 | 93.9 |
| SimCLR [3, 4] | — | 94.0 |
| BYOL [6] | 92.4 | 93.9 |
| DACL [29] | — | 94.4 |
| CLSA (strong) [12] | 93.4 | 94.9 |
| i-Mix (+MoCo) [28] | — | 95.9 |
| SANE | 95.2 | 96.1 |
| SANE (strong) | 95.5 | 96.5 |
| Supervised [28] | — | 95.5 |

Table 2: Top-1 accuracy (%) under linear evaluation on ImageNet.

| augmentation | method (200 epochs) | Top 1 | method ($\geq$800 epochs) | Top 1 |
|---|---|---|---|---|
| weak | MoCo [1] | 60.8 | PIRL-800epochs [51] | 63.6 |
| | SimCLR [4] | 61.9 | CMC [52] | 66.2 |
| | CPC v2 [53] | 63.8 | SimCLR-800epochs [4] | 70.0 |
| | PCL [7] | 65.9 | MoCo v2-800epochs [2] | 71.1 |
| | MoCo v2 [2] | 67.5 | BYOL-1000epochs [6] | 74.3 |
| | CO2 [20] | 68.0 | SimSiam-800epochs [54] | 71.3 |
| | MixCo [27] | 68.4 | i-Mix-800epochs [28] | 71.3 |
| | SWAV-Multi [5] | 72.7 | SWAV-Multi-800epochs [5] | 75.3 |
| | SANE-Single | 70.6 | SANE-Single-800epochs | 73.0 |
| | SANE-Multi | 73.5 | SANE-Multi-800epochs | 75.7 |
| strong | CLSA-Single [12] | 69.4 | CLSA-Single-800epochs [12] | 72.2 |
| | CLSA-Multi [12] | 73.3 | CLSA-Multi-800epochs [12] | 76.2 |
| | SANE-Single | 70.1 | SANE-Single-800epochs | 73.5 |
| | SANE-Multi | 73.7 | SANE-Multi-800epochs | 76.4 |
| strong + JigSaw | InfoMin Aug [55] | 70.1 | InfoMin Aug-800epochs [55] | 73.0 |
| others | InstDisc [56] | 54.0 | BigBiGAN [57] | 56.6 |
| | LocalAgg [58] | 58.8 | SeLa-400epochs [59] | 61.5 |
| | Supervised [4] | 76.5 | Supervised [4] | 76.5 |

Appendix A. We use standard data augmentations in [1] for pretraining and test unless otherwise stated. E.g., for test, we perform normalization on CIFAR10, and use center crop and normalization on ImageNet. For SANE, we set $\tau = 0.2, \tau' = 0.8, \kappa = 2$ in $\mathsf{Beta}(\kappa, \kappa)$ on CIFAR10, and $\tau = 0.2, \tau' = 1, \kappa = 0.1$ on ImageNet. For confidence $\mu$, we increase it as $\mu_t = m_2 - (m_2 - m_1)(\cos(\pi t/T) + 1)/2$ with current iteration $t$ and total training iteration $T$. We set $m_1 = 0, m_2 = 1$ on CIFAR10, and $m_1 = 0.5, m_2 = 10$ on ImageNet. For KNN on CIFAR10, its neighborhood number is 50 and its temperature is 0.05.

For CIFAR10, to fairly compare with [28], we crop each image into two views to construct the loss (9). For ImageNet, we follow CLSA [12] and train SANE in two settings. SANE-Single uses a single crop in momentum mixup loss $\mathcal{L}_c(\boldsymbol{w}, \{(\boldsymbol{x}_i', \boldsymbol{y}_i')\})$ in (9) that crops each image to a smaller size of $96 \times 96$, without much extra computational cost to process these small images. SANE-Multi crop each image into five sizes $224 \times 224, 192 \times 192, 160 \times 160, 128 \times 128$, and $96 \times 96$ and averages their momentum mixup losses. This ensures a fair comparison with CLSA and SwAV. Moreover, we use strong augmentation strategy in CLSA. Spefically, for the above small image, we randomly select an operation from 14 augmentations used in CLSA, and apply it to the image with a probability of 0.5, which is repeated 5 times. We use "(strong)" to mark whether we use strong augmentations on the small images in momentum mixup loss. Thus, SANE has almost the same training cost with CLSA, i.e. about 75 (198) hours with 8 GPUs, 200 epochs, batch size of 256 for SANE-Single (-Multi). For vanilla contrastive loss on ImageNet, we always use weak augmentations. See more details of the augmentation, loss construction, and pretraining cost on CIFAR10 and ImageNet in Appendix A.

**Results.** Table 1 shows that with weak or strong augmentations, SANE always surpasses the baselines on CIFAR10. Moreover, SANE with strong (weak) augmentation improves supervised baseline by 1.0% (0.6%).

Table 2 also shows that for ImageNet under weak augmentation setting, for 200 (800) epochs SANE-Multi respectively brings 0.8% (0.6%) improvements over SwAV; with 200 (800) epochs, SANE-Single also beats the runner-up MixCo (i-Mix and SimSiam). Note, BYOL outperforms SANE-Single but was trained 1,000 epochs. With strong augmentation, SANE-Single and SANE-Multi also respectively outperform CLSA-Single and CLSA-Multi. Moreover, our self-supervised accuracy 76.4% is very close to the accuracy 76.5% of supervised baseline, and still improves 0.2% over CLEAN-Multi even for this challenging case. These results show the superiority and robustness of SANE, thanks to its self-labeling refinery and momentum mixup which both improve label quality and thus bring semantically similar samples together.

## 4.2 Transfer Results on Downstream Tasks

**Settings.** We evaluate the pretrained SANE model on VOC [61] and COCO [62]. For classification, we train a linear classifier upon ResNet50 100 epochs by SGD. For object detection, we use the same protocol in [1] to fine-tune the pretrained ResNet50 based on detectron2 [63] for fairness. On VOC,

| Table 3: Transfer learning results. | | | |
|---|---|---|---|
| method | classification VOC07 Accuracy | object detection VOC07+12 AP$_{50}$ | COCO AP |
| NPID++ [56] | 76.6 | 79.1 | — |
| MoCo [1] | 79.8 | 81.5 | — |
| PIRL [51] | 81.1 | 80.7 | — |
| BoWNet [60] | 79.3 | 81.3 | — |
| SimCLR [4] | 86.4 | — | — |
| CO2 [20] | 85.2 | 82.7 | — |
| i-Mix [28] | — | 82.7 | — |
| MoCo v2 [2] | 87.1 | 82.5 | 42.0 |
| SWAV-Multi [5] | 88.9 | 82.6 | 42.1 |
| CLSA-Multi(strong)[12] | 93.6 | 83.2 | 42.3 |
| SANE-Multi | 92.9 | 82.9 | 42.2 |
| SANE-Multi (strong) | 94.0 | 83.4 | 42.4 |
| Supervised [12] | 87.5 | 81.3 | 40.8 |

Table 4: Effects of the components in SANE with strong augmentation on CIFAR10.

| label $p$ in (5) | label $q$ in (5) | momentum mixup | accuracy (%) |
|:---:|:---:|:---:|:---:|
| ✓ | | | 93.7 |
| | ✓ | | 94.6 |
| | | ✓ | 94.5 |
| ✓ | | ✓ | 94.8 |
| ✓ | ✓ | | 94.9 |
| ✓ | | ✓ | 95.2 |
| | ✓ | ✓ | 95.1 |
| ✓ | ✓ | ✓ | 95.9 |

Table 5: Effects of parameter $\lambda$ in SANE with strong augmentation on CIFAR10.

| regularization $\lambda$ | 0 | 0.25 | 0.5 | 0.75 | 1 |
|---|---|---|---|---|---|
| accuracy (%) | 94.3 | 95.8 | 95.9 | 95.5 | 94.5 |

Table 6: Effects of various mixups on ImageNet.

| Accuracy (%) | MoCo+mixup | MoCo+momentum mixup |
|---|---|---|
| CIFAR10 (weak) | 93.7 | 94.2 |
| CIFAR10 (strong) | 93.3 | 94.8 |
| ImageNet (weak) | 68.4 [27] | 69.0 |

we train detection head with VOC07+12 trainval data and tested on VOC07 test data. On COCO, we train the head on train2017 set and evaluate on the val2017. See optimization settings in Appendix A.

**Results.** Table 3 shows that SANE consistently outperforms the compared state-of-the-art approaches on both classification and object detection tasks, and enjoys better performance than supervised method pretrained on ImageNet. These results show the superior transferability of SANE behind which the potential reasons have been discussed in Sec. 4.1.

### 4.3  Ablation Study

We train SANE 1,000 epochs on CIFAR10 to investigate the effects of each component in SANE using strong augmentation. Table 4 shows the benefits of each component, i.e. the label estimations $p$ and $q$ in self-labeling refinery, and momentum mixup.

To investigate the robustness of our SANE to the regularization parameter $\lambda$ in (9), we run 2,000 epochs on CIFAR10, and report the performance in Table 5. From the results, one can observe that the stable performance (robustness) of SANE on CIAFR10 when regularization parameter $\lambda$ in (9) varies in a large range, thus testifying the robustness of SANE.

Then we compare our momentum mixup (8) with vanilla mixup in the works [27, 28]. Specifically, we use one-hot label in MoCo and replace $\widetilde{x}_j$ in (8) with the query $x_j$ to obtain "MoCo+ mixup", and ours with one-hot label can be viewed as "MoCo+momentum mixup". Then we train them 1,000 epochs on CIFAR10 with weak/strong augmentation, and 200 epochs on ImageNet with weak augmentations. The results in Table 6 show that with weak augmentation, momentum mixup respectively makes about $1.1\%$ and $0.6\%$ improvements over vanilla mixup in [27, 28] on CIFAR10 and ImageNet. Moreover, momentum mixup using strong augmentation has accuracy $94.8\%$ and improves its weak augmentation version, while vanilla mixup with strong augmentation suffers from performance degradation. It is because as discussed in Sec. 3.2, momentum mixup well reduces the possible label noise, especially for strong augmentations, and can enhance the performance more.

## 5  Conclusion

In this work, we prove the benefits of accurate labels to the generalization of contrastive learning. Inspired by this theory, we propose SANE to improve label quality in contrastive learning via self-labeling refinery and momentum mixup. The former uses the positive of a query to generate informative soft labels and combines with vanilla one-hot label to improve label quality. The latter randomly combines queries and positives to make virtual queries more similar to their corresponding positives, improving label accuracy. Experimental results testified the advantages of SANE.

## Acknowledgements

The authors sincerely thank the anonymous reviewers for their constructive comments on this work.

**1. Funding.** Pan Zhou, Caiming Xiong and Steven HOI are supported by Salesforce, mainly for their GPU resource support. Xiao-Tong Yuan is supported in part by the National Key Research and Development Program of China under Grant No. 2018AAA0100400 and in part by the Natural Science Foundation of China (NSFC) under Grant No.61876090 and No.61936005.

**2. Competing Interests.** Pan Zhou, Caiming Xiong and Steven HOI are staffs in Salesforce. Xiao-Tong Yuan works as a professor in Nanjing University of Information Science & Technology, Nanjing, China.

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
