# A Theory-Driven Self-Labeling Refinement Method for Contrastive Representation Learning (Supplementary File)

**Pan Zhou**[*]    **Caiming Xiong**[*]    **Xiao-Tong Yuan**[†]    **Steven Hoi**[*]

[*] Salesforce Research
[†] Nanjing University of Information Science & Technology
panzhou3@gmail.com    {cxiong, shoi}@salesforce.com    xtyuan@nuist.edu.cn

## Abstract

This supplementary document contains more additional experimental details and the technical proofs of convergence results of the NeurIPS'21 submission entitled "A Theory-Driven Self-Labeling Refinement Method for Contrastive Representation Learning". It is structured as follows. In Appendix A, we provides more experimental details, including training algorithm, network architecture, optimizer details, loss construction and training cost of SANE. Appendix B presents the proof and details of the main results, namely, Theorem 1, in Section 2, which analyzes the generalization performance of MoCo.

Next, Appendix C introduces the *proof roadmap* and details of the main results, i.e. Theorem 2, in Section 3.1. Since the proof framework is relatively complex, we first introduce some necessary preliminaries, including notations, conceptions and assumptions that are verified in subsequent analysis in Appendix C.2.4. Then we provide the proofs of Theorem 2 in Appendix C.2. Specifically, we first introduce the *proof roadmap of Theorem 2* in Appendix C.2.1. Then we present several auxiliary theories in Appendix C.2.2. Next, we prove our Theorem 2 in Appendix C.2.3. Finally, we present all proof details of auxiliary theories in Appendix C.2.4.

## A   More Experimental Details

Due to space limitation, we defer more experimental details to this appendix. Here we first introduce the training algorithm of SANE, and then present more setting details of optimizers, architectures, loss construction for CIFAR10 and ImageNet.

### A.1   Algorithm Framework of SANE

In this subsection, we introduce the training algorithm of SANE in details, which is summarized in Algorithm 1. Same as MoCo [1] and CLSA [2], we alternatively update the online network $f_{\boldsymbol{w}}$ and target network $g_{\boldsymbol{\xi}}$ via SGD optimizer. Our codes are implemented based on MoCo and CLSA. The code of MoCo and CLSA satisfies "Creative Commons Attribution-NonCommercial 4.0 International Public License".

35th Conference on Neural Information Processing Systems (NeurIPS 2021).

---

**Algorithm 1** Algorithm Framework for SANE

---

**Input:** online network $f_{\boldsymbol{w}}$, target network $g_{\boldsymbol{\xi}}$, dictionary $\boldsymbol{B}$, temperature parameter $\tau$, momentum-update parameter $\iota$, sharpness parameter $\tau'$, prior confidence $\mu$, regularization weight $\lambda$, parameter $\kappa$ for $\mathsf{Beta}(\kappa,\kappa)$, weak augmentation $T_1$, and weak or strong augmentation $T_2$

**Initialization:** initialize online network $f_{\boldsymbol{w}}$, target network $g_{\boldsymbol{\xi}}$, dictionary $\boldsymbol{B}$ as MoCo.

**for** $i = 1 \cdots T$ **do**

    1. sample a minibatch of vanilla samples $\{\boldsymbol{c}_i\}_{i=1}^s$

    2. use $T_1$ to augment $\{\boldsymbol{c}_i\}_{i=1}^s$ to obtain weak augmentations $\{(\boldsymbol{x}_i, \widetilde{\boldsymbol{x}}_i)\}_{i=1}^s$, i.e. $\boldsymbol{x}_i = T_1(\boldsymbol{c}_i)$ and $\widetilde{\boldsymbol{x}}_i = T_1(\boldsymbol{c}_i)$.

    3. compute feature $\{f(\boldsymbol{x}_i)\}_{i=1}^s$ and $\boldsymbol{B}' = \{g(\widetilde{\boldsymbol{x}}_i)\}_{i=1}^s$

    4. compute the contrastive loss $\mathcal{L}_{\mathrm{c}}\big(\boldsymbol{w}, \{(\boldsymbol{x}_i, \boldsymbol{y}_i)\}\big)$ in Eqn. (9)

    5. use $\widetilde{\boldsymbol{x}}_i$ to compute the estimated labels $\bar{\boldsymbol{y}}_i^t$ of query $\boldsymbol{x}_i$ by self-labeling refinery (5) ($\forall i = 1, \cdots, s$)

    6. if using strong augmentation for momentum mixup, use $T_2$ to augment $\{\boldsymbol{c}_i\}_{i=1}^s$ for obtaining strong augmentations $\{\widetilde{\boldsymbol{x}}_i\}_{i=1}^s$ to replace the previous $\{\widetilde{\boldsymbol{x}}_i\}_{i=1}^s$ in $\{(\boldsymbol{x}_i, \widetilde{\boldsymbol{x}}_i)\}_{i=1}^s$

    7. use momentum mixup (8) and samples $\{(\boldsymbol{x}_i, \widetilde{\boldsymbol{x}}_i, \bar{\boldsymbol{y}}_i^t)\}_{i=1}^s$ to obtain new virtual queries and labels $\{(\boldsymbol{x}_i', \boldsymbol{y}_i')\}_{i=1}^s$

    8. use $\{(\boldsymbol{x}_i', \boldsymbol{y}_i')\}_{i=1}^s$ to compute the momentum mixup contrastive loss $\mathcal{L}_{\mathrm{c}}\big(\boldsymbol{w}, \{(\boldsymbol{x}_i', \boldsymbol{y}_i')\}\big)$ in Eqn. (9)

    9. update online network $f_{\boldsymbol{w}}$ by minimizing $(1-\lambda)\mathcal{L}_{\mathrm{c}}\big(\boldsymbol{w}, \{(\boldsymbol{x}_i, \boldsymbol{y}_i)\}\big) + \lambda\mathcal{L}_{\mathrm{c}}\big(\boldsymbol{w}, \{(\boldsymbol{x}_i', \boldsymbol{y}_i')\}\big)$

    10. update target network $g_{\boldsymbol{\xi}}$ by exponential moving average

    11. update the dictionary $\boldsymbol{B}$ via minibatch feature $\boldsymbol{B}'$ in a first-in first-out order.

**end for**

**Output:**

---

## A.2   Algorithm Parameter Settings

**Experimental Settings for Linear Evaluation on CIFAR10 and ImageNet.** For CIFAR10 and ImageNet, we follow [1, 3] and use ResNet50 [4] as a backbone. Then we first pretrain SANE on the corresponding training data, and then train a linear classifier on top of 2048-dimensional frozen features provided by ResNet50. For pretraining on both datasets, we use SGD with an initial learning rate 0.03 (annealed down to zero via cosine decay [5]), a momentum of 0.9, and a weight decay of $10^{-4}$. Such optimizer parameters are the same with MoCo and CLSA.

Next, we pretrain 2,000 epochs on CIFAR10 with minibatch size 256 and dictionary size 4,096. For pretraining on Imagenet, the dictionary size is always 65,536; the batch size is often 256 on a cluster of 8 GPUs and is linearly scaled together with learning rate on multiple clusters. For linear classifier training, we use ADAM [6] with a learning rate of 0.01 and without weight decay to train 200 epochs on CIFAR10, and adopt SGD with an initial learning 10 (cosine decayed to zero) and a momentum of 0.9 to train 100 epochs on ImageNet. We use standard data augmentations in [1] for pretraining unless otherwise stated. Specifically, for pretraining on CIFAR10 and ImageNet, we follow MoCo and use RandomResizedCrop, ColorJitter, RandomGrayscale, GaussianBlur, RandomHorizontalFlip, and Normalization. For CIFAR10, please find its pretraining augmentation in the example[1]. Except the above random augmentation, we also use the proposed momentum mixup to generate the virtual instances for constructing the momentum mixup loss.

For CIFAR10, to fairly compare with [7], we crop each image into two views to construct the loss (9). Specifically, for a minibatch of vanilla samples $\{\boldsymbol{c}_i\}_{i=1}^s$, we use weak augmentation $T_1$ to augment $\{\boldsymbol{c}_i\}_{i=1}^s$ to obtain weak augmentations $\{(\boldsymbol{x}_i, \widetilde{\boldsymbol{x}}_i)\}_{i=1}^s$, i.e. $\boldsymbol{x}_i = T_1(\boldsymbol{c}_i)$ and $\widetilde{\boldsymbol{x}}_i = T_1(\boldsymbol{c}_i)$. Then same as MoCo, we can compute the contrastive loss by using $\{(\boldsymbol{x}_i, \widetilde{\boldsymbol{x}}_i)\}_{i=1}^s$. Meanwhile, we use $\widetilde{\boldsymbol{x}}_i$ to compute the soft label $\bar{\boldsymbol{y}}_i^t$ of $\widetilde{\boldsymbol{x}}_i$ via (5). Next, we use momentum mixup (8) and samples $\{(\boldsymbol{x}_i, \widetilde{\boldsymbol{x}}_i, \bar{\boldsymbol{y}}_i^t)\}_{i=1}^s$ to obtain new virtual queries and labels $\{(\boldsymbol{x}_i', \boldsymbol{y}_i')\}_{i=1}^s$, and then use $\{(\boldsymbol{x}_i', \boldsymbol{y}_i')\}_{i=1}^s$ to compute the momentum mixup contrastive loss $\mathcal{L}_{\mathrm{c}}\big(\boldsymbol{w}, \{(\boldsymbol{x}_i', \boldsymbol{y}_i')\}\big)$ in Eqn. (9). For strong augmentation, after we compute the vanilla contrastive loss in MoCo, and then use strong augmentation to augment $\{\boldsymbol{c}_i\}_{i=1}^s$ to replace $\widetilde{\boldsymbol{x}}_i$ in $\{(\boldsymbol{x}_i, \widetilde{\boldsymbol{x}}_i, \bar{\boldsymbol{y}}_i^t)\}_{i=1}^s$. Then we can generate virtual

---

[1]https://colab.research.google.com/github/facebookresearch/moco/blob/colab-notebook/colab/moco_cifar10_demo.ipynb

query instances and their labels ($\{(\boldsymbol{x}_i', \boldsymbol{y}_i')\}_{i=1}^s$) by using $\{(\boldsymbol{x}_i, \widetilde{\boldsymbol{x}}_i, \bar{\boldsymbol{y}}_i^t)\}_{i=1}^s$. The training cost on CIFAR10 for 2,000 epochs is about 11 days on single V100 GPU.

For ImageNet, we follow CLSA for fair comparison. For SANE-Single, we use the same way to construct the contrastive loss, and then use augmentation $T_1$ to augment $\{\boldsymbol{c}_i\}_{i=1}^s$ to replace $\widetilde{\boldsymbol{x}}_i$ in $\{(\boldsymbol{x}_i, \widetilde{\boldsymbol{x}}_i, \bar{\boldsymbol{y}}_i^t)\}_{i=1}^s$ to construct the momentum mixup loss. Indeed, we also can do not replace $\widetilde{\boldsymbol{x}}_i$ in $\{(\boldsymbol{x}_i, \widetilde{\boldsymbol{x}}_i, \bar{\boldsymbol{y}}_i^t)\}_{i=1}^s$ for momentum mixup loss, which actually did not affect the performance. We do it, since SANE-Multi crops each image into five different crops for constructing momentum mixup loss, and thus SANE-Single and SANE-Multi will be more consistent, i.e. SANE-Multi uses 5 crops while SANE-Single uses one crop. For strong augmentation, we replace the augmentation $T_1$ in momentum mixup with strong augmentation, which is the same on CIFAR10. As mentioned above, to construct the momentum mixup loss, SANE-Multi crops each image into five sizes $224 \times 224$, $192 \times 192$, $160 \times 160$, $128 \times 128$, and $96 \times 96$ and averages their momentum mixup losses. For the vanilla contrastive loss, SANE-Multi uses the same way in SANE-Single to compute. In this way, SANE-Single and SANE-Multi respectively have the same settings with CLSA-Single and CLSA-Multi. Thus, ELSE has almost the same training cost with CLSA, i.e. about 75 (188) hours with 8 GPUs, 200 epochs, batch size of 256 for SANE-Single (-Multi). It should be mentioned that for vanilla contrastive loss in both CLSA-Single and CLSA-Multi, we always use weak augmentations.

**Transfer Evaluation Settings.** We evaluate the pretrained model on ImageNet on VOC [8] and COCO [9]. For VOC, similar to linear evaluation, we train a linear classifier upon ResNet50 100 epochs by SGD with a learning rate 0.05, a momentum 0.9, batch size 256, and without weight and learning rate decay. For COCO, we adopt the same protocol in [1] to fine-tune the pretrained ResNet50 based on detectron2 [10] for fairness. We evaluate the transfer ability of the cells selected on CIFAR10 by testing them on ImageNet. Following DARTS, we use momentum SGD with an initial learning 0.025 (cosine decayed to zero), a momentum of 0.9, a weight decay of $3 \times 10^{-4}$, and gradient norm clipping parameter 5.0.

# B   Proofs of The Results in Section 2

**Lemma 1.** *[11] Suppose the loss $\ell$ is bounded by the range $[a, b]$, namely $\ell(f(\boldsymbol{x}; \boldsymbol{w}), \boldsymbol{y}) \in [a, b]$. Then let $\mathcal{F}$ be a finite class of hypotheses $\ell(f(\boldsymbol{x}; \boldsymbol{w}), \boldsymbol{y}) : \boldsymbol{\mathcal{X}} \to \mathbb{R}$. Let*

$$\boldsymbol{\mathcal{Q}}_e(f) = \frac{1}{n} \sum_{i=1}^n \ell(f(\boldsymbol{x}_i; \boldsymbol{w}), \boldsymbol{y}_i), \quad \boldsymbol{\mathcal{Q}}(f) = \mathbb{E}_{(\boldsymbol{x}, \boldsymbol{y}) \in \boldsymbol{\mathcal{S}}} [\ell(f(\boldsymbol{x}; \boldsymbol{w}), \boldsymbol{y})]$$

*respectively denote the empirical and population risk, where $\boldsymbol{\mathcal{S}}$ denote the unknown data distribution and the sampled dataset $\boldsymbol{\mathcal{D}} = \{(\boldsymbol{x}_i, \boldsymbol{y}_i)\}_{i=1}^n \sim \boldsymbol{\mathcal{S}}$ is of size $n$. Then for any $\delta \in (0, 1)$, with probability at least $1 - \delta$ we have*

$$\boldsymbol{\mathcal{Q}}(f) \le \boldsymbol{\mathcal{Q}}_e(f) + \sqrt{\frac{2(b-a)^2 V_{\boldsymbol{\mathcal{D}}} \ln(2|\mathcal{F}|/\delta)}{n}} + \frac{7(b-a)^2 \ln(2|\mathcal{F}|/\delta)}{3(n-1)}, \tag{7}$$

*where $V_{\boldsymbol{\mathcal{D}}}$ denotes the variance of the loss $\ell(f(\boldsymbol{x}; \boldsymbol{w}), \boldsymbol{y})$ on the dataset $\boldsymbol{\mathcal{D}}$, and $|\mathcal{F}|$ denotes the covering number of $\mathcal{F}$ in the uniform norm $\|\cdot\|_\infty$.*

**Lemma 2.** *[12] For any polynomials $f(x) = \sum_{i=0}^p a_i x^i$, $x \in [0,]$ and $\sum_{i=1}^p |a_i| < 1$, there exists a multilayer neural network $\hat{f}(x)$ with $\mathcal{O}\left(p + \log \frac{p}{\epsilon}\right)$ layers, $O(\log \frac{p}{\epsilon})$ binary step units and $O(p \log \frac{p}{\epsilon})$ rectifier linear units such that $|f(x) - \hat{f}(x)| \le \epsilon$, $\forall x \in [0, 1]$.*
*Assume that function $f$ is continuous on $[0, 1]$ and $\lceil \log \frac{2}{\epsilon} \rceil + 1$ times differential in $(0, 1)$. Let $f^{(n)}$ denote the derivative of $f$ of $n$-th order and $\|f\| = \max_{x \in [0, 1]} f(x)$. If $\|f^{(n)}\| \le n!$ holds for all $n \in [\lceil \log \frac{2}{\epsilon} \rceil + 1]$, then there exists a deep network $f$ with $\mathcal{O}\left(\log \frac{1}{\epsilon}\right)$ layers, $O(\log \frac{1}{\epsilon})$ binary step units and $O(\log^2 \frac{1}{\epsilon})$ rectifier linear units such that $|f(x) - \hat{f}(x)| \le \epsilon$, $\forall x \in [0, 1]$.*

For expression power analysis of deep network, more stronger results can be found in [13, 14, 15, 16] and all show that any function can be approximately can be approximated by a deep network to arbitrary accuracy.

### B.1 Proof of Theorem 1

*Proof.* Here we use two steps to prove our results in Theorem 1.

$$\widetilde{\mathcal{Q}}(f_{\boldsymbol{w}}) = \frac{1}{n} \sum_{i=1}^{n} \ell(h(f_{\boldsymbol{w}}(\boldsymbol{x}_i), \boldsymbol{B}_i), \boldsymbol{y}_i), \tag{8}$$

**Step 1. proof for first part results.** To begin with, we first define an empirical risk $\mathcal{Q}_e(f)$:

$$\mathcal{Q}_e(f) = \frac{1}{n} \sum_{i=1}^{n} \ell(h(f_{\boldsymbol{w}}(\boldsymbol{x}_i), \boldsymbol{B}_i), \boldsymbol{y}_i^*),$$

where $\mathcal{Q}_e(f)$ uses the ground truth label $\boldsymbol{y}_i^*$ for training. From Lemma 1, with probability at least $1 - \delta$, we have

$$\mathcal{Q}(f) \leq \mathcal{Q}_e(f) + \sqrt{\frac{2(b-a)^2 V_{\boldsymbol{\mathcal{D}}} \ln(2|\mathcal{F}|/\delta)}{n}} + \frac{7(b-a)^2 \ln(2|\mathcal{F}|/\delta)}{3(n-1)},$$

where $\mathcal{Q}(f)$ is the population risk, and $\mathcal{Q}_e(f)$ is the empirical risk. Both are trained with the ground truth $\boldsymbol{y}_i^*$. So the remaining work is to upper bound $\mathcal{Q}_e(f)$ via $\widetilde{\mathcal{Q}}(f)$. Towards this end, we can bound it as follows

$$\begin{aligned}
\mathcal{Q}_e(f) - \widetilde{\mathcal{Q}}(f) =& \frac{1}{n} \sum_{i=1}^{n} \left( \ell(h(f_{\boldsymbol{w}}(\boldsymbol{x}_i), \boldsymbol{B}_i), \boldsymbol{y}_i^*) - \ell(h(f_{\boldsymbol{w}}(\boldsymbol{x}_i), \boldsymbol{B}_i), \boldsymbol{y}_i)) \right) \\
\overset{\text{①}}{\leq}& \frac{1}{n} \sum_{i=1}^{n} \|\nabla_{\boldsymbol{y}} \ell(h(f_{\boldsymbol{w}}(\boldsymbol{x}_i), \boldsymbol{B}_i), \boldsymbol{y})\| \cdot \|\boldsymbol{y}_i^* - \boldsymbol{y}_i\|_2 \\
\overset{\text{②}}{\leq}& L_y \mathbb{E}_i \|\boldsymbol{y}_i^* - \boldsymbol{y}_i\|_2 \\
\overset{\text{②}}{\leq}& L_y \mathbb{E}_{\boldsymbol{\mathcal{D}} \sim \boldsymbol{\mathcal{S}}} \left[ \|\boldsymbol{y}^* - \boldsymbol{y}\|_2 \right],
\end{aligned}$$

where ① holds by using $\boldsymbol{y} = \boldsymbol{y}_i + \theta(\boldsymbol{y}_i^* - \boldsymbol{y}_i)$ for certain $\theta \in (0, 1)$; ② holds since we use the $L_y$-Lipschitz property of $\ell(h(f_{\boldsymbol{w}}(\boldsymbol{x}_i), \boldsymbol{B}_i), \boldsymbol{y}_i)$. Then combining these results together, we can obtain the desired results:

$$|\mathcal{Q}(f) - \widetilde{\mathcal{Q}}(f)| \leq + L_y \mathbb{E}_{\boldsymbol{\mathcal{D}} \sim \boldsymbol{\mathcal{S}}} \|\boldsymbol{y}_i^* - \boldsymbol{y}_i\|_2 + \sqrt{\frac{2(b-a)^2 V_{\boldsymbol{\mathcal{D}}} \ln(2|\mathcal{F}|/\delta)}{n}} + \frac{7(b-a)^2 \ln(2|\mathcal{F}|/\delta)}{3(n-1)}.$$

**Step 2. proof for second part results.** Here we can construct a simple two-classification problem for clarity. Suppose we have two classes: class one with training data $\mathcal{D}_1 = \{(\boldsymbol{x}_1, \boldsymbol{x}_1, \boldsymbol{y}_1^*)\}_{i=1}^{n/2}$ and class two with training data $\mathcal{D}_2 = \{(\boldsymbol{x}_2, \boldsymbol{x}_2, \boldsymbol{y}_2^*)\}_{i=1}^{n/2}$, where $\boldsymbol{y}_1^*$ denotes the ground truth label of $\boldsymbol{x}_1$ on the set $\boldsymbol{B}_1 = \{\boldsymbol{x}_1 \cup \boldsymbol{B}\}$, and $\boldsymbol{y}_2^*$ denotes the ground truth label of $\boldsymbol{x}_2$ on the set $\boldsymbol{B}_2 = \{\boldsymbol{x}_2 \cup \boldsymbol{B}\}$. Both training datasets $\mathcal{D}_1$ and $\mathcal{D}_2$ have $\frac{n}{2}$ samples. Here we assume there is no data augmentation which means $\boldsymbol{x}_i = \widetilde{\boldsymbol{x}}_i$ in the manuscript. In $\mathcal{D}_1$, its samples are the same, namely $(\boldsymbol{x}_1, \boldsymbol{x}_1, \boldsymbol{y}_1^*)$. Similarly, $\mathcal{D}_2$ also has the same samples, namely $(\boldsymbol{x}_2, \boldsymbol{x}_2, \boldsymbol{y}_2^*)$. Then the predicted class probability $\boldsymbol{y}_{ij}$ of sample $\boldsymbol{x}_i$ on class $j$ is as follows:

$$\boldsymbol{y}_{i0} = \frac{e^{\delta(\boldsymbol{x}_i, \boldsymbol{x}_i)/t}}{e^{\delta(\boldsymbol{x}_i, \boldsymbol{x}_i)/\tau} + \sum_{j=1}^{k} e^{\delta(\boldsymbol{x}_i, \boldsymbol{b}_j)/\tau}}, \quad \boldsymbol{y}_{ij} = \frac{e^{\delta(\boldsymbol{x}_i, \boldsymbol{b}_j)/\tau}}{e^{\delta(\boldsymbol{x}_i, \boldsymbol{x}_i)/\tau} + \sum_{j=1}^{k} e^{\delta(\boldsymbol{x}_i, \boldsymbol{b}_j)/\tau}} \, (j = 1, \cdots, k), \tag{9}$$

where $\delta(\boldsymbol{x}_i, \widetilde{\boldsymbol{x}}_i) = -\frac{\langle f(\boldsymbol{x}_i), g(\widetilde{\boldsymbol{x}}_i) \rangle}{\|f(\boldsymbol{x}_i)\|_2 \cdot \|g(\widetilde{\boldsymbol{x}}_i)\|_2}$, $\tau$ denotes a temperature. For simplicity, we let dictionary $\boldsymbol{B} = \{\boldsymbol{x}_1, \boldsymbol{x}_2\}$. In this way, we have for both ground truth label $\boldsymbol{y}_1^*$ and $\boldsymbol{y}_2^*$ that satisfy $\boldsymbol{y}_{10}^* = \boldsymbol{y}_{11}^*$, $\boldsymbol{y}_{10}^* + \boldsymbol{y}_{11}^* + \boldsymbol{y}_{12}^* = 1$, $\boldsymbol{y}_{20}^* = \boldsymbol{y}_{22}^*$, $\boldsymbol{y}_{20}^* + \boldsymbol{y}_{21}^* + \boldsymbol{y}_{22}^* = 1$. For this setting, here we assume the training labels are denoted by $\boldsymbol{y}_1$ and $\boldsymbol{y}_2$. Moreover, they satisfy $\boldsymbol{y}_{10} = \boldsymbol{y}_{11} > 0$, $\boldsymbol{y}_{10} + \boldsymbol{y}_{11} + \boldsymbol{y}_{12} = 1$, $\boldsymbol{y}_{20} = \boldsymbol{y}_{22} > 0$, $\boldsymbol{y}_{20} + \boldsymbol{y}_{21} + \boldsymbol{y}_{22} = 1$. The reason that we do not use one-hot labels. This is because for dictionary $\boldsymbol{B} = \{\boldsymbol{x}_1, \boldsymbol{x}_2\}$, given a sample $\boldsymbol{x}_i \, (i = 1, 2)$, $\boldsymbol{x}_i$ needs to predict the labels on the set $\{\boldsymbol{x}_i \cup \boldsymbol{B}\} = \{\boldsymbol{x}_i, \boldsymbol{x}_1, \boldsymbol{x}_2\}$, where the labels are not one-hot obviously and satisfy $\boldsymbol{y}_{i1} = \boldsymbol{y}_{ii} > 0$. In the following, we will train the model on the training data $\widetilde{\mathcal{D}} = \widetilde{\mathcal{D}}_1 \cup \widetilde{\mathcal{D}}_2$ where $\widetilde{\mathcal{D}}_1 = \{(\boldsymbol{x}_1, \boldsymbol{x}_1, \boldsymbol{y}_1)\}$ and $\widetilde{\mathcal{D}}_2 = \{(\boldsymbol{x}_2, \boldsymbol{x}_2, \boldsymbol{y}_2)\}$. We use $\widetilde{\boldsymbol{y}}_i$ to denote the model predicted label of $\boldsymbol{x}_i$.

Then for the test samples, we assume that half of samples are $(\boldsymbol{x}_1, \boldsymbol{x}_1, \boldsymbol{y}_1^*)$ and remaining samples are $(\boldsymbol{x}_2, \boldsymbol{x}_2, \boldsymbol{y}_2^*)$. Then for any network $f$, we always have

$$\boldsymbol{\mathcal{Q}}(f) - \boldsymbol{\mathcal{Q}}_e(f) = \frac{1}{n}\sum_{i=1}^{n}\left(\mathbb{E}[\ell(h(f_{\boldsymbol{w}}(\boldsymbol{x}_i), \boldsymbol{B}_i), \boldsymbol{y}_i^*)] - \ell(h(f_{\boldsymbol{w}}(\boldsymbol{x}_i), \boldsymbol{B}_i), \boldsymbol{y}_i^*)\right) = 0.$$

Then we attempt to lower bound $\boldsymbol{\mathcal{Q}}_e(f) - \widetilde{\boldsymbol{\mathcal{Q}}}(f)$. Our training dataset is $\widetilde{\mathcal{D}} = \widetilde{\mathcal{D}}_1 \cup \widetilde{\mathcal{D}}_2$ where $\widetilde{\mathcal{D}}_1 = \{(\boldsymbol{x}_1, \boldsymbol{x}_1, \boldsymbol{y}_1)\}$ and $\widetilde{\mathcal{D}}_2 = \{(\boldsymbol{x}_2, \boldsymbol{x}_2, \boldsymbol{y}_2)\}$. Then we discuss whether the network $f$ can perfectly fit the labels (9) of data $\widetilde{\mathcal{D}}$. For both cases, our results can hold.

**Perfectly fitting.** Network $f$ has the capacity to perfectly fit the label $\widetilde{\boldsymbol{y}}_1$ in $\widetilde{\mathcal{D}}_1$ and the label $\widetilde{\boldsymbol{y}}_2$ in $\widetilde{\mathcal{D}}_2$ when $\boldsymbol{x}_1$ are different $\boldsymbol{x}_2$. In this case, we have

$$
\begin{aligned}
&\boldsymbol{\mathcal{Q}}_e(f) - \widetilde{\boldsymbol{\mathcal{Q}}}(f) \\
=& \frac{1}{n}\sum_{i=1}^{n}\left(\ell(h(f_{\boldsymbol{w}}(\boldsymbol{x}_i), \boldsymbol{B}_i), \boldsymbol{y}_i^*) - \ell(h(f_{\boldsymbol{w}}(\boldsymbol{x}_i), \boldsymbol{B}_i), \boldsymbol{y}_i)\right) \\
=& \frac{1}{n}\sum_{i=1}^{n}\sum_{s=1}^{k}\left(\boldsymbol{y}_{i,s}^* \log(h(f_{\boldsymbol{w}}(\boldsymbol{x}_i), \boldsymbol{B}_i)) - \boldsymbol{y}_{i,s}\log(h(f_{\boldsymbol{w}}(\boldsymbol{x}_i), \boldsymbol{B}_i))\right) \\
=& \frac{1}{n}\sum_{i=1}^{n}\sum_{s=1}^{k}(\boldsymbol{y}_{i,s}^* - \boldsymbol{y}_{i,s})\log(\widetilde{\boldsymbol{y}}_{i,s}) \\
\overset{\text{①}}{=}& \frac{1}{n}\sum_{i=1}^{n}\sum_{s=1}^{k}(\boldsymbol{y}_{i,s}^* - \boldsymbol{y}_{i,s})\log(\boldsymbol{y}_{i,s}) \\
=& \frac{1}{6}\big[(\boldsymbol{y}_{10}^* - \boldsymbol{y}_{10})\log(\boldsymbol{y}_{10}) + (\boldsymbol{y}_{11}^* - \boldsymbol{y}_{11})\log(\boldsymbol{y}_{11}) + (\boldsymbol{y}_{12}^* - \boldsymbol{y}_{12})\log(\boldsymbol{y}_{12}) \\
&+ (\boldsymbol{y}_{20}^* - \boldsymbol{y}_{20})\log(\boldsymbol{y}_{20}) + (\boldsymbol{y}_{21}^* - \boldsymbol{y}_{21})\log(\boldsymbol{y}_{21}) + (\boldsymbol{y}_{22}^* - \boldsymbol{y}_{22})\log(\boldsymbol{y}_{22})\big] \\
=& \frac{1}{6}\big[2(\boldsymbol{y}_{10}^* - \boldsymbol{y}_{10})\log(\boldsymbol{y}_{10}) + (\boldsymbol{y}_{12}^* - \boldsymbol{y}_{12})\log(\boldsymbol{y}_{12}) + 2(\boldsymbol{y}_{20}^* - \boldsymbol{y}_{20})\log(\boldsymbol{y}_{20}) + (\boldsymbol{y}_{21}^* - \boldsymbol{y}_{21})\log(\boldsymbol{y}_{21})\big] \\
\overset{\text{②}}{=}& \frac{1}{3}\left[(\boldsymbol{y}_{10}^* - \boldsymbol{y}_{10})\log\frac{\boldsymbol{y}_{10}}{1 - 2\boldsymbol{y}_{10}} + (\boldsymbol{y}_{20}^* - \boldsymbol{y}_{20})\log\frac{\boldsymbol{y}_{20}}{1 - 2\boldsymbol{y}_{20}}\right],
\end{aligned}
$$

where ① holds since $\widetilde{\boldsymbol{y}}_{i,s} = \boldsymbol{y}_{i,s}$, and ② uses $\boldsymbol{y}_{10}^* = \boldsymbol{y}_{11}^*$, $\boldsymbol{y}_{10}^* + \boldsymbol{y}_{11}^* + \boldsymbol{y}_{12}^* = 1$, $\boldsymbol{y}_{20}^* = \boldsymbol{y}_{22}^*$, $\boldsymbol{y}_{20}^* + \boldsymbol{y}_{21}^* + \boldsymbol{y}_{22}^* = 1$, $\boldsymbol{y}_{10} = \boldsymbol{y}_{11}$, $\boldsymbol{y}_{10} + \boldsymbol{y}_{11} + \boldsymbol{y}_{12} = 1$, $\boldsymbol{y}_{20} = \boldsymbol{y}_{22}$, $\boldsymbol{y}_{20} + \boldsymbol{y}_{21} + \boldsymbol{y}_{22} = 1$. Then we can choose proper values such that

$$\boldsymbol{y}_{10}^* = \boldsymbol{y}_{11}^* > \boldsymbol{y}_{10} = \boldsymbol{y}_{11} > \frac{1}{3}, \boldsymbol{y}_{20}^* = \boldsymbol{y}_{22}^* > \boldsymbol{y}_{20} = \boldsymbol{y}_{22} > \frac{1}{3}.$$

For example, we can let $\boldsymbol{y}_1 = (0.4, 0.4, 0.2)$, $\boldsymbol{y}_1^* = (0.45, 0.45, 0.1)$, $\boldsymbol{y}_2 = (0.4, 0.2, 0.4)$, $\boldsymbol{y}_2^* = (0.45, 0.1, 0.45)$. In this way, we have $(\boldsymbol{y}_{10}^* - \boldsymbol{y}_{10})\log\frac{\boldsymbol{y}_{10}}{1 - 2\boldsymbol{y}_{10}} \geq c_1(\boldsymbol{y}_{10}^* - \boldsymbol{y}_{10}) > 0$ and $(\boldsymbol{y}_{20}^* - \boldsymbol{y}_{20})\log\frac{\boldsymbol{y}_{20}}{1 - 2\boldsymbol{y}_{20}} \geq c_2(\boldsymbol{y}_{20}^* - \boldsymbol{y}_{20}) > 0$. So this means that there exists a constant $C$ such that

$$\boldsymbol{\mathcal{Q}}_e(f) - \widetilde{\boldsymbol{\mathcal{Q}}}(f) \geq C \cdot \mathbb{E}_i\left[\|\boldsymbol{y}_i^* - \boldsymbol{y}_i\|_2\right] = C \cdot \mathbb{E}_{\mathcal{D}\sim\mathcal{S}}\left[\|\boldsymbol{y}^* - \boldsymbol{y}\|_2\right].$$

So combining the above results gives the following desired result:

$$\boldsymbol{\mathcal{Q}}(f) - \widetilde{\boldsymbol{\mathcal{Q}}}(f) \geq C \cdot \mathbb{E}_{\mathcal{D}\sim\mathcal{S}}\left[\|\boldsymbol{y}^* - \boldsymbol{y}\|_2\right].$$

**Non-perfectly fitting.** From Lemma 2 (other more results in [13, 14, 15, 16]), one can approximate any function by a deep network to arbitrary accuracy. Specifically, for the polynomial function in Eqn. (9), there exists a multilayer neural network $\hat{f}(x)$ with proper width and depth such that $\|\boldsymbol{y}_1 - \widetilde{\boldsymbol{y}}_1\|_1 \leq \epsilon$ and $\|\boldsymbol{y}_2 - \widetilde{\boldsymbol{y}}_2\|_1 \leq \epsilon$, where $\widetilde{\boldsymbol{y}}_1$ and $\widetilde{\boldsymbol{y}}_2$ are the predicted labels of samples $\boldsymbol{x}_1$ and $\boldsymbol{x}_2$ by using (9). The labels $\boldsymbol{y}_1$ and $\boldsymbol{y}_2$ are associated with our training dataset $\widetilde{\mathcal{D}} = \widetilde{\mathcal{D}}_1 \cup \widetilde{\mathcal{D}}_2$ where

$\widetilde{\mathcal{D}}_1 = \{(\boldsymbol{x}_1, \boldsymbol{x}_1, \boldsymbol{y}_1)\}$ and $\widetilde{\mathcal{D}}_2 = \{(\boldsymbol{x}_2, \boldsymbol{x}_2, \boldsymbol{y}_2)\}$. In this case, we have

$$\mathcal{Q}_e(f) - \widetilde{\mathcal{Q}}(f)$$

$$= \frac{1}{n} \sum_{i=1}^{n} \left( \ell(h(f_{\boldsymbol{w}}(\boldsymbol{x}_i), \boldsymbol{B}_i), \boldsymbol{y}_i^*) - \ell(h(f_{\boldsymbol{w}}(\boldsymbol{x}_i), \boldsymbol{B}_i), \boldsymbol{y}_i) \right)$$

$$= \frac{1}{n} \sum_{i=1}^{n} \sum_{s=1}^{k} (\boldsymbol{y}_{i,s}^* \log(h(f_{\boldsymbol{w}}(\boldsymbol{x}_i), \boldsymbol{B}_i)) - \boldsymbol{y}_{i,s} \log(h(f_{\boldsymbol{w}}(\boldsymbol{x}_i), \boldsymbol{B}_i)))$$

$$= \frac{1}{n} \sum_{i=1}^{n} \sum_{s=1}^{k} (\boldsymbol{y}_{i,s}^* - \boldsymbol{y}_{i,s}) \log(\widetilde{\boldsymbol{y}}_{i,s})$$

$$= \frac{1}{6} \left[ (\boldsymbol{y}_{10}^* - \boldsymbol{y}_{10}) \log(\widetilde{\boldsymbol{y}}_{10}) + (\boldsymbol{y}_{11}^* - \boldsymbol{y}_{11}) \log(\widetilde{\boldsymbol{y}}_{11}) + (\boldsymbol{y}_{12}^* - \boldsymbol{y}_{12}) \log(\widetilde{\boldsymbol{y}}_{12}) \right.$$

$$\left. + (\boldsymbol{y}_{20}^* - \boldsymbol{y}_{20}) \log(\widetilde{\boldsymbol{y}}_{20}) + (\boldsymbol{y}_{21}^* - \widetilde{\boldsymbol{y}}_{21}) \log(\widetilde{\boldsymbol{y}}_{21}) + (\boldsymbol{y}_{22}^* - \boldsymbol{y}_{22}) \log(\widetilde{\boldsymbol{y}}_{22}) \right]$$

$$= \frac{1}{6} \left[ 2(\boldsymbol{y}_{10}^* - \boldsymbol{y}_{10}) \log(\widetilde{\boldsymbol{y}}_{10}) + (\boldsymbol{y}_{12}^* - \boldsymbol{y}_{12}) \log(\widetilde{\boldsymbol{y}}_{12}) + 2(\boldsymbol{y}_{20}^* - \boldsymbol{y}_{20}) \log(\widetilde{\boldsymbol{y}}_{20}) + (\boldsymbol{y}_{21}^* - \boldsymbol{y}_{21}) \log(\widetilde{\boldsymbol{y}}_{21}) \right]$$

$$\overset{①}{=} \frac{1}{3} \left[ (\boldsymbol{y}_{10}^* - \boldsymbol{y}_{10}) \log \frac{\widetilde{\boldsymbol{y}}_{10}}{1 - 2\widetilde{\boldsymbol{y}}_{10}} + (\boldsymbol{y}_{20}^* - \boldsymbol{y}_{20}) \log \frac{\widetilde{\boldsymbol{y}}_{20}}{1 - 2\widetilde{\boldsymbol{y}}_{20}} \right],$$

where ① uses $\boldsymbol{y}_{10}^* = \boldsymbol{y}_{11}^*$, $\boldsymbol{y}_{10}^* + \boldsymbol{y}_{11}^* + \boldsymbol{y}_{12}^* = 1$, $\boldsymbol{y}_{20}^* = \boldsymbol{y}_{21}^*$, $\boldsymbol{y}_{20}^* + \boldsymbol{y}_{21}^* + \boldsymbol{y}_{22}^* = 1$, $\boldsymbol{y}_{10} = \boldsymbol{y}_{11}$, $\boldsymbol{y}_{10} + \boldsymbol{y}_{11} + \boldsymbol{y}_{12} = 1$, $\boldsymbol{y}_{20} = \boldsymbol{y}_{22}$, $\boldsymbol{y}_{20} + \boldsymbol{y}_{21} + \boldsymbol{y}_{22} = 1$. Then we can choose proper values such that

$$\boldsymbol{y}_{10}^* = \boldsymbol{y}_{11}^* > \boldsymbol{y}_{10} = \boldsymbol{y}_{11} > \frac{1}{3} + \epsilon, \boldsymbol{y}_{20}^* = \boldsymbol{y}_{22}^* > \boldsymbol{y}_{20} = \boldsymbol{y}_{22} > \frac{1}{3} + \epsilon.$$

For example, we can let $\boldsymbol{y}_1 = (0.4, 0.4, 0.2)$, $\boldsymbol{y}_1^* = (0.45, 0.45, 0.1)$, $\boldsymbol{y}_2 = (0.4, 0.2, 0.4)$, $\boldsymbol{y}_2^* = (0.45, 0.1, 0.45)$, and $\epsilon = 0.0001$. In this way, we have $(\boldsymbol{y}_{10}^* - \boldsymbol{y}_{10}) \log \frac{\widetilde{\boldsymbol{y}}_{10}}{1 - 2\widetilde{\boldsymbol{y}}_{10}} \ge c_1(\boldsymbol{y}_{10}^* - \boldsymbol{y}_{10}) > 0$ and $(\boldsymbol{y}_{20}^* - \boldsymbol{y}_{20}) \log \frac{\widetilde{\boldsymbol{y}}_{20}}{1 - 2\widetilde{\boldsymbol{y}}_{20}} \ge c_2(\boldsymbol{y}_{20}^* - \boldsymbol{y}_{20}) > 0$. So this means that there exists a constant $C$ such that

$$\mathcal{Q}_e(f) - \widetilde{\mathcal{Q}}(f) \ge C \cdot \mathbb{E}_i \left[ \|\boldsymbol{y}_i^* - \boldsymbol{y}_i\|_2 \right] = C \cdot \mathbb{E}_{\mathcal{D} \sim \mathcal{S}} \left[ \|\boldsymbol{y}^* - \boldsymbol{y}\|_2 \right].$$

So combining the above results gives the following desired result:

$$\mathcal{Q}(f) - \widetilde{\mathcal{Q}}(f) \ge C \cdot \mathbb{E}_{\mathcal{D} \sim \mathcal{S}} \left[ \|\boldsymbol{y}^* - \boldsymbol{y}\|_2 \right].$$

The proof is completed. $\qquad\qquad\square$

## C   Proof of Results in Section 3.1

In this section, we first introduce some necessary preliminaries, including notations, conceptions and assumptions that are verified in subseqent analysis in Appendix C.2.4. Then we provide the proofs of Theorem 2 in Appendix C.2. Specifically, we first introduce the proof roadmap in Appendix C.2.1. Then we present several auxiliary theories in Appendix C.2.2. Next, we prove our Theorem 2 in Appendix C.2.3. Finally, we present all proof details of auxiliary theories in Appendix C.2.2.

### C.1   Preliminaries

#### C.1.1   General Model Formulation

In this section, we outline our approach to proving robustness of overparameterized neural networks. Towards this goal, we consider a general formulation where we aim to fit a general nonlinear model of the form $\boldsymbol{x} \mapsto f(\boldsymbol{w}, \boldsymbol{x})$ with $\boldsymbol{w} \in \mathbb{R}^p$ denoting the parameters of the model. For instance in the case of neural networks $\boldsymbol{w}$ represents its weights. Given a data set of $n$ input/label pairs $\{(\boldsymbol{x}_i, \boldsymbol{y}_i)\}_{i=1}^{n} \subset \mathbb{R}^d \times \mathbb{R}$, we fit to this data by minimizing a nonlinear least-squares loss of the form

$$\mathcal{L}_t(\boldsymbol{w}) = \frac{1}{2} \sum_{i=1}^{n} (\bar{\boldsymbol{y}}_i^t - f(\boldsymbol{w}, \boldsymbol{x}_i))^2.$$

where $\bar{\boldsymbol{y}}_i^t = (1-\alpha_t)\boldsymbol{y}_i + \alpha_t \boldsymbol{p}^t = (1-\alpha_t)\boldsymbol{y}_i + \alpha_t f(\boldsymbol{w}_t, \boldsymbol{x}_i)$ denotes the estimated label of sample $\boldsymbol{x}_i$. In Assumption 2 we assume $\beta_t = 0$ and $\tau' = 1$ for simplicity, since performing nonlinear mapping on network output greatly increases analysis difficulty. But we will show that even though $\beta_t = 0$ and $\tau' = 1$, our refinery (5) is still sufficient to refine labels. It can also be written in the more compact form

$$\mathcal{L}_t(\boldsymbol{w}) = \frac{1}{2}\left\|f(\boldsymbol{w}) - \bar{\boldsymbol{y}}^t\right\|_{\ell_2}^2 \quad \text{with} \quad f(\boldsymbol{w}) := \begin{bmatrix} f(\boldsymbol{w}, \boldsymbol{x}_1) \\ f(\boldsymbol{w}, \boldsymbol{x}_2) \\ \vdots \\ f(\boldsymbol{w}, \boldsymbol{x}_n) \end{bmatrix}. \tag{10}$$

To solve this problem we run gradient descent iterations with a constant learning rate $\eta$ starting from an initial point $\boldsymbol{w}_0$. These iterations take the form

$$\boldsymbol{w}_{t+1} = \boldsymbol{w}_t - \eta \nabla \mathcal{L}_t(\boldsymbol{w}_t) \quad \text{with} \quad \nabla \mathcal{L}(\boldsymbol{w}) = \mathcal{J}^T(\boldsymbol{w})\left(f(\boldsymbol{w}) - \bar{\boldsymbol{y}}^t\right). \tag{11}$$

Here, $\mathcal{J}(\boldsymbol{w})$ is the $n \times p$ Jacobian matrix associated with the nonlinear mapping $f$ defined via

$$\mathcal{J}(\boldsymbol{w}) = \begin{bmatrix} \frac{\partial f(\boldsymbol{w}, \boldsymbol{x}_1)}{\partial \boldsymbol{w}} & \cdots & \frac{\partial f(\boldsymbol{w}, \boldsymbol{x}_n)}{\partial \boldsymbol{w}} \end{bmatrix}^T. \tag{12}$$

Define the $n$-dimensional residual vector and corrupted residual vector $\boldsymbol{e}$ where

$$\boldsymbol{r}_t = \boldsymbol{r}_t(\boldsymbol{w}) = \begin{bmatrix} f(\boldsymbol{x}_1, \boldsymbol{w}_t) - \bar{\boldsymbol{y}}_1^t & \cdots & f(\boldsymbol{x}_n, \boldsymbol{w}_t) - \bar{\boldsymbol{y}}_n^t \end{bmatrix}^T \quad \text{and} \quad \boldsymbol{e}_t = \bar{\boldsymbol{y}}^t - \boldsymbol{y}^*.$$

A key idea in our approach is that we argue that (1) in the absence of any corruption $\boldsymbol{r}(\boldsymbol{w})$ approximately lies on the subspace $\mathcal{S}_+$ and (2) if the labels are corrupted by a vector $\boldsymbol{e}$, then $\boldsymbol{e}$ approximately lies on the complement space.

Throughout, $\sigma_{\min}(\cdot)$ denotes the smallest singular value of a given matrix. We first introduce helpful definitions that will be used in our proofs. Given a matrix $\boldsymbol{X} \in \mathbb{R}^{n \times d}$ and a subspace $\mathcal{S} \subset \mathbb{R}^n$, we define the minimum singular value of the matrix over this subspace by $\sigma_{\min}(\boldsymbol{X}, \mathcal{S})$ which is defined as

$$\sigma_{\min}(\boldsymbol{X}, \mathcal{S}) = \sup_{\|\boldsymbol{v}\|_2 = 1, \boldsymbol{U}\boldsymbol{U}^T = \mathcal{P}_{\mathcal{S}}} \|\boldsymbol{v}^T \boldsymbol{U}^T \boldsymbol{X}\|_2.$$

Here, $\mathcal{P}_{\mathcal{S}} \in \mathbb{R}^{n \times n}$ is the projection operator to the subspace. Hence, this definition essentially projects the matrix on $\mathcal{S}$ and then takes the minimum singular value over that projected subspace.

Since augmentations are produced by using the vanilla sample $\boldsymbol{c}_i$ and the augmentation $\boldsymbol{x}$ obeys $\|\boldsymbol{x} - \boldsymbol{c}_i\|_2 \leq \epsilon_0$. So in this sense, we often call the vanilla sample and its augmentations as cluster, and call the vanilla sample as cluster center.

### C.1.2 Definitions and Assumptions

To begin with, we define $(\varepsilon, \delta)$-clusterable dataset. As aforementioned, we often call the vanilla sample and its augmentations as cluster, and call the vanilla sample as cluster center, because augmentations are produced by using the vanilla sample $\boldsymbol{c}_i$ and the augmentation $\boldsymbol{x}$ obeys $\|\boldsymbol{x} - \boldsymbol{c}_i\|_2 \leq \epsilon_0$.

**Definition 1** (($\varepsilon, \delta$)-clusterable dataset). *Suppose $\{(\boldsymbol{x}_i, \boldsymbol{y}_i^*)\}_{i=1}^n$ denote the pairs of augmentation and ground-truth label, where augmentation $\boldsymbol{x}_i$ generated from the $t$-th sample $\boldsymbol{c}_t$ obeys $\|\boldsymbol{x} - \boldsymbol{c}_t\|_2 \leq \varepsilon$ with a constant $\varepsilon$, and $\boldsymbol{y}_i^* \in \{\gamma_1, \gamma_2, \ldots, \gamma_{\bar{K}}\}$ of $\boldsymbol{x}_i$ is the label of $\boldsymbol{c}_t$. Moreover, samples and its augmentations are normalized, i.e. $\|\boldsymbol{c}_i\|_2 = \|\boldsymbol{x}_i\|_2 = 1$. Each vanilla sample $\boldsymbol{c}_i$ has $n_i$ augmentations, where $c_l \frac{n}{K} \leq n_i \leq c_u \frac{n}{K}$ with two constants $c_l$ and $c_u$. Moreover, the classes are separated such that*

$$|\gamma_r - \gamma_s| \geq \delta, \quad \|\boldsymbol{c}_r - \boldsymbol{c}_s\|_2 \geq 2\varepsilon, \quad (\forall r \neq s),$$

*where $\delta$ is the label separation.*

Our approach is based on the hypothesis that the nonlinear model has a Jacobian matrix with *bimodal spectrum where few singular values are large and remaining singular values are small*. This assumption is inspired by the fact that realistic datasets are clusterable in a proper, possibly nonlinear, representation space. Indeed, one may argue that one reason for using neural networks is to automate the learning of such a representation (essentially the input to the softmax layer). We formalize the notion of bimodal spectrum below.

**Assumption 1** (Bimodal Jacobian). *Let $\beta \geq \alpha \geq \epsilon > 0$ be scalars. Let $f : \mathbb{R}^p \to \mathbb{R}^n$ be a nonlinear mapping and consider a set $\mathcal{D} \subset \mathbb{R}^p$ containing the initial point $\boldsymbol{w}_0$ (i.e. $\boldsymbol{w}_0 \in \mathcal{D}$). Let $\mathcal{S}_+ \subset \mathbb{R}^n$ be a subspace and $\mathcal{S}_-$ be its complement. We say the mapping $f$ has a Bimodal Jacobian with respect to the complementary subpspaces $\mathcal{S}_+$ and $\mathcal{S}_-$ as long as the following two assumptions hold for all $\boldsymbol{w} \in \mathcal{D}$.*

- **Spectrum over $\mathcal{S}_+$:** *For all $\boldsymbol{v} \in \mathcal{S}_+$ with unit Euclidian norm we have*

$$\alpha \leq \left\| \mathcal{J}^T(\boldsymbol{w})\boldsymbol{v} \right\|_{\ell_2} \leq \beta.$$

- **Spectrum over $\mathcal{S}_-$:** *For all $\boldsymbol{v} \in \mathcal{S}_-$ with unit Euclidian norm we have*

$$\left\| \mathcal{J}^T(\boldsymbol{w})\boldsymbol{v} \right\|_{\ell_2} \leq \epsilon.$$

*We will refer to $\mathcal{S}_+$ as the* signal subspace *and $\mathcal{S}_-$ as the* noise subspace.

When $\epsilon << \alpha$ the Jacobian is approximately low-rank. An extreme special case of this assumption is where $\epsilon = 0$ so that the Jacobian matrix is exactly low-rank. We formalize this assumption below for later reference.

**Assumption 2** (Low-rank Jacobian). *Let $\beta \geq \alpha > 0$ be scalars. Consider a set $\mathcal{D} \subset \mathbb{R}^p$ containing the initial point $\boldsymbol{w}_0$ (i.e. $\boldsymbol{w}_0 \in \mathcal{D}$). Let $\mathcal{S}_+ \subset \mathbb{R}^n$ be a subspace and $\mathcal{S}_-$ be its complement. For all $\boldsymbol{w} \in \mathcal{D}$, $\boldsymbol{v} \in \mathcal{S}_+$ and $\boldsymbol{v}' \in \mathcal{S}_-$ with unit Euclidian norm, we have that*

$$\alpha \leq \left\| \mathcal{J}^T(\boldsymbol{w})\boldsymbol{v} \right\|_{\ell_2} \leq \beta \quad and \quad \left\| \mathcal{J}^T(\boldsymbol{w})\boldsymbol{v}' \right\|_{\ell_2} = 0.$$

In Theorem 7, we verify that the Jacobian matrix of real datasets indeed have a bimodal structure i.e. there are few large singular values and the remaining singular values are small which further motivate Assumption 2. This is inline with earlier papers which observed that Hessian matrices of deep networks have bimodal spectrum (approximately low-rank) [17] and is related to various results demonstrating that there are flat directions in the loss landscape [18].

Our dataset model in Definition 1 naturally has a low-rank Jacobian when $\epsilon_0 = 0$ and each augmentation is equal to one of the $K$ centers (vanilla samples) $\{\boldsymbol{c}_\ell\}_{\ell=1}^K$. In this case, the Jacobian will be at most rank $K$ since each row will be in the span of $\left\{ \frac{\partial f(\boldsymbol{c}_\ell, \boldsymbol{w})}{\partial \boldsymbol{w}} \right\}_{\ell=1}^K$. The subspace $\mathcal{S}_+$ is dictated by the *membership* of each cluster center (vanilla example) as follows: Let $\Lambda_\ell \subset \{1, \ldots, n\}$ be the set of coordinates $i$ such that $\boldsymbol{x}_i = \boldsymbol{c}_\ell$. Then, subspace is characterized by $\mathcal{S}_+ = \left\{ \boldsymbol{v} \in \mathbb{R}^n \middle| \boldsymbol{v}_{i_1} = \boldsymbol{v}_{i_2} \text{ for all } i_1, i_2 \in \Lambda_\ell \text{ and } 1 \leq \ell \leq K \right\}$. When $\epsilon_0 > 0$ and the augmentation points of each cluster (vanilla sample ) are not the same as the cluster we have the bimodal Jacobian structure of Assumption 1 where over $\mathcal{S}_-$ the spectral norm is small but nonzero.

**Definition 2** (Support subspace). *Let $\{\boldsymbol{x}_i\}_{i=1}^n$ be an input dataset generated according to Definition 1. Also let $\{\widetilde{\boldsymbol{x}}_i\}_{i=1}^n$ be the associated vanilla samples, that is, $\widetilde{\boldsymbol{x}}_i = \boldsymbol{c}_\ell$ iff $\boldsymbol{x}_i$ is from the $\ell$th vanilla sample. We define the support subspace $\mathcal{S}_+$ as a subspace of dimension $K$, dictated by the cluster center membership as follows. Let $\Lambda_\ell \subset \{1, \ldots, n\}$ be the set of coordinates $i$ such that $\widetilde{\boldsymbol{x}}_i = \boldsymbol{c}_\ell$. Then, $\mathcal{S}_+$ is characterized by*

$$\mathcal{S}_+ = \left\{ \boldsymbol{v} \in \mathbb{R}^n \middle| \boldsymbol{v}_{i_1} = \boldsymbol{v}_{i_2} \quad for \ all \quad i_1, i_2 \in \Lambda_\ell \quad and \ for \ all \ 1 \leq \ell \leq K \right\}.$$

Before we state our general result we need to discuss another assumption and definition.

**Assumption 3** (Smoothness). *The Jacobian mapping $\mathcal{J}(\boldsymbol{w})$ associated to a nonlinear mapping $f : \mathbb{R}^p \to \mathbb{R}^n$ is $L$-smooth if for all $\boldsymbol{w}_1, \boldsymbol{w}_2 \in \mathbb{R}^p$ we have $\|\mathcal{J}(\boldsymbol{w}_2) - \mathcal{J}(\boldsymbol{w}_1)\| \leq L \|\boldsymbol{w}_2 - \boldsymbol{w}_1\|_{\ell_2}$.*

In Theorem 7, we verify this assumption. Note that, if $\frac{\partial \mathcal{J}(\boldsymbol{w})}{\partial \boldsymbol{w}}$ is continuous, the smoothness condition holds over any compact domain (albeit for a possibly large $L$.

Additionally, to connect our results to the number of corrupted labels, we introduce the notion of subspace diffusedness defined below.

**Definition 3** (Diffusedness). *$\mathcal{S}_+$ is $\zeta$ diffused if for any vector $\boldsymbol{v} \in \mathcal{S}_+$*

$$\|\boldsymbol{v}\|_\infty \leq \sqrt{\zeta/n}\|\boldsymbol{v}\|_2,$$

*holds for some $\zeta > 0$.*

We begin by defining the average Jacobian which will be used throughout our analysis.

**Definition 4** (Average Jacobian). *We define the average Jacobian along the path connecting two points $\boldsymbol{x}, \boldsymbol{y} \in \mathbb{R}^p$ as*

$$\mathcal{J}(\boldsymbol{y}, \boldsymbol{x}) := \int_0^1 \mathcal{J}(\boldsymbol{x} + \alpha(\boldsymbol{y} - \boldsymbol{x})) d\alpha.$$

**Definition 5** (Neural Net Jacobian). *Given input samples $(\boldsymbol{x}_i)_{i=1}^n$, form the input matrix $\boldsymbol{X} = [\boldsymbol{x}_1 \ \ldots \ \boldsymbol{x}_n]^T \in \mathbb{R}^{n \times d}$. The Jacobian of our learning problem, i.e. $\boldsymbol{x} \mapsto f(\boldsymbol{W}, \boldsymbol{x}) = \boldsymbol{v}^T \phi(\boldsymbol{W} \boldsymbol{x})$ and $\mathcal{L}_t(\boldsymbol{W}) = \frac{1}{2} \sum_{i=1}^n (\boldsymbol{y}_{ti} - f(\boldsymbol{W}, \boldsymbol{x}_i))^2$, at a matrix $\boldsymbol{W}$ is denoted by $\mathcal{J}(\boldsymbol{W}, \boldsymbol{X}) \in \mathbb{R}^{n \times kd}$ and is given by*

$$\mathcal{J}(\boldsymbol{W}, \boldsymbol{X})^T = (diag(\boldsymbol{v})\phi'(\boldsymbol{W} \boldsymbol{X}^T)) * \boldsymbol{X}^T.$$

*Here $*$ denotes the Khatri-Rao product.*

### C.1.3 Auxiliary Lemmas

**Lemma 3** (Linearization of the residual). *For the general problem (10) in Appendix C.1.1, we define*

$$\boldsymbol{G}(\boldsymbol{w}_t) = \mathcal{J}(\boldsymbol{w}_{t+1}, \boldsymbol{w}_t)\mathcal{J}(\boldsymbol{w}_t)^T.$$

*where $\mathcal{J}(\boldsymbol{w}_t)$ denotes the Jacobian matrix defined in Eqn. (12), and $\mathcal{J}(\boldsymbol{w}_{t+1}, \boldsymbol{w}_t) = \int_0^1 \mathcal{J}(\boldsymbol{w}_t + \alpha(\boldsymbol{w}_{t+1} - \boldsymbol{w}_t)) d\alpha$ denotes the average Jacobian matrix defined in Definition (4). When using the gradient descent iterate $\boldsymbol{w}_{t+1} = \boldsymbol{w}_t - \eta \nabla \mathcal{L}_t(\boldsymbol{w}_t)$, then residuals*

$$\boldsymbol{r}_{t+1} = f(\boldsymbol{w}_{t+1}) - \bar{\boldsymbol{y}}^{t+1}, \quad \boldsymbol{r}_t = f(\boldsymbol{w}_t) - \bar{\boldsymbol{y}}^t$$

*obey the following equation*

$$\boldsymbol{r}_{t+1} = (\boldsymbol{I} - \eta \boldsymbol{G}(\boldsymbol{w}_t))\boldsymbol{r}_t + \bar{\boldsymbol{y}}^t - \bar{\boldsymbol{y}}^{t+1}.$$

*Proof.* Here we follow [19] to prove our result. Following Definition 4, denoting $\boldsymbol{r}_{t+1} = f(\boldsymbol{w}_{t+1}) - \bar{\boldsymbol{y}}^{t+1}$ and $\boldsymbol{r}_t = f(\boldsymbol{w}_t) - \bar{\boldsymbol{y}}^t$ , we find that

$$
\begin{aligned}
\boldsymbol{r}_{t+1} =& \boldsymbol{r}_t - f(\boldsymbol{w}_t) + f(\boldsymbol{w}_{t+1}) + \bar{\boldsymbol{y}}^t - \bar{\boldsymbol{y}}^{t+1} \\
\overset{①}{=}& \boldsymbol{r}_t + \mathcal{J}(\boldsymbol{w}_{t+1}, \boldsymbol{w}_t)(\boldsymbol{w}_{t+1} - \boldsymbol{w}_t) + \bar{\boldsymbol{y}}^t - \bar{\boldsymbol{y}}^{t+1} \\
\overset{②}{=}& \boldsymbol{r}_t - \eta \mathcal{J}(\boldsymbol{w}_{t+1}, \boldsymbol{w}_t)\mathcal{J}(\boldsymbol{w}_t)^T \boldsymbol{r}_t + \bar{\boldsymbol{y}}^t - \bar{\boldsymbol{y}}^{t+1} \\
=& (\boldsymbol{I} - \eta \boldsymbol{G}(\boldsymbol{w}_t))\boldsymbol{r}_t + \bar{\boldsymbol{y}}^t - \bar{\boldsymbol{y}}^{t+1}.
\end{aligned}
$$

where ① uses the fact that Jacobian is the derivative of $f$ and ② uses the fact that $\nabla \mathcal{L}_t(\boldsymbol{w}) = \mathcal{J}(\boldsymbol{w})^T \boldsymbol{r}_t$. $\qquad \square$

Using Assumption 3, one can show that sparse vectors have small projection on $\mathcal{S}_+$.

**Lemma 4.** *[19] Suppose Assumption 3 holds. If $\boldsymbol{r} \in \mathbb{R}^n$ is a vector with $s$ nonzero entries, we have that*

$$\|\mathcal{P}_{\mathcal{S}_+}(\boldsymbol{r})\|_\infty \leq \frac{\zeta\sqrt{s}}{n}\|\boldsymbol{r}\|_2,$$

*where $\mathcal{P}_{\mathcal{S}_+}(\boldsymbol{r})$ projects $\boldsymbol{r}$ onto the space $\mathcal{S}_+$.*

**Lemma 5.** *For the general problem (10) in Appendix C.1.1, let $\boldsymbol{r}_t = f(\boldsymbol{w}_t) - \bar{\boldsymbol{y}}^t$ and $\widehat{\boldsymbol{r}}_t = \mathcal{P}_{\mathcal{S}_+}(\boldsymbol{r}_t)$. Suppose Assumption 2 holds and $\eta \leq \frac{1}{\beta^2}$. If $\|\boldsymbol{w}_t - \boldsymbol{w}_0\|_2 + \frac{\|\widehat{\boldsymbol{r}}_t\|_2}{\alpha} \leq \frac{4(1+\psi)\|\boldsymbol{r}_0\|_2}{\alpha}$, then*

$$\boldsymbol{w}_{t+1} \in \mathcal{D} = \big\{ \boldsymbol{w} \in \mathbb{R}^p \ \big| \ \|\boldsymbol{w} - \boldsymbol{w}_0\|_2 \leq \frac{4(1+\psi)\|\boldsymbol{r}_0\|_2}{\alpha} \big\}.$$

*Proof.* Since range space of Jacobian is in $\mathcal{S}_+$ and $\eta \leq 1/\beta^2$, we can easily obtain

$$
\begin{aligned}
\|\boldsymbol{w}_{t+1} - \boldsymbol{w}_t\|_2 &= \eta \|\mathcal{J}^T(\boldsymbol{w}_t) \left(f(\boldsymbol{w}_t) - \bar{\boldsymbol{y}}^t\right)\|_2 \\
&\overset{①}{=} \eta \|\mathcal{J}^T(\boldsymbol{w}_t) \left(\mathcal{P}_{\mathcal{S}_+}(f(\boldsymbol{w}_t) - \bar{\boldsymbol{y}}^t)\right)\|_2 \\
&\overset{②}{=} \eta \|\mathcal{J}^T(\boldsymbol{w}_t)\widehat{\boldsymbol{r}}_t\|_2 \\
&\overset{③}{\leq} \eta\beta\|\widehat{\boldsymbol{r}}_t\|_2 \\
&\overset{④}{\leq} \frac{\|\widehat{\boldsymbol{r}}_t\|_2}{\beta} \\
&\overset{⑤}{\leq} \frac{\|\widehat{\boldsymbol{r}}_t\|_2}{\alpha}
\end{aligned}
$$

In the above, ① follows from the fact that row range space of Jacobian is subset of $\mathcal{S}_+$ via Assumption 2. ② follows from the definition of $\widehat{\boldsymbol{r}}_t = \mathcal{P}_{\mathcal{S}_+}(f(\boldsymbol{w}_t) - \bar{\boldsymbol{y}}^t)$. ③ follows from the upper bound on the spectral norm of the Jacobian over $\mathcal{D}$ per Assumption 2, ④ from the fact that $\eta \leq \frac{1}{\beta^2}$, ⑤ from $\alpha \leq \beta$. The latter combined with the triangular inequality and the assumption

$$
\|\boldsymbol{w}_{t+1} - \boldsymbol{w}_0\|_2 \leq \|\boldsymbol{w}_{t+1} - \boldsymbol{w}_t\|_2 + \|\boldsymbol{w}_0 - \boldsymbol{w}_t\|_2 \leq \|\boldsymbol{w}_t - \boldsymbol{w}_0\|_2 + \frac{\|\widehat{\boldsymbol{r}}_t\|_2}{\alpha} \leq \frac{4(1+\psi)\|\boldsymbol{r}_0\|_2}{\alpha},
$$

concluding the proof of $\boldsymbol{r}_{t+1} \in \mathcal{D}$. $\qquad\square$

**Lemma 6.** *[19] Let $\mathcal{P}_{\mathcal{S}_+} \in \mathbb{R}^{n \times n}$ be the projection matrix to $\mathcal{S}_+$ i.e. it is a positive semi-definite matrix whose eigenvectors over $\mathcal{S}_+$ is 1 and its complement is 0. Let $\boldsymbol{r}_t = f(\boldsymbol{w}_t) - \boldsymbol{y}_t$, $\widehat{\boldsymbol{r}}_t = \mathcal{P}_{\mathcal{S}_+}(\boldsymbol{r}_t)$, and $\boldsymbol{G}(\boldsymbol{w}_t) = \mathcal{J}(\boldsymbol{w}_{t+1}, \boldsymbol{w}_t)\mathcal{J}(\boldsymbol{w}_t)^T$. Suppose Assumptions 2 and 3 hold, the learning rate $\eta$ satisfies $\eta \leq \frac{\alpha}{L\beta\|\boldsymbol{r}_0\|_2}$, $\|\widehat{\boldsymbol{r}}_t\|_2 \leq \|\widehat{\boldsymbol{r}}_0\|_2$, then it holds*

$$
\beta^2 \mathcal{P}_{\mathcal{S}_+} \succeq \boldsymbol{G}(\boldsymbol{w}_t) \succeq \frac{1}{2}\mathcal{J}(\boldsymbol{w}_t)\mathcal{J}(\boldsymbol{w}_t)^T \succeq \frac{\alpha^2}{2}\mathcal{P}_{\mathcal{S}_+}.
$$

In the above context, we focus on introducing theoretical results for the general problem (10) in Appendix C.1.1. Now we introduce lemmas and theories for our network learning problem, i.e. $\boldsymbol{x} \mapsto f(\boldsymbol{W}, \boldsymbol{x}) = \boldsymbol{v}^T\phi(\boldsymbol{W}\boldsymbol{x})$ and $\mathcal{L}_t(\boldsymbol{W}) = \frac{1}{2}\sum_{i=1}^n (\bar{\boldsymbol{y}}_i^t - f(\boldsymbol{W}, \boldsymbol{x}_i))^2$ used in our manuscript. Specifically, we introduce some theoretical results in [20] and characterizes three key properties of the neural network Jacobian. These are smoothness, spectral norm, and minimum singular value at initialization which correspond to Lemmas 6.6, 6.7, and 6.8 in that paper.

**Theorem 3** (Jacobian Properties at Cluster Center). *[20] Suppose $\boldsymbol{X} = [\boldsymbol{x}_1 \ \ldots \ \boldsymbol{x}_n]^T \in \mathbb{R}^{n \times d}$ be an input dataset satisfying $\lambda(\boldsymbol{X}) > 0$, where $\lambda(\boldsymbol{X})$ denotes the smallest eigenvalue of matrix $\boldsymbol{X}$. Suppose $|\phi'|, |\phi''| \leq \Gamma$ where $\phi'$ and $\phi''$ respectively denotes the first and second order derivatives. The Jacobian mapping with respect to the input-to-hidden weights obey the following properties. Let $\mathcal{J}(\boldsymbol{W}, \boldsymbol{X})$ denote the neural net Jacobian defined in Definition 5.*

*(1) Smoothness is bounded by*

$$
\left\|\mathcal{J}(\widetilde{\boldsymbol{W}}, \boldsymbol{X}) - \mathcal{J}(\boldsymbol{W}, \boldsymbol{X})\right\| \leq \frac{\Gamma}{\sqrt{k}}\|\boldsymbol{X}\|\left\|\widetilde{\boldsymbol{W}} - \boldsymbol{W}\right\|_F \quad for \ all \quad \widetilde{\boldsymbol{W}}, \boldsymbol{W} \in \mathbb{R}^{k \times d}.
$$

*(2) Top singular value is bounded by*

$$
\|\mathcal{J}(\boldsymbol{W}, \boldsymbol{X})\| \leq \Gamma \|\boldsymbol{X}\|.
$$

*(3) Let $C > 0$ be an absolute constant. As long as*

$$
k \geq \frac{C\Gamma^2 \log n \|\boldsymbol{X}\|^2}{\lambda(\boldsymbol{X})}
$$

*At random Gaussian initialization $\boldsymbol{W}_0 \sim \mathcal{N}(0, 1)^{k \times d}$, with probability at least $1 - 1/K^{100}$, we have*

$$
\sigma_{\min}(\mathcal{J}(\boldsymbol{W}_0, \boldsymbol{X})) \geq \sqrt{\lambda(\boldsymbol{X})/2}.
$$

The following theorem states the properties of the Jacobian at a $(\epsilon_0, \delta)$ clusterable dataset defined in Definition 1. That is, $(\boldsymbol{x}_i)_{i=1}^n$ are generated from $(\boldsymbol{c}_i)_{i=1}^K$, and their augmentation distance is at most $\epsilon_0$ and label separation is at least $\delta$.

**Theorem 4** (Jacobian Properties at Cluster Center). *[19] Let input samples $(\boldsymbol{x}_i)_{i=1}^n$ be generated according to $(\epsilon_0, \delta)$ clusterable dataset model of Definition 1. Define $\boldsymbol{X} = [\boldsymbol{x}_1 \ \ldots \ \boldsymbol{x}_n]^T$ and $\boldsymbol{C} = [\boldsymbol{c}_1 \ \ldots \ \boldsymbol{c}_k]^T$. Let $\mathcal{S}_+$ be the support space and $(\widetilde{\boldsymbol{x}}_i)_{i=1}^n$ be the associated clean dataset as described by Definition 2. Set $\widetilde{\boldsymbol{X}} = [\widetilde{\boldsymbol{x}}_1 \ \ldots \ \widetilde{\boldsymbol{x}}_n]^T$. Assume $|\phi'|, |\phi''| \leq \Gamma$ and $\lambda(\boldsymbol{C}) > 0$. Let $\mathcal{J}(\boldsymbol{W}, \boldsymbol{X})$ denote the neural net Jacobian defined in Definition 5. The Jacobian mapping at $\widetilde{\boldsymbol{X}}$ with respect to the input-to-hidden weights obey the following properties.*

*(1) Smoothness is bounded by*

$$\left\| J(\widetilde{\boldsymbol{W}}, \widetilde{\boldsymbol{X}}) - \mathcal{J}(\boldsymbol{W}, \widetilde{\boldsymbol{X}}) \right\| \leq \Gamma \sqrt{\frac{c_{up}n}{kK}} \left\| \boldsymbol{C} \right\| \left\| \widetilde{\boldsymbol{W}} - \boldsymbol{W} \right\|_F \quad \text{for all} \quad \widetilde{\boldsymbol{W}}, \boldsymbol{W} \in \mathbb{R}^{k \times d}.$$

*(2) Top singular value is bounded by*

$$\left\| \mathcal{J}(\boldsymbol{W}, \widetilde{\boldsymbol{X}}) \right\| \leq \sqrt{\frac{c_{up}n}{K}} \Gamma \left\| \boldsymbol{C} \right\|.$$

*(3) As long as*

$$k \geq \frac{C\Gamma^2 \log K \left\| \boldsymbol{C} \right\|^2}{\lambda(\boldsymbol{C})}$$

*At random Gaussian initialization $\boldsymbol{W}_0 \sim \mathcal{N}(0, 1)^{k \times d}$, with probability at least $1 - 1/K^{100}$, we have*

$$\sigma_{\min}\left( \mathcal{J}(\boldsymbol{W}_0, \widetilde{\boldsymbol{X}}), \mathcal{S}_+ \right) \geq \sqrt{\frac{c_{low}n\lambda(\boldsymbol{C})}{2K}}$$

*(4) The range space obeys $\text{range}(\mathcal{J}(\boldsymbol{W}_0, \widetilde{\boldsymbol{X}})) \subset \mathcal{S}_+$ where $\mathcal{S}_+$ is given by Definition 2.*

**Lemma 7** (Upper bound on initial misfit). *[19] Consider a one-hidden layer neural network model of the form $\boldsymbol{x} \mapsto \boldsymbol{v}^T \phi(\boldsymbol{W}\boldsymbol{x})$ where the activation $\phi$ has bounded derivatives obeying $|\phi(0)|, |\phi'(z)| \leq \Gamma$. Suppose entries of $\boldsymbol{v} \in \mathbb{R}^k$ are half $1/\sqrt{k}$ and half $-1/\sqrt{k}$ so that $\|\boldsymbol{v}\|_2 = 1$. Also assume we have $n$ data points $\boldsymbol{x}_1, \boldsymbol{x}_2, \ldots, \boldsymbol{x}_n \in \mathbb{R}^d$ with unit euclidean norm ($\|\boldsymbol{x}_i\|_2 = 1$) aggregated as rows of a matrix $\boldsymbol{X} \in \mathbb{R}^{n \times d}$ and the corresponding labels given by $\boldsymbol{y} \in \mathbb{R}^n$ generated accoring to $(\rho, \varepsilon = 0, \delta)$ noisy dataset (Definition 1). Then for $\boldsymbol{W}_0 \in \mathbb{R}^{k \times d}$ with i.i.d. $\mathcal{N}(0, 1)$ entries*

$$\|\boldsymbol{v}^T \phi(\boldsymbol{W}_0 \boldsymbol{X}^T) - \boldsymbol{y}\|_2 \leq \mathcal{O}\left( \Gamma\sqrt{n \log K} \right),$$

*holds with probability at least $1 - K^{-100}$.*

Then we introduce a lemma regarding the projection of label noise on the vanilla sample (cluster) induced subspace. Since augmentations are produced by using the vanilla sample $\boldsymbol{c}_i$ and the augmentation $\boldsymbol{x}$ obeys $\|\boldsymbol{x} - \boldsymbol{c}_i\|_2 \leq \epsilon_0$. So in this sense, we sometimes call the vanilla sample and its augmentations as cluster, and call the vanilla sample as cluster center.

**Lemma 8.** *[19] Let $\{(\boldsymbol{x}_i, \boldsymbol{y}_i)\}_{i=1}^n$ be an $(\rho, \varepsilon = 0, \delta)$ clusterable noisy dataset as described in Definition 1. Let $\{\boldsymbol{y}_i^*\}_{i=1}^n$ be the corresponding ground truth labels. Let $\mathcal{J}(\boldsymbol{W}, \boldsymbol{C})$ be the Jacobian at the cluster center matrix which is rank $K$ and $\mathcal{S}_+$ be its column space. Then, the difference between noiseless and noisy labels satisfy the bound*

$$\|\mathcal{P}_{\mathcal{S}_+}(\boldsymbol{y} - \boldsymbol{y}^*)\|_\infty \leq 2\rho.$$

**Theorem 5.** *[19] Assume $|\phi'|, |\phi''| \leq \Gamma$ and $k \gtrsim d$. Suppose $\boldsymbol{W}_0 \sim \mathcal{N}(0, 1)$. Let $\boldsymbol{c}_1, \ldots, \boldsymbol{c}_K$ be cluster centers. Then, with probability at least $1 - 2e^{-(k+d)} - Ke^{-100d}$ over $\boldsymbol{W}_0$, any matrix $\boldsymbol{W}$ satisfying $\|\boldsymbol{W} - \boldsymbol{W}_0\|_F \lesssim \sqrt{k}$ satisfies the following. For all $1 \leq i \leq K$,*

$$\sup_{\|\boldsymbol{x} - \boldsymbol{c}_i\|_2, \|\widetilde{\boldsymbol{x}} - \boldsymbol{c}_i\|_2 \leq \varepsilon} |f(\boldsymbol{W}, \boldsymbol{x}) - f(\boldsymbol{W}, \widetilde{\boldsymbol{x}})| \leq C\Gamma\varepsilon(\|\boldsymbol{W} - \boldsymbol{W}_0\| + \sqrt{d}).$$

**Lemma 9** (Perturbed Jacobian Distance). *[19] Let $\boldsymbol{X} = [\boldsymbol{x}_1 \ \ldots \ \boldsymbol{x}_n]^T$ be the input matrix obtained from Definition 1. Let $\widetilde{\boldsymbol{X}}$ be the noiseless inputs where $\widetilde{\boldsymbol{x}}_i$ is the cluster center corresponding to $\boldsymbol{x}_i$. Let $\mathcal{J}(\boldsymbol{W}, \boldsymbol{X})$ denote the neural net Jacobian defined in Definition 5 and define $\mathcal{J}(\boldsymbol{W}_1, \boldsymbol{W}_2, \boldsymbol{X}) = \int_0^1 \mathcal{J}(\alpha \boldsymbol{W}_1 + (1 - \alpha)\boldsymbol{W}_2, \boldsymbol{X})d\alpha$. Given weight matrices $\boldsymbol{W}_1, \boldsymbol{W}_2, \widetilde{\boldsymbol{W}}_1, \widetilde{\boldsymbol{W}}_2$, we have that*

$$\|\mathcal{J}(\boldsymbol{W}, \boldsymbol{X}) - \mathcal{J}(\widetilde{\boldsymbol{W}}, \widetilde{\boldsymbol{X}})\| \leq \Gamma\sqrt{n}\left(\frac{\|\widetilde{\boldsymbol{W}} - \boldsymbol{W}\|_F}{\sqrt{k}} + \varepsilon\right).$$

*and*

$$\|\mathcal{J}(\boldsymbol{W}_1, \boldsymbol{W}_2, \boldsymbol{X}) - \mathcal{J}(\widetilde{\boldsymbol{W}}_1, \widetilde{\boldsymbol{W}}_2, \widetilde{\boldsymbol{X}})\| \leq \Gamma\sqrt{n}\left(\frac{\|\widetilde{\boldsymbol{W}}_1 - \boldsymbol{W}_1\|_F + \|\widetilde{\boldsymbol{W}}_2 - \boldsymbol{W}_2\|_F}{2\sqrt{k}} + \varepsilon\right).$$

## C.2 Proof of Theorem 2

The subsection has four parts. In the first part, we introduce the proof roadmap in Appendix C.2.1. Then in the second part, we present several auxiliary theories in Appendix C.2.2. Next, we prove our Theorem 2 in Appendix C.2.3. Finally, we present all proof details of auxiliary theories in Appendix C.2.2.

### C.2.1 Proof roadmap

Before proving Theorem 2, we first briefly introduce our main idea. In the **first step**, we analyze the general model introduced in Appendix C.1.1. For the solution $\boldsymbol{w}_t$ at the $t$-th iteration, Theorem 6 proves that (1) the distance of $\|\boldsymbol{w}_t - \boldsymbol{w}_0\|_2$ can be upper bounded; (2) both residual $\|\mathcal{P}_{\mathcal{S}_+}(f(\boldsymbol{w}_t) - \bar{\boldsymbol{y}}^t)\|_2$ and $\|f(\boldsymbol{w}_t) - \boldsymbol{y}^*\|_\infty$ can be upper bound. Result (1) means that the gradient descent algorithm gives solutions in a ball around the initialization $\boldsymbol{w}_0$, and helps us verify our assumptions, e.g. Assumptions 3 and 2 and upper bound some variables in our analysis. Results (2) directly bound the label estimation error which plays key role in subsequent analysis.

In the **second step**, we prove Theorem 7 for the perfectly clustered data ($\epsilon_0 = 0$) by using Theorem 6. We consider $\epsilon_0 \to 0$ which means that the input data set is perfectly clean. In this setting, let $\widetilde{\boldsymbol{X}} = [\widetilde{\boldsymbol{x}}_1, \cdots, \widetilde{\boldsymbol{x}}_n]$ be the clean input sample matrix obtained by mapping $\boldsymbol{x}_i$ to its associated cluster center, i.e. $\widetilde{\boldsymbol{x}}_i = \boldsymbol{c}_\ell$ if $\boldsymbol{x}_i$ belongs to the $\ell$-th cluster. In this way, we update network parameter $\widetilde{\boldsymbol{W}}_t$ as follows:

$$\widetilde{\boldsymbol{W}}_{t+1} = \widetilde{\boldsymbol{W}}_t - \nabla\widetilde{\mathcal{L}}_t(\widetilde{\boldsymbol{W}}_t) \quad \text{where} \quad \widetilde{\mathcal{L}}_t(\widetilde{\boldsymbol{W}}) = \frac{1}{2}\sum_{i=1}^{n}(\boldsymbol{y}_{ti} - f(\widetilde{\boldsymbol{W}}, \widetilde{\boldsymbol{x}}_i))^2$$

Theorem 7 shows that for neural networks, our method still can upper bound the distance $\|\widetilde{\boldsymbol{W}}_t - \widetilde{\boldsymbol{W}}_0\|_F$ and the residuals $\|f(\widetilde{\boldsymbol{W}}_t) - \widetilde{\boldsymbol{y}}\|_\infty$ if the network, learning rate, the weight $\alpha_i$ for refining label satisfy certain conditions.

In the **third step**, we consider the realistic setting, where we update the parameters on the corrupted data $\boldsymbol{X} = [\boldsymbol{x}_1, \cdots, \boldsymbol{x}_n]$ as follows:

$$\boldsymbol{W}_{t+1} = \boldsymbol{W}_t - \eta\nabla\mathcal{L}_t(\boldsymbol{W}_t) \quad \text{where} \quad \mathcal{L}_t(\boldsymbol{W}) = \frac{1}{2}\sum_{i=1}^{n}(\boldsymbol{y}_{ti} - f(\boldsymbol{W}, \boldsymbol{x}_i))^2. \tag{13}$$

Then to upper bound $\|f(\boldsymbol{W}_t) - \widetilde{\boldsymbol{y}}\|_\infty$ which measures the error between the predicted label $f(\boldsymbol{W}_t)$ and the ground truth label $\widetilde{\boldsymbol{y}}$, we upper bound $\|f(\boldsymbol{W}_t, \boldsymbol{X}) - f(\widetilde{\boldsymbol{W}}_t, \widetilde{\boldsymbol{X}})\|_2$ and $\|\boldsymbol{W}_t - \widetilde{\boldsymbol{W}}_t\|_F$. These results are formally stated in Theorem 8.

In the **fourth step**, we combine the above results together. Specifically, Theorem 7 upper bounds the residuals $\|f(\widetilde{\boldsymbol{W}}_t, \widetilde{\boldsymbol{X}}) - \widetilde{\boldsymbol{y}}\|_\infty$ and Theorem 8 upper bounds $\|f(\boldsymbol{W}_t, \boldsymbol{X}) - f(\widetilde{\boldsymbol{W}}_t, \widetilde{\boldsymbol{X}})\|_2$. So combining these two results and other results in Theorem 7 & 8, we can upper bound $\|f(\boldsymbol{W}_t, \boldsymbol{X}) - \widetilde{\boldsymbol{y}}\|_\infty$ which is our desired results. At the same time, by using similar method, we can also bound the label estimation error by our self-labeling refinery, since $\|\bar{\boldsymbol{y}}^t - \boldsymbol{y}^*\|_2 = \|(1 - \alpha_t)\boldsymbol{y} + \alpha_t f(\boldsymbol{w}) - \boldsymbol{y}^*\|_2 \leq (1 - \alpha_t)\|\boldsymbol{y} - \boldsymbol{y}^*\|_2 + \alpha_t\|f(\boldsymbol{w}) - \boldsymbol{y}^*\|_2$. The term $\|\boldsymbol{y} - \boldsymbol{y}^*\|_2$ denotes the initial label error and can be bounded by a factor related to $\rho$, while the second term is well upper bounded by the above results.

It should be note that our proof framework follows the recent works [21, 19] which shows that gradient descent is robust to label corruptions. The main difference is that this work uses the label estimation $\bar{\boldsymbol{y}}^t = \alpha_t \boldsymbol{y} + (1 - \alpha_t) f(\boldsymbol{w})$ and minimizes the squared loss, while both works [21, 19] use the corrupted label $\boldsymbol{y}$ and then minimize the squared loss. By comparison, our method is much more complicated and gives different proofs.

### C.2.2 Auxiliary Theories

The following theorem is to analyze the general model introduced in Appendix C.1.1. It guarantees that the estimated label by our method is close to the ground truth label when the Jacobian mapping is exactly low-rank. By using this results, one can obtain Theorem 7 for the perfectly clustered data ($\epsilon_0 = 0$) which will be stated later.

**Theorem 6** (Gradient descent with label corruption). *Consider a nonlinear least squares problem of the form $\mathcal{L}_t(\boldsymbol{w}) = \frac{1}{2} \|f(\boldsymbol{w}) - \bar{\boldsymbol{y}}^t)\|_{\ell_2}^2$ with the nonlinear mapping $f : \mathbb{R}^p \to \mathbb{R}^n$ obeying assumptions 2 and 3 over a unit Euclidian ball of radius $\frac{4(1+\psi_1)\|f(\boldsymbol{w}_0) - \boldsymbol{y}\|_2}{\alpha}$ around an initial point $\boldsymbol{w}_0$ and $\boldsymbol{y} = [y_1 \ \ldots \ y_n] \in \mathbb{R}^n$ denoting the corrupted labels. We also assume $\alpha_t \geq 1 - \frac{\alpha^2}{4\beta^2}$ and $2\sqrt{n} \lim_{t \to +\infty} \sum_{t=0}^{t} |\alpha_t - \alpha_{t+1}| \leq \psi_1 \|f(\boldsymbol{w}_0) - \bar{\boldsymbol{y}}^0\|_2$. Also let $\boldsymbol{y}^* = [y_1^* \ \ldots \ y_n^*] \in \mathbb{R}^n$ denote the ground truth labels and $\boldsymbol{e} = \boldsymbol{y} - \boldsymbol{y}^*$ the corruption. Furthermore, suppose the initial residual $f(\boldsymbol{w}_0) - \widetilde{\boldsymbol{y}}$ with respect to the uncorrupted labels obey $f(\boldsymbol{w}_0) - \boldsymbol{y}^* \in \mathcal{S}_+$. Then, running gradient descent updates of the from (11) with a learning rate $\eta \leq \frac{1}{2\beta^2} \min\left(1, \frac{\alpha\beta}{L\|f(\boldsymbol{w}_0) - \bar{\boldsymbol{y}}^0\|_2}\right)$, all iterates obey*

$$\|\boldsymbol{w}_t - \boldsymbol{w}_0\|_2 \leq \frac{4\|\boldsymbol{r}_0\|_2}{\alpha} + 2\sqrt{n} \lim_{t \to +\infty} \sum_{t=0}^{t} |\alpha_t - \alpha_{t+1}| \leq \frac{4(1+\psi)\|f(\boldsymbol{w}_0) - \bar{\boldsymbol{y}}^0\|_2}{\alpha}.$$

*and*

$$\|\widehat{\boldsymbol{r}}_t\|_2^2 \leq \left(1 - \frac{\eta\alpha^2}{4}\right)^t \|\widehat{\boldsymbol{r}}_0\|_2^2 + 2\sqrt{n} \sum_{i=0}^{t-1} \left(1 - \frac{\eta\alpha^2}{4}\right)^{t-i} |\alpha_i - \alpha_{i+1}|,$$

*where $\boldsymbol{r}_t = f(\boldsymbol{w}_t) - \bar{\boldsymbol{y}}^t$ and let $\boldsymbol{r}_0 = f(\boldsymbol{w}_0) - \bar{\boldsymbol{y}}^0$ be the initial residual, and $\widehat{\boldsymbol{r}}_t = \mathcal{P}_{\mathcal{S}_+}(\boldsymbol{r}_t)$. Furthermore, assume $\nu > 0$ is a precision level obeying $\nu \geq \|\mathcal{P}_{\mathcal{S}_+}(\boldsymbol{e})\|_\infty$. Then, after $t \geq \frac{5}{\eta\alpha^2} \log\left(\frac{\|f(\boldsymbol{w}_0) - \bar{\boldsymbol{y}}^0\|_2}{(1-\alpha_{max})\nu}\right)$ iterations where $\alpha_{max} = \max_t \alpha_t$, $\boldsymbol{w}_t$ achieves the following error bound with respect to the true labels*

$$\|f(\boldsymbol{w}_t) - \boldsymbol{y}^*\|_\infty \leq 2\nu + \frac{2\sqrt{n}}{1 - \alpha_t} \sum_{i=0}^{t-1} \left(1 - \frac{\eta\alpha^2}{4}\right)^{t-i} |\alpha_i - \alpha_{i+1}|.$$

*Furthermore, if $\boldsymbol{e}$ has at most $s$ nonzeros and $\mathcal{S}_+$ is $\zeta$ diffused per Definition 3, then using $\nu = \|\mathcal{P}_{\mathcal{S}_+}(\boldsymbol{e})\|_\infty$*

$$\|f(\boldsymbol{w}_t) - \boldsymbol{y}^*\|_\infty \leq 2\|\mathcal{P}_{\mathcal{S}_+}(\boldsymbol{e})\|_\infty + \frac{2\sqrt{n}}{1 - \alpha_t} \sum_{i=0}^{t-1} \left(1 - \frac{\eta\alpha^2}{4}\right)^{t-i} |\alpha_i - \alpha_{i+1}|$$

$$\leq \frac{\zeta\sqrt{s}}{n} \|\boldsymbol{e}\|_2 + \frac{2\sqrt{n}}{1 - \alpha_t} \sum_{t=0}^{t-1} \left(1 - \frac{\eta\alpha^2}{4}\right)^{t-i} |\alpha_i - \alpha_{i+1}|,$$

*where $\mathcal{P}_{\mathcal{S}_+}(\boldsymbol{e})$ denotes projection of $\boldsymbol{e}$ on $\mathcal{S}_+$.*

See its proof in Appendix C.2.5. This result shows that when the Jacobian of the nonlinear mapping is low-rank, our method enjoys two good properties.

For the solution $\boldsymbol{w}_t$ at the $t$-th iteration, (1) the distance of $\|\boldsymbol{w}_t - \boldsymbol{w}_0\|_2$ can be upper bounded; (2) both residual $\|\mathcal{P}_{\mathcal{S}_+}(f(\boldsymbol{w}_t) - \boldsymbol{y}_t)\|_2$ and $\|f(\boldsymbol{w}_t) - \widetilde{\boldsymbol{y}}\|_\infty$ can be upper bound. Result (1) means that the gradient descent algorithm gives solutions in a ball around the initialization $\boldsymbol{w}_0$, and helps us verify our assumptions, e.g. Assumptions 3 and 2 and upper bound some variables in our analysis. Results (2) directly bound the label estimation error which plays key role in subsequent analysis. This theorem is the key result that allows us to prove Theorem 7 when the data points are perfectly

clustered ($\epsilon_0 = 0$). Furthermore, this theorem when combined with a perturbation analysis allows us to deal with data that is not perfectly clustered ($\epsilon_0 > 0$) and to conclude the recovery ability of our method (Theorem 2).

When $\epsilon_0 \to 0$ which means that the input data set is perfectly clustered, our method can be expected to exactly recover the ground truth label by using neural networks.

**Theorem 7** (Training with perfectly clustered data). *Consider the setting and assumptions of Theorem 6 with $\epsilon_0 = 0$. Starting from an initial weight matrix $\boldsymbol{w}_0$ selected at random with i.i.d. $\mathcal{N}(0, 1)$ entries we run gradient descent updates of the form $\boldsymbol{W}_{t+1} = \boldsymbol{W}_t - \eta \nabla \mathcal{L}_t(\boldsymbol{W}_t)$ on the least-squares loss in the manuscript with step size $\eta \leq \frac{K}{2c_{up} n \Gamma^2 \|\boldsymbol{C}\|^2}$. Furthermore, assume the number of hidden nodes obey*

$$k \geq C(1 + \psi_1)^2 \Gamma^4 \frac{K \log(K) \|\boldsymbol{C}\|^2}{\lambda(\boldsymbol{C})^2},$$

*with $\lambda(\boldsymbol{C})$ is the minimum eigenvalue of $\Sigma(\boldsymbol{C})$ in Assumption 2. Then, with probability at least $1 - 2/K^{100}$ over randomly initialized $\boldsymbol{W}_0 \overset{i.i.d.}{\sim} \mathcal{N}(0, 1)$, the iterates $\boldsymbol{W}_t$ obey the following properties.*

*(1) The distance to initial point $\mathcal{W}_0$ is upper bounded by*

$$\|\boldsymbol{W}_t - \boldsymbol{W}_0\|_F \leq c\Gamma \sqrt{\frac{K \log K}{\lambda(\boldsymbol{C})}}.$$

*(2) After $t \geq t_0 := \frac{cK}{\eta n \lambda(\boldsymbol{C})} \log\left(\frac{\Gamma \sqrt{n \log K}}{(1 - \alpha_{max})\rho}\right)$ iterations where $\alpha_{max} = \max_{0 \leq t \leq t_0} \alpha_t$, the entrywise predictions of the learned network with respect to the ground truth labels $\{\boldsymbol{y}_i^*\}_{i=1}^n$ satisfy*

$$|f(\boldsymbol{W}_t, \boldsymbol{x}_i) - \boldsymbol{y}_i^*| \leq 4\rho,$$

*for all $1 \leq i \leq n$. Furthermore, if the noise level $\rho$ obeys $\rho \leq \delta/8$ the network predicts the correct label for all samples i.e.*

$$\arg \min_{i:1 \leq i \leq \bar{K}} |f(\boldsymbol{W}_t, \boldsymbol{x}_i) - \gamma_i| = \boldsymbol{y}_i^* \quad for \quad i = 1, 2, \ldots, n. \tag{14}$$

See its proof in Appendix C.2.6. This result shows that in the limit $\epsilon_0 \to 0$ where the data points are perfectly clustered, if the width of network and the iterations satisfy $k \geq C(1 + \psi_1)^2 \Gamma^4 \frac{K \log(K) \|\boldsymbol{C}\|^2}{\lambda(\boldsymbol{C})^2}$ and $t \geq t_0 := \frac{cK}{\eta n \lambda(\boldsymbol{C})} \log\left(\frac{\Gamma \sqrt{n \log K}}{\tilde{\alpha} \rho}\right)$, then our method can exactly recover the ground truth label. This result can be interpreted as ensuring that the network has enough capacity to fit the cluster centers $\{c_\ell\}_{\ell=1}^K$ and the associated true labels.

Then we consider the perturbed data $\boldsymbol{X} = [\boldsymbol{x}_1, \cdots, \boldsymbol{x}_n]$ instead of the perfectly clustered data $\widetilde{\boldsymbol{X}} = [\widetilde{\boldsymbol{x}}_1, \cdots, \widetilde{\boldsymbol{x}}_n]$ obtained by mapping $\boldsymbol{x}_i$ to its associated cluster center, i.e. $\widetilde{\boldsymbol{x}}_i = \boldsymbol{c}_\ell$ if $\boldsymbol{x}_i$ belongs to the $\ell$-th cluster. In Theorem 8, we upper bound the parameter distance and output distance under the two kinds of data $\boldsymbol{X}$ and $\widetilde{\boldsymbol{X}}$.

**Theorem 8** (Robustness of gradient path to perturbation). *Generate samples $(\boldsymbol{x}_i, \boldsymbol{y}_i)_{i=1}^n$ according to $(\rho, \varepsilon, \delta)$ corrupted dataset and form the concatenated input/labels $\boldsymbol{X} \in \mathbb{R}^{d \times n}, \boldsymbol{y} \in \mathbb{R}^n$. Let $\widetilde{\boldsymbol{X}}$ be the clean input sample matrix obtained by mapping $\boldsymbol{x}_i$ to its associated cluster center. Set learning rate $\eta \leq \frac{K}{2c_{up} n \Gamma^2 \|\boldsymbol{C}\|^2}$ and maximum iterations $t_0$ satisfying*

$$\eta t_0 = C_1 \frac{K}{n \lambda(\boldsymbol{C})} \log(\frac{\Gamma \sqrt{n \log K}}{\rho}).$$

*where $C_1 \geq 1$ is a constant of our choice. Suppose input noise level $\varepsilon$ and number of hidden nodes obey*

$$\varepsilon \leq \mathcal{O}\left(\frac{\lambda(\boldsymbol{C})}{\Gamma^2 K \log(\frac{\Gamma \sqrt{n \log K}}{\rho})}\right) \quad and \quad k \geq \mathcal{O}\left(\Gamma^{10} \frac{K^2 \|\boldsymbol{C}\|^4}{\alpha_{max}^2 \lambda(\boldsymbol{C})^4} \log(\frac{\Gamma \sqrt{n \log K}}{\rho})^6\right).$$

*where $\alpha_{max} = \max_{1 \leq t \leq t_0} \alpha_t$. Assume $2\sqrt{n}\sum_{i=0}^{t-1}\left(1 - \frac{\eta\alpha^2}{4}\right)^{t-i}|\alpha_i - \alpha_{i+1}| \leq \psi_2\|\boldsymbol{r}_0\|_2^2$ and $2\sqrt{n}\sum_{i=0}^{t-1}|\alpha_i - \alpha_{i+1}| \leq \psi_1\|\boldsymbol{r}_0\|_2$. Set $\boldsymbol{W}_0 \sim \mathcal{N}(0,1)$. Starting from $\boldsymbol{W}_0 = \widetilde{\boldsymbol{W}}_0$ consider the gradient descent iterations over the losses*

$$\boldsymbol{W}_{t+1} = \boldsymbol{W}_t - \eta\nabla\mathcal{L}_t(\boldsymbol{W}_t) \quad where \quad \mathcal{L}_t(\boldsymbol{W}) = \frac{1}{2}\sum_{i=1}^n (\boldsymbol{y}_{ti} - f(\boldsymbol{W}, \boldsymbol{x}_i))^2 \tag{15}$$

$$\widetilde{\boldsymbol{W}}_{t+1} = \widetilde{\boldsymbol{W}}_t - \nabla\widetilde{\mathcal{L}}_t(\widetilde{\boldsymbol{W}}_t) \quad where \quad \widetilde{\mathcal{L}}_t(\widetilde{\boldsymbol{W}}) = \frac{1}{2}\sum_{i=1}^n (\boldsymbol{y}_{ti} - f(\widetilde{\boldsymbol{W}}, \widetilde{\boldsymbol{x}}_i))^2 \tag{16}$$

*Then, for all gradient descent iterations satisfying $t \leq t_0$, we have that*

$$\|f(\boldsymbol{W}_t, \boldsymbol{X}) - f(\widetilde{\boldsymbol{W}}_t, \widetilde{\boldsymbol{X}})\|_2 \leq c_0\psi' t\eta\varepsilon\Gamma^3 n^{3/2}\sqrt{\log K},$$

*and*

$$\|\boldsymbol{W}_t - \widetilde{\boldsymbol{W}}_t\|_F \leq \mathcal{O}\left(t\psi'\eta\varepsilon\frac{\Gamma^4 Kn}{\lambda(\boldsymbol{C})}\log\left(\frac{\Gamma\sqrt{n\log K}}{\rho}\right)^2\right).$$

*where $\psi' = 1 + \frac{\psi_1}{2} + \sqrt{\psi_2}$.*

See its proof in Appendix C.2.7. Theorem 2 is obtained by combining the above results together.

### C.2.3   Proof of Theorem 2

*Proof of Theorem 2.* Here we prove our results by three steps. In these steps, each step proves one of the three results in our theory. To begin with, we consider two parameter update settings with initialization as $\boldsymbol{W}_0$:

$$\widetilde{\boldsymbol{W}}_{t+1} = \widetilde{\boldsymbol{W}}_t - \nabla\widetilde{\mathcal{L}}_t(\widetilde{\boldsymbol{W}}_t) \quad \text{where} \quad \widetilde{\mathcal{L}}_t(\widetilde{\boldsymbol{W}}) = \frac{1}{2}\sum_{i=1}^n (\widetilde{\boldsymbol{y}}_i^t - f(\widetilde{\boldsymbol{W}}, \widetilde{\boldsymbol{x}}_i))^2,$$

$$\boldsymbol{W}_{t+1} = \boldsymbol{W}_t - \eta\nabla\mathcal{L}_t(\boldsymbol{W}_t) \quad \text{where} \quad \mathcal{L}_t(\boldsymbol{W}) = \frac{1}{2}\sum_{i=1}^n (\bar{\boldsymbol{y}}_i^t - f(\boldsymbol{W}, \boldsymbol{x}_i))^2,$$

where $\widetilde{\boldsymbol{y}}_i^t = (1-\alpha_t)\boldsymbol{y} + \alpha_t f(\widetilde{\boldsymbol{W}}_t, \widetilde{\boldsymbol{x}}_i)$, $\bar{\boldsymbol{y}}_i^t = (1-\alpha_t)\boldsymbol{y} + \alpha_t f(\boldsymbol{W}_t, \boldsymbol{x}_i)$, $\widetilde{\boldsymbol{X}} = [\widetilde{\boldsymbol{x}}_1, \cdots, \widetilde{\boldsymbol{x}}_n]$ denotes the clean input sample matrix obtained by mapping $\boldsymbol{x}_i$ to its associated cluster center, i.e. $\widetilde{\boldsymbol{x}}_i = \boldsymbol{c}_\ell$ if $\boldsymbol{x}_i$ belongs to the $\ell$-th cluster, and $\boldsymbol{X} = [\boldsymbol{x}_1, \cdots, \boldsymbol{x}_n]$ denotes corrupted data matrix. Denote the prediction residual vectors of the noiseless and original problems with respect true ground truth labels $\boldsymbol{y}^*$ by $\widetilde{\boldsymbol{r}}_t = f(\widetilde{\boldsymbol{W}}_t, \widetilde{\boldsymbol{X}}) - \boldsymbol{y}^*$ and $\boldsymbol{r}_t = f(\boldsymbol{W}_t, \boldsymbol{X}) - \boldsymbol{y}^*$ respectively.

Theorem 7 shows that if number of iterations $t$ and network width receptively satisfy $t \geq t_0 := \frac{cK}{\eta n\lambda(\boldsymbol{C})}\log\left(\frac{\Gamma\sqrt{n\log K}}{\widetilde{\alpha}\rho}\right)$ and $k \geq C(1+\psi_1)^2\Gamma^4\frac{K\log(K)\|\boldsymbol{C}\|^2}{\lambda(\boldsymbol{C})^2}$, then it holds

$$\|\widetilde{\boldsymbol{r}}_t\|_\infty = \|f(\widetilde{\boldsymbol{W}}_t, \widetilde{\boldsymbol{X}}) - \boldsymbol{y}^*\|_\infty \leq 4\rho \quad \text{and} \quad \|\widetilde{\boldsymbol{W}}_t - \boldsymbol{W}_0\|_F \leq c\Gamma\sqrt{\frac{K\log K}{\lambda(\boldsymbol{C})}}.$$

Meanwhile, Theorems 8 proves that if $\varepsilon \leq \mathcal{O}\left(\frac{\lambda(\boldsymbol{C})}{\Gamma^2 K\log(\frac{\Gamma\sqrt{n\log K}}{\rho})}\right)$ and $k \geq \mathcal{O}\left(\Gamma^{10}\frac{K^2\|\boldsymbol{C}\|^4}{\alpha_{\max}^2\lambda(\boldsymbol{C})^4}\log(\frac{\Gamma\sqrt{n\log K}}{\rho})^6\right)$, then it holds

$$\|\widetilde{\boldsymbol{r}}_t - \boldsymbol{r}_t\|_2 \leq c\varepsilon\frac{\psi' K}{n\lambda(\boldsymbol{C})}\log(\frac{\Gamma\sqrt{n\log K}}{\rho})\Gamma^3 n^{3/2}\sqrt{\log K} = c\frac{\psi'\varepsilon\Gamma^3 K\sqrt{n\log K}}{\lambda(\boldsymbol{C})}\log(\frac{\Gamma\sqrt{n\log K}}{\rho})$$

and

$$\|\boldsymbol{W}_t - \widetilde{\boldsymbol{W}}_t\|_F \leq \mathcal{O}\left(t\psi'\eta\varepsilon\frac{\Gamma^4 Kn}{\lambda(\boldsymbol{C})}\log\left(\frac{\Gamma\sqrt{n\log K}}{\rho}\right)^2\right).$$

where $\psi' = 1 + \frac{\psi_1}{2} + \sqrt{\psi_2}$.

**Step 1.** By using the above two results, we have

$$\frac{\|f(\boldsymbol{W}_t, \boldsymbol{X}) - \widetilde{\boldsymbol{y}}\|_2}{\sqrt{n}} = \frac{1}{\sqrt{n}}\left(\|\widetilde{\boldsymbol{r}}_t\|_2 + \|\boldsymbol{r}_t - \widetilde{\boldsymbol{r}}_t\|_2\right) \leq 4\rho + c\frac{\varepsilon\psi'\Gamma^3 K\sqrt{\log K}}{\lambda(\boldsymbol{C})}\log\left(\frac{\Gamma\sqrt{n\log K}}{\rho}\right).$$

Moreover, we can also upper bound

$$
\begin{aligned}
\frac{\|\bar{\boldsymbol{y}}^t - \boldsymbol{y}^*\|_2}{\sqrt{n}} &\leq \frac{(1-\alpha_t)\|\boldsymbol{y} - \boldsymbol{y}^*\|_2}{\sqrt{n}} + \frac{\alpha_t\|f(\boldsymbol{W}_t, \boldsymbol{X}) - \boldsymbol{y}^*\|_2}{\sqrt{n}} \\
&= \frac{(1-\alpha_t)\|\boldsymbol{y} - \boldsymbol{y}^*\|_2}{\sqrt{n}} + 4\alpha_t\rho + c\alpha_t\frac{\varepsilon\psi'\Gamma^3 K\sqrt{\log K}}{\lambda(\boldsymbol{C})}\log\left(\frac{\Gamma\sqrt{n\log K}}{\rho}\right).
\end{aligned}
$$

**Step 2.** Now we consider what cases that our method can exactly recover the ground truth label. Assume an input $\boldsymbol{x}$ is within $\varepsilon$-neighborhood of one of the cluster centers $\boldsymbol{c} \in (\boldsymbol{c}_\ell)_{\ell=1}^K$. Then we try to upper bound $|f(\boldsymbol{W}_t, \boldsymbol{x}) - f(\widetilde{\boldsymbol{W}}_t, \boldsymbol{c})|$ where $f(\widetilde{\boldsymbol{W}}_t, \boldsymbol{c})$ corresponds to $f(\widetilde{\boldsymbol{W}}_t, \widetilde{\boldsymbol{x}})$. To begin with, we have

$$|f(\boldsymbol{W}_t, \boldsymbol{x}) - f(\widetilde{\boldsymbol{W}}_t, \boldsymbol{c})| \leq |f(\boldsymbol{W}_t, \boldsymbol{x}) - f(\widetilde{\boldsymbol{W}}_t, \boldsymbol{x})| + |f(\widetilde{\boldsymbol{W}}_t, \boldsymbol{x}) - f(\widetilde{\boldsymbol{W}}_t, \boldsymbol{c})|$$

We upper bound the first term as follows:

$$
\begin{aligned}
|f(\boldsymbol{W}_t, \boldsymbol{x}) - f(\widetilde{\boldsymbol{W}}_t, \boldsymbol{x})| &= |\boldsymbol{v}^T\phi(\boldsymbol{W}_t\boldsymbol{x}) - \boldsymbol{v}^T\phi(\widetilde{\boldsymbol{W}}_t\boldsymbol{x})| \leq \|\boldsymbol{v}\|_2\|\phi(\boldsymbol{W}_t\boldsymbol{x}) - \phi(\widetilde{\boldsymbol{W}}_t\boldsymbol{x})\|_2 \\
&\leq \Gamma\|\boldsymbol{W}_t - \widetilde{\boldsymbol{W}}_t\|_F \\
&\leq \mathcal{O}\left(\varepsilon\psi'\frac{\Gamma^5 K^2}{\lambda(\boldsymbol{C})^2}\log(\frac{\Gamma\sqrt{n\log K}}{\rho})^3\right)
\end{aligned}
$$

where we use the results $\|\boldsymbol{W}_t - \widetilde{\boldsymbol{W}}_t\|_F \leq \mathcal{O}\left(t\psi'\eta\varepsilon\frac{\Gamma^4 Kn}{\lambda(\boldsymbol{C})}\log\left(\frac{\Gamma\sqrt{n\log K}}{\rho}\right)^2\right)$ with $\psi' = 1 + \frac{\psi_1}{2} + \sqrt{\psi_2}$ in Theorem 8, and $t = t_0$. Next, we need to bound

$$|f(\widetilde{\boldsymbol{W}}_t, \boldsymbol{x}) - f(\widetilde{\boldsymbol{W}}_t, \boldsymbol{c})| \leq |\boldsymbol{v}^T\phi(\widetilde{\boldsymbol{W}}_t\boldsymbol{x}) - \boldsymbol{v}^T\phi(\widetilde{\boldsymbol{W}}_t\boldsymbol{c})|.$$

On the other hand, we have $\|\widetilde{\boldsymbol{W}}_t - \boldsymbol{W}_0\|_F \leq \mathcal{O}\left(\Gamma\sqrt{\frac{K\log K}{\lambda(\boldsymbol{C})}}\right)$ in Theorem 7, $\|\boldsymbol{x} - \boldsymbol{c}\|_2 \leq \varepsilon$ and $\boldsymbol{W}_0 \sim \mathcal{N}(0, \boldsymbol{I})$ in assumption. Moreover, using by assumption we have

$$k \geq \mathcal{O}\left(\|\widetilde{\boldsymbol{W}}_t - \boldsymbol{W}_0\|_F^2\right) = \mathcal{O}\left(\Gamma^2\frac{K\log K}{\lambda(\boldsymbol{C})}\right).$$

By using the above results, Theorem 5 guarantees that with probability at $1 - K\exp(-100d)$, for all inputs $\boldsymbol{x}$ lying $\varepsilon$ neighborhood of cluster centers, it holds that

$$|f(\boldsymbol{W}_t, \boldsymbol{x}) - f(\widetilde{\boldsymbol{W}}_t, \boldsymbol{c})| \leq C'\Gamma\varepsilon(\|\widetilde{\boldsymbol{W}}_t - \boldsymbol{W}_0\|_F + \sqrt{d}) \leq C\Gamma\varepsilon\left(\Gamma\sqrt{\frac{K\log K}{\lambda(\boldsymbol{C})}} + \sqrt{d}\right). \quad (17)$$

Combining the two bounds above we get

$$
\begin{aligned}
|f(\boldsymbol{W}_t, \boldsymbol{x}) - f(\widetilde{\boldsymbol{W}}_t, \boldsymbol{c})| &\leq \varepsilon\mathcal{O}\left(\frac{\psi'\Gamma^5 K^2}{\lambda(\boldsymbol{C})^2}\log(\frac{\Gamma\sqrt{n\log K}}{\rho})^3 + \Gamma(\Gamma\sqrt{\frac{K\log K}{\lambda(\boldsymbol{C})}} + \sqrt{d})\right) \\
&\leq \varepsilon\mathcal{O}\left(\frac{\psi'\Gamma^5 K^2}{\lambda(\boldsymbol{C})^2}\log(\frac{\Gamma\sqrt{n\log K}}{\rho})^3\right).
\end{aligned}
$$

Hence, if $\varepsilon \leq c'\delta\min\left(\frac{\lambda(\boldsymbol{C})^2}{\psi'\Gamma^5 K^2\log(\frac{\Gamma\sqrt{n\log K}}{\rho})^3}, \frac{1}{\Gamma\sqrt{d}}\right)$, we obtain that, for all $\boldsymbol{x}$, the associated cluster $\boldsymbol{c}$ and true label assigned to cluster $\boldsymbol{y}^* = \boldsymbol{y}^*(\boldsymbol{c})$, we have that

$$|f(\boldsymbol{W}_t, \boldsymbol{x}) - \boldsymbol{y}^*| < |f(\widetilde{\boldsymbol{W}}_t, \boldsymbol{c}) - f(\boldsymbol{W}_t, \boldsymbol{x})| + |f(\widetilde{\boldsymbol{W}}_t, \boldsymbol{c}) - \boldsymbol{y}^*| \leq 4\rho + \frac{\delta}{8}.$$

Meanwhile, we can upper bound

$$|\bar{y}_x^t - y_x^*| \le (1 - \alpha_t)|y_x - y_x^*| + \alpha_t|f(W_t, x) - y^*| \le (1 - \alpha_t)|y_x - y_x^*| + \alpha_t(4\rho + \frac{\delta}{8}).$$

where $\bar{y}_x^t = (1 - \alpha_t)y_x + \alpha_t f(W_t, x)$ and $y_x^*$ receptively denote the estimated label by our label refinery and the ground truth label of sample $x$. Since $|y_x - y_x^*| < 1$, by setting $1 \ge \alpha_t \ge 1 - \frac{3}{4}\delta$ and $\rho \le \delta/32$, we have

$$|\bar{y}_x^t - y_x^*| < \frac{\delta}{2}$$

This means that for any sample $x_i$, we have $|\bar{y}_i^t - y_i^*| < \delta/2$. Therefore, our label refinery gives the correct estimated labels for all samples. By using the same setting, we obtain

$$|f(W_t, x) - y^*| < \delta/2.$$

This means that for any sample $x_i$, we have $|f(W_t, x_i) - y_i^*| < \delta/2$. Therefore, $W_t$ gives the correct estimated labels for all samples. This competes all proofs.

$\square$

### C.2.4 Proofs of Auxiliary Theories in Appendix C.2

### C.2.5 Proof of Theorem 6

*Proof.* The proof will be done inductively over the properties of gradient descent iterates and is inspired from the recent work [21, 19]. The main difference is that this work uses the label estimation $\bar{y}^t = (1 - \alpha_t)y + \alpha_t f(w_t)$ and minimizes the squared loss, while both [21, 19] use the corrupted label $y$ and then minimize the squared loss. By comparison, our method is much more complicated and gives different proofs. Let us introduce the notation related to the residual. Set $r_t = f(w_t) - \bar{y}^t$ and let $r_0 = f(w_0) - \bar{y}^0$ be the initial residual. We keep track of the growth of the residual by partitioning the residual as $r_t = \hat{r}_t + \hat{e}_t$ where

$$\hat{e}_t = \mathcal{P}_{\mathcal{S}_-}(r_t) \quad , \quad \hat{r}_t = \mathcal{P}_{\mathcal{S}_+}(r_t).$$

We claim that for all iterations $t \ge 0$, the following conditions hold.

$$\|\hat{e}_t\|_2 \le \|\hat{e}_0\|_2 + \sqrt{n}\sum_{i=0}^{t}|\alpha_i - \alpha_{i+1}| \le \|\hat{e}_0\|_2 + \frac{\psi_1}{2}\|r_0\|_2, \tag{18}$$

$$\|\hat{r}_t\|_2^2 \le \left(1 - \frac{\eta\alpha^2}{4}\right)^t\|\hat{r}_0\|_2^2 + 2\sqrt{n}\sum_{t=0}^{t-1}\left(1 - \frac{\eta\alpha^2}{4}\right)^{t-t}|\alpha_t - \alpha_{t+1}|, \tag{19}$$

$$\frac{\alpha}{4}\|w_t - w_0\|_2 + \|\hat{r}_t\|_2 \le \|\hat{r}_0\|_2 + 2\sqrt{n}\sum_{i=0}^{t}|\alpha_i - \alpha_{i+1}| \le \|r_0\|_2 + 2\sqrt{n}\sum_{i=0}^{t}|\alpha_i - \alpha_{i+1}|$$
$$\le (1 + \phi)\|r_0\|_2, \tag{20}$$

where the last line uses the assumption that $2\sqrt{n}\lim_{t\to+\infty}\sum_{i=0}^{t}|\alpha_i - \alpha_{i+1}| \le \psi_1\|r_0\|_2$. Assuming these conditions hold till some $t > 0$, inductively, we focus on iteration $t + 1$. First, note that these conditions imply that for all $t \ge i \ge 0$, $w_i \in \mathcal{D}$ where $\mathcal{D} = \left\{w \in \mathbb{R}^p \mid \|w - w_0\|_2 \le \frac{4(1+\psi_1)\|r_0\|_2}{\alpha}\right\}$ is the Euclidian ball around $w_0$ of radius $\frac{4(1+\psi_1)\|r_0\|_2}{\alpha}$. This directly follows from (20) induction hypothesis. Next, we claim that $w_{t+1}$ is still within the set $\mathcal{D}$. From Lemma 5, we have that if the results in Eqn. (20) holds, then it holds that

$$w_{t+1} \in \mathcal{D} = \left\{w \in \mathbb{R}^p \mid \|w - w_0\|_2 \le \frac{4(1 + \psi_1)\|r_0\|_2}{\alpha}\right\}.$$

In this way, we can directly use the results in previous lemmas and assumptions. Then we will prove that (19) and (20) hold for $t + 1$ as well. Note that, following Lemma 3, gradient descent iterate can be written as

$$r_{t+1} = (I - \eta G(w_t))r_t + \bar{y}^t - \bar{y}^{t+1}.$$

Since both column and row space of $\boldsymbol{G}(\boldsymbol{w}_t)$ is subset of $\mathcal{S}_+$, we have that

$$\widehat{\boldsymbol{e}}_{t+1} = \mathcal{P}_{\mathcal{S}_-}((\boldsymbol{I} - \eta\boldsymbol{G}(\boldsymbol{w}_t))\boldsymbol{r}_t + \bar{\boldsymbol{y}}^t - \bar{\boldsymbol{y}}^{t+1}) \tag{21}$$

$$= \mathcal{P}_{\mathcal{S}_-}(\boldsymbol{r}_t) + \mathcal{P}_{\mathcal{S}_-}(\bar{\boldsymbol{y}}^t - \bar{\boldsymbol{y}}^{t+1}) \tag{22}$$

$$= \widehat{\boldsymbol{e}}_t + \mathcal{P}_{\mathcal{S}_-}(\bar{\boldsymbol{y}}^t - \bar{\boldsymbol{y}}^{t+1}) \tag{23}$$

$$= \widehat{\boldsymbol{e}}_t + \mathcal{P}_{\mathcal{S}_-}((\alpha_{t+1} - \alpha_t)\boldsymbol{y}) \tag{24}$$

$$= \widehat{\boldsymbol{e}}_0 + \sum_{t=0}^{t} \mathcal{P}_{\mathcal{S}_-}((\alpha_{t+1} - \alpha_t)\boldsymbol{y}) \tag{25}$$

$$\tag{26}$$

So we can upper bound

$$\|\widehat{\boldsymbol{e}}_t\|_2 \le \|\widehat{\boldsymbol{e}}_0\|_2 + 2\sqrt{n}\sum_{i=0}^{t}|\alpha_i - \alpha_{i+1}| \le \|\widehat{\boldsymbol{e}}_0\|_2 + \psi_1\|\boldsymbol{r}_0\|_2. \tag{27}$$

This shows the first statement of the induction. Next, over $\mathcal{S}_+$, we have

$$\widehat{\boldsymbol{r}}_{t+1} = \mathcal{P}_{\mathcal{S}_+}((\boldsymbol{I} - \eta\boldsymbol{G}(\boldsymbol{w}_t))\boldsymbol{r}_t + \bar{\boldsymbol{y}}^t - \bar{\boldsymbol{y}}^{t+1}) \tag{28}$$

$$= \mathcal{P}_{\mathcal{S}_+}((\boldsymbol{I} - \eta\boldsymbol{G}(\boldsymbol{w}_t))\widehat{\boldsymbol{r}}_t) + \mathcal{P}_{\mathcal{S}_+}((\boldsymbol{I} - \eta\boldsymbol{G}(\boldsymbol{w}_t))\widehat{\boldsymbol{e}}_t) + \mathcal{P}_{\mathcal{S}_+}(\bar{\boldsymbol{y}}^t - \bar{\boldsymbol{y}}^{t+1}) \tag{29}$$

$$= \mathcal{P}_{\mathcal{S}_+}((\boldsymbol{I} - \eta\boldsymbol{G}(\boldsymbol{w}_t))\widehat{\boldsymbol{r}}_t) + \mathcal{P}_{\mathcal{S}_+}(\bar{\boldsymbol{y}}^t - \bar{\boldsymbol{y}}^{t+1}) \tag{30}$$

$$= (\boldsymbol{I} - \eta\boldsymbol{G}(\boldsymbol{w}_t))\widehat{\boldsymbol{r}}_t + \bar{\boldsymbol{y}}^t - \bar{\boldsymbol{y}}^{t+1} \tag{31}$$

where the second line uses the fact that $\widehat{\boldsymbol{e}}_t \in \mathcal{S}_-$ and last line uses the fact that $\widehat{\boldsymbol{r}}_t \in \mathcal{S}_+$, in the last line, we let $\widehat{\boldsymbol{y}}_t = \mathcal{P}_{\mathcal{S}_+}(\bar{\boldsymbol{y}}^t)$. Then we can rewrite $\bar{\boldsymbol{y}}^t - \bar{\boldsymbol{y}}^{t+1}$ as

$$\widehat{\boldsymbol{y}}_t - \widehat{\boldsymbol{y}}_{t+1} = (1 - \alpha_t)\boldsymbol{y} + \alpha_t f(\boldsymbol{w}_t) - (1 - \alpha_{t+1})\boldsymbol{y} + \alpha_{t+1}f(\boldsymbol{w}_{t+1})$$
$$= (\alpha_{t+1} - \alpha_t)\boldsymbol{y} + \alpha_t(f(\boldsymbol{w}_t) - f(\boldsymbol{w}_{t+1})) - (\alpha_{t+1} - \alpha_t)f(\boldsymbol{w}_{t+1}).$$

At the same time, we can upper bound

$$\|\boldsymbol{w}_{t+1} - \boldsymbol{w}_t\|_F = \eta\|\mathcal{J}(\boldsymbol{w}_t)^T\boldsymbol{r}_t\|_2 \overset{①}{\le} \eta\|\mathcal{J}(\boldsymbol{w}_t)^T\widehat{\boldsymbol{r}}_t\|_2 \le \eta\beta\|\widehat{\boldsymbol{r}}_t\|_2.$$

In this way, we can obtain

$$\|\widehat{\boldsymbol{r}}_{t+1}\|_2$$
$$\le \|(\boldsymbol{I} - \eta\boldsymbol{G}(\boldsymbol{w}_t))\widehat{\boldsymbol{r}}_t\|_2 + \|(\alpha_t - \alpha_{t+1})\boldsymbol{y}\|_2 + \alpha_t\|f(\boldsymbol{w}_t) - f(\boldsymbol{w}_{t+1})\|_2 + \|(\alpha_{t+1} - \alpha_t)f(\boldsymbol{w}_{t+1})\|_2$$
$$\overset{①}{\le} \left(1 - \frac{\eta\alpha^2}{2}\right)\|\widehat{\boldsymbol{r}}_t\|_2 + \alpha_t\beta\|\boldsymbol{w}_t - \boldsymbol{w}_{t+1}\|_2 + 2\sqrt{n}\cdot|\alpha_t - \alpha_{t+1}|$$
$$\le \left(1 - \frac{\eta\alpha^2}{2}\right)\|\widehat{\boldsymbol{r}}_t\|_2 + \alpha_t\beta^2\eta\|\widehat{\boldsymbol{r}}_t\|_2 + 2\sqrt{n}\cdot|\alpha_t - \alpha_{t+1}|$$
$$\overset{②}{\le} \left(1 - \frac{\eta\alpha^2}{4}\right)\|\widehat{\boldsymbol{r}}_t\|_2 + 2\sqrt{n}\cdot|\alpha_t - \alpha_{t+1}|$$

where ① uses in Lemma 6, $\|\boldsymbol{y}\|_2 \le \sqrt{n}$ and $\|f(\boldsymbol{w}_{t+1})\|_2 \le \sqrt{n}$, ② uses $\alpha_t \le \frac{\alpha^2}{4\beta^2}$. This result further yields

$$\|\widehat{\boldsymbol{r}}_t\|_2 \le \left(1 - \frac{\eta\alpha^2}{4}\right)^t \|\widehat{\boldsymbol{r}}_0\|_2 + 2\sqrt{n}\sum_{t=0}^{t-1}\left(1 - \frac{\eta\alpha^2}{4}\right)^{t-t}|\alpha_t - \alpha_{t+1}|$$

On the other hand, we have

$$\|(\boldsymbol{I} - \eta\boldsymbol{G}(\boldsymbol{w}_t))\widehat{\boldsymbol{r}}_t\|_2^2 \le \|\widehat{\boldsymbol{r}}_t\|_2^2 - 2\eta\widehat{\boldsymbol{r}}_t^T\boldsymbol{G}(\boldsymbol{w}_t)\widehat{\boldsymbol{r}}_t + \eta^2\widehat{\boldsymbol{r}}_t^T\boldsymbol{G}^T(\boldsymbol{w}_t)\boldsymbol{G}(\boldsymbol{w}_t)\widehat{\boldsymbol{r}}_t$$
$$\le \|\widehat{\boldsymbol{r}}_t\|_2^2 - 2\eta\widehat{\boldsymbol{r}}_t^T\mathcal{J}(\boldsymbol{w}_t)\mathcal{J}^T(\boldsymbol{w}_t)\widehat{\boldsymbol{r}}_t + \eta^2\beta^2\widehat{\boldsymbol{r}}_t^T\mathcal{J}(\boldsymbol{w}_t)\mathcal{J}^T(\boldsymbol{w}_t)\widehat{\boldsymbol{r}}_t$$
$$= \|\widehat{\boldsymbol{r}}_t\|_2^2 - \eta(2 - \eta\beta^2)\|\mathcal{J}^T(\boldsymbol{w}_t)\widehat{\boldsymbol{r}}_t\|_2^2$$
$$\le \|\widehat{\boldsymbol{r}}_t\|_2^2 - \eta\|\mathcal{J}^T(\boldsymbol{w}_t)\widehat{\boldsymbol{r}}_t\|_2^2,$$

where the last line use $\eta \leq \frac{1}{\beta^2}$. This further gives

$$\|(\boldsymbol{I} - \eta \boldsymbol{G}(\boldsymbol{w}_t))\widehat{\boldsymbol{r}}_t\|_2 \leq \sqrt{\|\widehat{\boldsymbol{r}}_t\|_2^2 - \eta\|\mathcal{J}^T(\boldsymbol{w}_t)\widehat{\boldsymbol{r}}_t\|_2^2} \leq \|\widehat{\boldsymbol{r}}_t\|_2 - \frac{\eta}{2}\frac{\|\mathcal{J}^T(\boldsymbol{w}_t)\widehat{\boldsymbol{r}}_t\|_2^2}{\|\widehat{\boldsymbol{r}}_t\|_2}.$$

Therefore, we can upper bound $\|\widehat{\boldsymbol{r}}_t\|_2$ in another way which can help to bound $\|\boldsymbol{w}_{t+1} - \boldsymbol{w}_0\|_2$:

$$\|\widehat{\boldsymbol{r}}_{t+1}\|_2$$
$$\leq \|(\boldsymbol{I} - \eta \boldsymbol{G}(\boldsymbol{w}_t))\widehat{\boldsymbol{r}}_t\|_2 + \|(\alpha_t - \alpha_{t+1})\boldsymbol{y}\|_2 + (1 - \alpha_t)\|f(\boldsymbol{w}_t) - f(\boldsymbol{w}_{t+1})\|_2 + \|(\alpha_{t+1} - \alpha_t)f(\boldsymbol{w}_{t+1})\|_2$$
$$\leq \|(\boldsymbol{I} - \eta \boldsymbol{G}(\boldsymbol{w}_t))\widehat{\boldsymbol{r}}_t\|_2 + (1 - \alpha_t)\beta\|\boldsymbol{w}_t - \boldsymbol{w}_{t+1}\|_2 + 2\sqrt{n} \cdot |\alpha_t - \alpha_{t+1}|$$
$$= \|(\boldsymbol{I} - \eta \boldsymbol{G}(\boldsymbol{w}_t))\widehat{\boldsymbol{r}}_t\|_2 + (1 - \alpha_t)\beta\eta\|\mathcal{J}^T(\boldsymbol{w}_t)\boldsymbol{r}_t\|_2 + 2\sqrt{n} \cdot |\alpha_t - \alpha_{t+1}|$$
$$\leq \|\widehat{\boldsymbol{r}}_t\|_2 - \frac{\eta}{2}\frac{\|\mathcal{J}^T(\boldsymbol{w}_t)\widehat{\boldsymbol{r}}_t\|_2^2}{\|\widehat{\boldsymbol{r}}_t\|_2} + (1 - \alpha_t)\beta\eta\|\mathcal{J}^T(\boldsymbol{w}_t)\boldsymbol{r}_t\|_2 + 2\sqrt{n} \cdot |\alpha_t - \alpha_{t+1}|.$$

Since the distance of $\boldsymbol{w}_{t+1}$ to initial point satisfies :

$$\|\boldsymbol{w}_{t+1} - \boldsymbol{w}_0\|_2 \leq \|\boldsymbol{w}_{t+1} - \boldsymbol{w}_t\|_2 + \|\boldsymbol{w}_t - \boldsymbol{w}_0\|_2 \leq \|\boldsymbol{w}_t - \boldsymbol{w}_0\|_2 + \eta\|\mathcal{J}^T(\boldsymbol{w}_t)\boldsymbol{r}_t\|_2,$$

we can further bound

$$\frac{\alpha}{4}\|\boldsymbol{w}_{t+1} - \boldsymbol{w}_0\|_2 + \|\widehat{\boldsymbol{r}}_{t+1}\|_2$$

$$\leq \frac{\alpha}{4}\left(\|\boldsymbol{w}_t - \boldsymbol{w}_0\|_2 + \eta\|\mathcal{J}^T(\boldsymbol{w}_t)\boldsymbol{r}_t\|_2\right) + \|\widehat{\boldsymbol{r}}_t\|_2 - \frac{\eta}{2}\frac{\|\mathcal{J}^T(\boldsymbol{w}_t)\widehat{\boldsymbol{r}}_t\|_2^2}{\|\widehat{\boldsymbol{r}}_t\|_2}$$
$$\quad + (1 - \alpha_t)\beta\eta\|\mathcal{J}^T(\boldsymbol{w}_t)\boldsymbol{r}_t\|_2 + 2\sqrt{n} \cdot |\alpha_t - \alpha_{t+1}|$$

$$\leq \frac{\alpha}{4}\|\boldsymbol{w}_t - \boldsymbol{w}_0\|_2 + \|\widehat{\boldsymbol{r}}_t\|_2 + \frac{\eta}{4}\|\mathcal{J}^T(\boldsymbol{w}_t)\boldsymbol{r}_t\|_2\left(\alpha + 4(1 - \alpha_t)\beta - 2\frac{\|\mathcal{J}^T(\boldsymbol{w}_t)\widehat{\boldsymbol{r}}_t\|_2}{\|\widehat{\boldsymbol{r}}_t\|_2}\right) + 2\sqrt{n} \cdot |\alpha_t - \alpha_{t+1}|$$

$$\overset{\textcircled{1}}{\leq} \frac{\alpha}{4}\|\boldsymbol{w}_t - \boldsymbol{w}_0\|_2 + \|\widehat{\boldsymbol{r}}_t\|_2 + 2\sqrt{n} \cdot |\alpha_t - \alpha_{t+1}|$$

$$\leq \|\widehat{\boldsymbol{r}}_0\|_2 + 2\sqrt{n}\sum_{i=0}^{t}|\alpha_i - \alpha_{i+1}| \leq \|\boldsymbol{r}_0\|_2 + 2\sqrt{n}\sum_{i=0}^{t}|\alpha_i - \alpha_{i+1}|,$$

where ① uses $\frac{\|\mathcal{J}^T(\boldsymbol{w}_t)\widehat{\boldsymbol{r}}_t\|_2}{\|\widehat{\boldsymbol{r}}_t\|_2} \geq \alpha$ and $\alpha_t \leq \frac{\alpha}{4\beta}$.

By setting $t \geq \frac{5}{\eta\alpha^2}\log\left(\frac{\|\boldsymbol{r}_0\|_2}{(1-\alpha_{\max})\nu}\right)$ and $\frac{\eta\alpha^2}{4} \leq \frac{\eta\beta^2}{4} \leq \frac{1}{8}$ where $\alpha_{\max} = \max_t \alpha_t$, then we have $\log\frac{1}{1-\frac{\eta\alpha^2}{4}} \geq \log\left(1 + \frac{\eta\alpha^2}{4}\right) \geq \frac{\eta\alpha^2}{5}$ and thus

$$\left(1 - \frac{\eta\alpha^2}{4}\right)^t\|\widehat{\boldsymbol{r}}_0\|_2 \leq \left(1 - \frac{\eta\alpha^2}{4}\right)^t\|\boldsymbol{r}_0\|_2 \leq \left(1 - \frac{\eta\alpha^2}{4}\right)^t\|\boldsymbol{r}_0\|_2 \leq (1 - \alpha_{\max})\nu.$$

In this way, we can further obtain

$$\|\widehat{\boldsymbol{r}}_t\|_\infty \leq \|\widehat{\boldsymbol{r}}_t\|_2 \leq (1 - \alpha_{\max})\nu + 2\sqrt{n}\sum_{i=0}^{t-1}\left(1 - \frac{\eta\alpha^2}{4}\right)^{t-i}|\alpha_i - \alpha_{i+1}|$$

and

$$(1 - \alpha_t)\|\mathcal{P}_{\mathcal{S}_+}(f(\boldsymbol{w}_t) - \boldsymbol{y})\|_\infty = \|\mathcal{P}_{\mathcal{S}_+}(f(\boldsymbol{w}_t) - (1 - \alpha_t)\boldsymbol{y} - \alpha_t f(\boldsymbol{w}_t))\|_\infty = \|\widehat{\boldsymbol{r}}_t\|_\infty \leq \|\widehat{\boldsymbol{r}}_t\|_2$$

$$\leq (1 - \alpha_{\max})\nu + 2\sqrt{n}\sum_{t=0}^{t-1}\left(1 - \frac{\eta\alpha^2}{4}\right)^{t-t}|\alpha_t - \alpha_{t+1}|$$

Finally, we can obtain the desired results:

$$\|f(\boldsymbol{w}_t) - \boldsymbol{y}^*\|_\infty \overset{\textcircled{1}}{=} \|\mathcal{P}_{\mathcal{S}_+}(f(\boldsymbol{w}_t)) - \mathcal{P}_{\mathcal{S}_+}(\boldsymbol{y}^*)\|_\infty$$
$$\leq \|\mathcal{P}_{\mathcal{S}_+}(f(\boldsymbol{w}_t) - \boldsymbol{y})\|_\infty + \|\mathcal{P}_{\mathcal{S}_+}(\boldsymbol{y} - \boldsymbol{y}^*)\|_\infty$$
$$\leq 2\nu + \frac{2\sqrt{n}}{1 - \alpha_t}\sum_{i=0}^{t-1}\left(1 - \frac{\eta\alpha^2}{4}\right)^{t-i}|\alpha_i - \alpha_{i+1}|,$$

where ① holds since $f(\boldsymbol{w}_t) - \boldsymbol{y}^* \in \mathcal{S}_+$ and $\|\mathcal{P}_{\mathcal{S}_+}(f(\boldsymbol{w}_t) - \boldsymbol{y})\|_\infty = \|\mathcal{P}_{\mathcal{S}_+}(f(\boldsymbol{w}_t) - \boldsymbol{y})\|_\infty$. If $\boldsymbol{e}$ is $s$ sparse and $\mathcal{S}_+$ is diffused, applying Definition 3 we have

$$\|\mathcal{P}_{\mathcal{S}_+}(\boldsymbol{e})\|_\infty \leq \frac{\gamma\sqrt{s}}{n}\|\boldsymbol{e}\|_\infty.$$

The proof is completed. $\qquad\square$

### C.2.6 Proof of Theorem 7

*Proof.* The proof is based on the meta Theorem 6, hence we need to verify its Assumptions 2 and 3 with proper values and apply Lemma 8 to get $\|\mathcal{P}_{\mathcal{S}_+}(\boldsymbol{e})\|_\infty$. We will also make significant use of Corollary 4.

Using Corollary 4, Assumption 3 holds with $L = \Gamma\sqrt{\frac{c_{up}n}{kK}}\|\boldsymbol{C}\|$ where $L$ is the Lipschitz constant of Jacobian spectrum. Denote Using Lemma 7 with probability $1 - K^{-100}$, we have that $\|\boldsymbol{r}_0\|_2 = \|\bar{\boldsymbol{y}}^0 - f(\boldsymbol{W}_0)\|_2 = \|\boldsymbol{y} - f(\boldsymbol{W}_0)\|_2 \leq \Gamma\sqrt{c_0 n \log K/128}$ for some $c_0 > 0$. Corollary 4 guarantees a uniform bound for $\beta$, hence in Assumption 2, we pick

$$\beta \leq \sqrt{\frac{c_{up}n}{K}}\Gamma\|\boldsymbol{C}\|.$$

We shall also pick the minimum singular value over $\mathcal{S}_+$ to be

$$\alpha = \frac{\alpha'}{2} \quad \text{where} \quad \alpha' = \sqrt{\frac{c_{low}n\lambda(\boldsymbol{C})}{2K}},$$

We wish to verify Assumption 2 over the radius of

$$R = \frac{4\|f(\boldsymbol{W}_0) - \boldsymbol{y}\|_2}{\alpha} \leq \frac{\Gamma\sqrt{c_0 n \log K/8}}{\alpha} = \Gamma\sqrt{\frac{c_0 n \log K/2}{\frac{c_{low}n\lambda(\boldsymbol{C})}{2K}}} = \Gamma\sqrt{\frac{c_0 K \log K}{c_{low}\lambda(\boldsymbol{C})}},$$

neighborhood of $\boldsymbol{W}_0$. What remains is ensuring that Jacobian over $\mathcal{S}_+$ is lower bounded by $\alpha$. Our choice of $k$ guarantees that at the initialization, with probability $1 - K^{-100}$, we have

$$\sigma(\mathcal{J}(\boldsymbol{W}_0, \boldsymbol{X}), \mathcal{S}_+) \geq \alpha'.$$

Suppose $LR \leq \alpha = \alpha'/2$ which can be achieved by using large $k$. Using triangle inequality on Jacobian spectrum, for any $\boldsymbol{W} \in \mathcal{D}$, using $\|\boldsymbol{W} - \boldsymbol{W}_0\|_F \leq R$, we would have

$$\sigma(\mathcal{J}(\boldsymbol{W}, \boldsymbol{X}), \mathcal{S}_+) \geq \sigma(\mathcal{J}(\boldsymbol{W}_0, \boldsymbol{X}), \mathcal{S}_+) - LR \geq \alpha' - \alpha = \alpha.$$

Now, observe that

$$LR = (1 + \psi_1)\Gamma\sqrt{\frac{c_{up}n}{kK}}\|\boldsymbol{C}\|\Gamma\sqrt{\frac{c_0 K \log(K)}{c_{low}\lambda(\boldsymbol{C})}} = (1 + \psi_1)\Gamma^2\|\boldsymbol{C}\|\sqrt{\frac{c_{up}c_0 n \log K}{c_{low}k\lambda(\boldsymbol{C})}} \tag{32}$$

$$\leq \frac{\alpha'}{2} = \sqrt{\frac{c_{low}n\lambda(\boldsymbol{C})}{8K}}, \tag{33}$$

as $k$ satisfies

$$k \geq \mathcal{O}\left((1 + \psi_1)^2\Gamma^4\|\boldsymbol{C}\|^2\frac{c_{up}K\log(K)}{c_{low}^2\lambda(\boldsymbol{C})^2}\right) \geq \mathcal{O}\left(\frac{(1 + \psi_1)^2\Gamma^4 K \log(K)\|\boldsymbol{C}\|^2}{\lambda(\boldsymbol{C})^2}\right).$$

Finally, since $LR = 4(1 + \psi_1)L\|\boldsymbol{r}_0\|_2/\alpha \leq \alpha$, the learning rate is

$$\eta \leq \frac{1}{2\beta^2}\min(1, \frac{\alpha\beta}{L\|\boldsymbol{r}_0\|_2}) = \frac{1}{2\beta^2} = \frac{K}{2c_{up}n\Gamma^2\|\boldsymbol{C}\|^2}.$$

Overall, the assumptions of Theorem 6 holds with stated $\alpha, \beta, L$ with probability $1 - 2K^{-100}$ (union bounding initial residual and minimum singular value events). This implies for all $t > 0$ the distance of current iterate to initial obeys

$$\|\boldsymbol{W}_t - \boldsymbol{W}_0\|_F \leq R.$$

The final step is the properties of the label corruption. Using Lemma 8, we find that

$$\|\mathcal{P}_{\mathcal{S}_+}(\boldsymbol{y}^* - \boldsymbol{y})\|_\infty \le 2\rho.$$

Substituting the values corresponding to $\alpha, \beta, L$ yields that, for all gradient iterations with

$$\frac{5}{\eta\alpha^2} \log\left(\frac{\|\boldsymbol{r}_0\|_2}{2(1 - \alpha_{\max})\rho}\right) \le \frac{5}{\eta\alpha^2} \log\left(\frac{\Gamma\sqrt{c_0 n \log K / 32}}{2(1 - \alpha_{\max})\rho}\right) = \mathcal{O}\left(\frac{K}{\eta n \lambda(\boldsymbol{C})} \log\left(\frac{\Gamma\sqrt{n \log K}}{(1 - \alpha_{\max})\rho}\right)\right) \le t,$$

denoting the clean labels by $\widetilde{\boldsymbol{y}}$ and applying Theorem 6, we have that, the infinity norm of the residual obeys (using $\|\mathcal{P}_{\mathcal{S}_+}(\boldsymbol{e})\|_\infty = \|\mathcal{P}_{\mathcal{S}_+}(\boldsymbol{y} - \boldsymbol{y}^*)\|_\infty \le 2\rho$)

$$\|f(\boldsymbol{W}) - \boldsymbol{y}^*\|_\infty \le 4\rho.$$

This implies that if $\rho \le \delta/8$, the network will miss the correct label by at most $\delta/2$, hence all labels (including noisy ones) will be correctly classified. $\qquad\square$

### C.2.7 Proof of Theorem 8

*Proof.* Since $\widetilde{\boldsymbol{W}}_t$ are the noiseless iterations, with probability $1 - 2K^{-100}$, the statements of Theorem 7 hold on $\widetilde{\boldsymbol{W}}_t$. To proceed with proof, we first introduce short hand notations. We use

$$\boldsymbol{r}_i = f(\boldsymbol{W}_i, \boldsymbol{X}) - \bar{\boldsymbol{y}}^i, \; \widetilde{\boldsymbol{r}}_i = f(\widetilde{\boldsymbol{W}}_i, \widetilde{\boldsymbol{X}}_i) - \widetilde{\boldsymbol{y}}^i \tag{34}$$

$$\mathcal{J}_i = \mathcal{J}(\boldsymbol{W}_i, \boldsymbol{X}), \; \mathcal{J}_{i+1,i} = \mathcal{J}(\boldsymbol{W}_{i+1}, \boldsymbol{W}_i, \boldsymbol{X}), \; \widetilde{\mathcal{J}}_i = \mathcal{J}(\widetilde{\boldsymbol{W}}_i, \widetilde{\boldsymbol{X}}), \; \widetilde{\mathcal{J}}_{i+1,i} = \mathcal{J}(\widetilde{\boldsymbol{W}}_{i+1}, \widetilde{\boldsymbol{W}}_i, \widetilde{\boldsymbol{X}}) \tag{35}$$

$$d_i = \|\boldsymbol{W}_i - \widetilde{\boldsymbol{W}}_i\|_F, \; p_i = \|\boldsymbol{r}_i - \widetilde{\boldsymbol{r}}_i\|_2, \; \beta = \Gamma\|\boldsymbol{C}\|\sqrt{c_{up} n / K}, \; L = \Gamma\|\boldsymbol{C}\|\sqrt{c_{up} n / Kk}. \tag{36}$$

Here $\beta$ is the upper bound on the Jacobian spectrum and $L$ is the spectral norm Lipschitz constant as in Theorem 4. Applying Lemma 9, note that

$$\|\mathcal{J}(\boldsymbol{W}_t, \boldsymbol{X}) - \mathcal{J}(\widetilde{\boldsymbol{W}}_t, \widetilde{\boldsymbol{X}})\| \le L\|\widetilde{\boldsymbol{W}}_t - \boldsymbol{W}_t\|_2 + \Gamma\sqrt{n}\varepsilon \le Ld_t + \Gamma\sqrt{n}\varepsilon \tag{37}$$

$$\|\mathcal{J}(\boldsymbol{W}_{t+1}, \boldsymbol{W}_t, \boldsymbol{X}) - \mathcal{J}(\widetilde{\boldsymbol{W}}_{t+1}, \widetilde{\boldsymbol{W}}_t, \widetilde{\boldsymbol{X}})\| \le L(d_t + d_{t+1})/2 + \Gamma\sqrt{n}\varepsilon. \tag{38}$$

By defining

$$\widehat{\boldsymbol{e}}_t = \mathcal{P}_{\mathcal{S}_-}(\widetilde{\boldsymbol{r}}_t) \quad, \quad \widehat{\boldsymbol{r}}_t = \mathcal{P}_{\mathcal{S}_+}(\widetilde{\boldsymbol{r}}_t),$$

then we can use Theorem 6 and the assumption that $2\sqrt{n} \sum_{i=0}^{t-1} \left(1 - \frac{\eta\alpha^2}{4}\right)^{t-i} |\alpha_i - \alpha_{i+1}| \le \psi_2\|\boldsymbol{r}_0\|_2^2$ to obtain

$$\|\widehat{\boldsymbol{e}}_t\|_2 \le \|\widehat{\boldsymbol{e}}_0\|_2 + \sqrt{n} \sum_{i=0}^t |\alpha_i - \alpha_{i+1}| \le \|\widehat{\boldsymbol{e}}_0\|_2 + \frac{\psi}{2}\|\boldsymbol{r}_0\|_2, \tag{39}$$

$$\|\widehat{\boldsymbol{r}}_t\|_2^2 \le \left(1 - \frac{\eta\alpha^2}{4}\right)^t \|\widehat{\boldsymbol{r}}_0\|_2^2 + 2\sqrt{n} \sum_{i=0}^{t-1} \left(1 - \frac{\eta\alpha^2}{4}\right)^{t-i} |\alpha_i - \alpha_{i+1}| \le \|\widehat{\boldsymbol{r}}_0\|_2^2 + \psi_2\|\boldsymbol{r}_0\|_2^2. \tag{40}$$

Therefore, we can upper bound

$$\|\widetilde{\boldsymbol{r}}_t\|_2 = \|\widehat{\boldsymbol{e}}_t\|_2 + \|\widehat{\boldsymbol{r}}_t\|_2 \le \|\widehat{\boldsymbol{e}}_0\|_2 + \frac{\psi}{2}\|\boldsymbol{r}_0\|_2 + \|\widehat{\boldsymbol{r}}_0\|_2 + \sqrt{\psi_2}\|\boldsymbol{r}_0\|_2 = \left(1 + \frac{\psi}{2} + \sqrt{\psi_2}\right)\|\boldsymbol{r}_0\|_2. \tag{41}$$

Following this and setting $\|\widetilde{\boldsymbol{r}}_t\|_2 \le \psi'\|\boldsymbol{r}_0\|_2$, note that parameter satisfies

$$\boldsymbol{W}_{i+1} = \boldsymbol{W}_i - \eta\mathcal{J}_i \boldsymbol{r}_i \quad, \quad \widetilde{\boldsymbol{W}}_{i+1} = \widetilde{\boldsymbol{W}}_i - \eta\widetilde{\mathcal{J}}_i^T \widetilde{\boldsymbol{r}}_i \tag{42}$$

$$\|\boldsymbol{W}_{i+1} - \widetilde{\boldsymbol{W}}_{i+1}\|_F \le \|\boldsymbol{W}_i - \widetilde{\boldsymbol{W}}_i\|_F + \eta\|\mathcal{J}_i - \widetilde{\mathcal{J}}_i\|\|\boldsymbol{r}_i\|_F + \eta\|\mathcal{J}_i\|\|\boldsymbol{r}_i - \widetilde{\boldsymbol{r}}_i\|_2 \tag{43}$$

$$d_{i+1} \le d_i + \eta(\psi'(Ld_i + \Gamma\sqrt{n}\varepsilon)\|\boldsymbol{r}_0\|_2 + \beta p_i), \tag{44}$$

and residual satisfies (using $\boldsymbol{I} \succeq \widetilde{\mathcal{J}}_{i+1,i}\widetilde{\mathcal{J}}_i^T/\beta^2 \succeq 0$)

$$\boldsymbol{r}_{i+1} = \boldsymbol{r}_i - \eta \mathcal{J}_{i+1,i}\mathcal{J}_i^T \boldsymbol{r}_i \implies \tag{45}$$

$$\boldsymbol{r}_{i+1} - \widetilde{\boldsymbol{r}}_{i+1} \tag{46}$$

$$= (\boldsymbol{r}_i - \widetilde{\boldsymbol{r}}_i) - \eta(\mathcal{J}_{i+1,i} - \widetilde{\mathcal{J}}_{i+1,i})\mathcal{J}_i^T \boldsymbol{r}_i - \eta\widetilde{\mathcal{J}}_{i+1,i}(\mathcal{J}_i^T - \widetilde{\mathcal{J}}_i^T)\boldsymbol{r}_i - \eta\widetilde{\mathcal{J}}_{i+1,i}\widetilde{\mathcal{J}}_i^T(\boldsymbol{r}_i - \widetilde{\boldsymbol{r}}_i). \tag{47}$$

$$\boldsymbol{r}_{i+1} - \widetilde{\boldsymbol{r}}_{i+1} = (\boldsymbol{I} - \eta\widetilde{\mathcal{J}}_{i+1,i}\widetilde{\mathcal{J}}_i^T)(\boldsymbol{r}_i - \widetilde{\boldsymbol{r}}_i) - \eta(\mathcal{J}_{i+1,i} - \widetilde{\mathcal{J}}_{i+1,i})\mathcal{J}_i^T \boldsymbol{r}_i - \eta\widetilde{\mathcal{J}}_{i+1,i}(\mathcal{J}_i^T - \widetilde{\mathcal{J}}_i^T)\boldsymbol{r}_i. \tag{48}$$

$$\|\boldsymbol{r}_{i+1} - \widetilde{\boldsymbol{r}}_{i+1}\|_2 \le \|\boldsymbol{r}_i - \widetilde{\boldsymbol{r}}_i\|_2 + \eta\beta\|\boldsymbol{r}_i\|_2(L(3d_t + d_{t+1})/2 + 2\Gamma\sqrt{n}\varepsilon). \tag{49}$$

$$\|\boldsymbol{r}_{i+1} - \widetilde{\boldsymbol{r}}_{i+1}\|_2 \le \|\boldsymbol{r}_i - \widetilde{\boldsymbol{r}}_i\|_2 + \eta\beta(\|\widetilde{\boldsymbol{r}}_0\|_2 + p_i)(L(3d_t + d_{t+1})/2 + 2\Gamma\sqrt{n}\varepsilon). \tag{50}$$

where we used $\|\boldsymbol{r}_i\|_2 \le p_i + \psi'\|\boldsymbol{r}_0\|_2$ and $\|(\boldsymbol{I} - \eta\widetilde{\mathcal{J}}_{i+1,i}\widetilde{\mathcal{J}}_i^T)\boldsymbol{v}\|_2 \le \|\boldsymbol{v}\|_2$ which follows from Lemma 6. This implies

$$p_{i+1} \le p_i + \eta\beta(\psi'\|\boldsymbol{r}_0\|_2 + p_i)(L(3d_t + d_{t+1})/2 + 2\Gamma\sqrt{n}\varepsilon). \tag{51}$$

**Finalizing proof:** Next, using Lemma 7, we have $\|\boldsymbol{r}_0\|_2 \le \Theta := C_0\Gamma\sqrt{n\log K}$. We claim that if

$$\boxed{\varepsilon \le \mathcal{O}\left(\frac{1}{t_0\eta\Gamma^2 n}\right) \le \frac{1}{8t_0\eta\beta\Gamma\sqrt{n}} \quad \text{and} \quad L \le \frac{2}{5t_0\eta\Theta(1 + 8\eta t_0\beta^2)} \le \frac{1}{30(t_0\eta\beta)^2\Theta},} \tag{52}$$

(where we used $\eta t_0\beta^2 \ge 1$), for all $t \le t_0$, we have that

$$p_t \le 8t(1 + \psi')\eta\Gamma\sqrt{n}\varepsilon\Theta\beta \le \Theta \quad, \quad d_t \le 2t\eta\Gamma\sqrt{n}\varepsilon\Theta(\psi' + 8\eta t_0\beta^2). \tag{53}$$

The proof is by induction. Suppose it holds until $t \le t_0 - 1$. At $t + 1$, via (44) we have that

$$\frac{d_{t+1} - d_t}{\eta} \le \psi'(Ld_t\Theta + \Gamma\sqrt{n}\varepsilon\Theta) + 8t_0\eta\beta^2\Gamma\sqrt{n}\varepsilon\Theta \overset{?}{\le} 2\Gamma\sqrt{n}\varepsilon\Theta(\psi' + 8\eta t_0\beta^2).$$

Right hand side holds since $L \le \frac{1}{2\eta t_0\Theta}$. This establishes the induction for $d_{t+1}$.

Next, we show the induction on $p_t$. Observe that $3d_t + d_{t+1} \le 10t_0\eta\Gamma\sqrt{n}\varepsilon\Theta(\psi' + 8\eta t_0\beta^2)$. Following (51) and using $p_t \le \Theta$, we need

$$\frac{p_{t+1} - p_t}{\eta} \le \beta(1 + \psi')\Theta(L(3d_t + d_{t+1}) + 4\Gamma\sqrt{n}\varepsilon) \overset{?}{\le} \frac{8}{\alpha_{\max}}(1 + \psi')\Gamma\sqrt{n}\varepsilon\Theta\beta \iff \tag{54}$$

$$L(3d_t + d_{t+1}) + 4\Gamma\sqrt{n}\varepsilon \overset{?}{\le} \frac{8}{\alpha_{\max}}\Gamma\sqrt{n}\varepsilon \iff \tag{55}$$

$$L(3d_t + d_{t+1}) \overset{?}{\le} \frac{4}{\alpha_{\max}}\Gamma\sqrt{n}\varepsilon \iff \tag{56}$$

$$10\alpha_{\max}Lt_0\eta(1 + 8\eta t_0\beta^2)\Theta \overset{?}{\le} 4 \iff \tag{57}$$

$$L \overset{?}{\le} \frac{2}{5t_0\alpha_{\max}\eta(1 + 8\eta t_0\beta^2)\Theta}, \tag{58}$$

where $\alpha_{\max} = \max_{1 \le t \le t_0} \alpha_t$. Concluding the induction since $L$ satisfies the final line. Consequently, for all $0 \le t \le t_0$, we have that

$$\begin{aligned}
p_t &= \|\boldsymbol{r}_i - \widetilde{\boldsymbol{r}}_i\|_2 = \|f(\boldsymbol{W}_i, \boldsymbol{X}) - \bar{\boldsymbol{y}}^i - f(\widetilde{\boldsymbol{W}}_i, \widetilde{\boldsymbol{X}}_i) + \widetilde{\boldsymbol{y}}^i\|_2 \\
&\overset{①}{\le} \alpha_{\max}\|f(\boldsymbol{W}_i, \boldsymbol{X}) - f(\widetilde{\boldsymbol{W}}_i, \widetilde{\boldsymbol{X}}_i)\|_2 \\
&\le 8t(1 + \psi')\eta\Gamma\sqrt{n}\varepsilon\Theta\beta = c_0 t(1 + \psi')\eta\varepsilon\Gamma^3 n^{3/2}\sqrt{\log K}.
\end{aligned}$$

where ① uses the definition of $\bar{\boldsymbol{y}}^i = (1 - \alpha_i)\boldsymbol{y} + \alpha_i f(\boldsymbol{W}_i, \boldsymbol{X})$ and $\widetilde{\boldsymbol{y}}^i = (1 - \alpha_i)\boldsymbol{y} + \alpha_i f(\widetilde{\boldsymbol{W}}_i, \widetilde{\boldsymbol{X}})$. In this way, we can obtain

$$\|f(\boldsymbol{W}_i, \boldsymbol{X}) - f(\widetilde{\boldsymbol{W}}_i, \widetilde{\boldsymbol{X}}_i)\|_2 \le c_0 t(1 + \psi')\eta\varepsilon\Gamma^3 n^{3/2}\sqrt{\log K}.$$

Next, note that, condition on $L$ is implied by

$$k \geq 1000\Gamma^2 n(t_0\eta\beta)^4\Theta^2/\alpha_{\max}^2 \tag{59}$$

$$= \mathcal{O}\left(\Gamma^4 n \frac{K^4}{\alpha_{\max}^2 n^4 \lambda(\boldsymbol{C})^4} \log(\frac{\Gamma\sqrt{n\log K}}{\rho})^4 (\|\boldsymbol{C}\|\Gamma\sqrt{n/K})^4 (\Gamma\sqrt{n\log K})^2\right) \tag{60}$$

$$= \mathcal{O}\left(\Gamma^{10} \frac{K^2\|\boldsymbol{C}\|^4}{\alpha_{\max}^2 \lambda(\boldsymbol{C})^4} \log(\frac{\Gamma\sqrt{n\log K}}{\rho})^4 \log^2(K)\right) \tag{61}$$

which is implied by $k \geq \mathcal{O}\left(\Gamma^{10} \frac{K^2\|\boldsymbol{C}\|^4}{\alpha_{\max}^2 \lambda(\boldsymbol{C})^4} \log(\frac{\Gamma\sqrt{n\log K}}{\rho})^6\right)$.

Finally, following (53), distance satisfies

$$d_t \leq 20t\psi'\eta^2 t_0 \Gamma\sqrt{n}\varepsilon\Theta\beta^2 \leq \mathcal{O}\left(t\psi'\eta\varepsilon \frac{\Gamma^4 Kn}{\lambda(\boldsymbol{C})} \log(\frac{\Gamma\sqrt{n\log K}}{\rho})^2\right).$$

The proof is completed. $\qquad\square$