# OpenReview forum: "A Theory-Driven Self-Labeling Refinement Method for Contrastive Representation Learning"
_NeurIPS.cc/2021/Conference — NeurIPS 2021 Spotlight_

### Official Review · Reviewer_wb6d · 2021-06-27

**Rating:** 6
**Confidence:** 3

**Summary:**

This paper discusses why inaccurate one-hot label in contrastive learning cannot reveal well semantic similarities between samples, and how these inaccurate label assignment would impair its generalization for semantic instance discrimination.

To address this issue, a novel self-labeling refinement method is proposed to obtain more accurate labels for contrastive learning. This is done via two complementary modules that respectively generates accurate soft-labels and enhances semantic similarity between queries and their positives during training.

Both theoretical analysis and empirical results are provided to support the claims made in this paper.

**Limitations And Societal Impact:**

The limitations and societal impact has been well discussed in the paper.

**Main Review:**

### Originality
The theoretical analysis shows that one should provide accurate labels close to the soft true labels to better capture the underlying semantic similarity between queries and different samples. This is an interesting discovery and it motivates the authors to develop a new self-labeling refinement method that includes soft labels and mixup. Although both of these two techniques are not new, the authors modify each of them in the framework of contrastive learning and make them complementary to each other.

### Quality
* **Theorem-1** shows that the generalization error of MoCo depends on the discrepancy between the assigned labels and the true labels that really reflect the semantical similarity. While this theorem is of vital importance for this paper, the demonstration of it is completely moved to the supplementary materials, which makes the paper not self-content.
* **Theorem-2** leads to several important findings of the method, however, it is difficult to understand the intuitions and mathematical proofs behind it due to the complicated  equations and abrupt transitions between them.
* **Robustness Analysis** is sort of missing for the five hyper-parameters of the proposed method. Regarding the choice for each dataset described in L334-337, more ablation studies should be included for a better understanding of how the proposed method is sensitive to these hyper-parameters.
* **Baseline Performance** is a bit confusing in Table-4 and Table-5. As shown in Eq-9, the SANE model combines the vanilla contrastive loss with weight (1-$\lambda$) and the the momentum mixup loss with weight $\lambda$, the first entry of Table-5 should be the same as the first entry of Table-4, why they are different?
* **Semi-Supervised** setting comparison could be added. Although the authors claim that they focus on self-supervised learning in this paper, a comparison in the semi-supervised setting, e.g., use 1% or 10% labels as done in SwAV, could be interesting and might help to understand better the proposed methods that leverage soft labels.

### Clarity
The motivation of this paper is clear, but the theoretical analysis provided in the main paper is not self-content. Key assumptions and theorems come out suddenly without a smooth roadmap, and the mathematical notations are over-complex (see L232-256). I'd recommend to reshape the Section-3 to make it easier to follow and more understandable.

Discussion of related works seem also a bit rushed, and extend it with more self-supervised learning methods considering label noises could be welcome.

### Significance

While the contrastive loss such as infoNCE become a popular choice in many self-supervised learning methods, the authors discuss why the one-hot label assignment commonly used in these methods could not reflect the true semantic similarity between different samples. This is an important problem and resolving it could benefit many contrastive-based methods as well as the semantic-related downstream tasks.


**Time Spent Reviewing:**

10

---

> ### Author Response · Authors · 2021-08-09
> **Response to Reviewer wb6d**
>
> Thank you for the insightful and encouraging comments! We hope your main concerns can be addressed by the following clarification.
>
> (1)  Due to the page limitation, we defer to the proof of Theorem 1 to appendix which is mentioned in line 150 (below Theorem 1). The proof is about two pages and is too long to put it in the main text which only allows 9 pages.  This is also the reason that many theoretical works  defer their long proof details into appendix. In revision, we will summarize the proofroad map of Theorem 1 in main text for self-contenting.
>
>
> (2) The proofs behind Theorem 2 is relatively complex which is unavoidable sometimes, especially for a theoretical work. So to help readers understand this proof, we already prepare a proof roadmap in Appendix C.2.1 which summarizes the key steps of the proof and can provide a whole picture to the proof. Then we present the proof details in Appendix C.2.  All these are emphasized in line 257 (below Theorem 2).  Moreover, at the beginning of the appendix, we also introduce the whole structure of the appendix, especially for Appendix C which presents the proof of Theorem 2. Finally, this is a theoretical analysis work which unavoidably involves substantial equations, notations, concepts and lemmas, to rigorously prove theoretical results, which could be really demanding in time and efforts for understanding.
>
>
> (3)  For robustness analysis, we investigate the effects of the most important hyper-parameter of SANE, the regularization constant $\lambda$, in Table 5. From the results in Table 5, one can find that our SANE is robust to $\lambda$. For other parameters, we often follow conventional settings, such as temperature parameter $\tau = 0.2$ in MoCo and CLSA, sharpen parameter $\tau' =0.8$ in [20,21],  cosine-increasing confidence $\mu$ in BYOL. This is because it is not affordable for us to select those parameters carefully or perform comprehensive robustness investigation  due to our limited GPUs, as training 2000 epochs on small dataset,  CIFAR10, already  takes about 6 days on a V100 GPU.  Actually limited GPU resource is also the reasons that most of other contrastive learning works, e.g. CLSA, do not perform comprehensive robustness investigation.
>
>
> (4)  The ablation study  in Table 4 to investigate the effects of SANE components are performed on CIFAR10 by running 1000 epochs, since Table 4 needs to run   8 experiments and our GPU resource is very limited. In Table 5, we run 2000 epochs on CIFAR10. So the results in Table 5 are higher than the ones in Table 4, and are consistent with the ones in Table 1 where we also run 2000 epochs. In revision, we will remark  that in Table 5, we train 2000 epochs.
>
>
> (5) For mathematical notations in lines 232-256, we will reduce unnecessary symbols and use a table to
> summarize the notations.  Since this work contributes in several aspects of contrastive learning including a new
> method, a rigorous theoretical justification and an extensive empirical study, the manuscript might read a bit dense especially in notations, definitions and theoretical claims. But we very much appreciate your suggestions about further
> improving the readability, and will try our best to revise Sec. 3, e.g. introducing the notations, assumptions and theories more smoothly.
>
>
> (6)  Thanks. Per your suggestions, we will add the results on simi-supervised setting, and also discuss  the related works in details.

---

> > ### Comment · Reviewer_wb6d · 2021-08-25
> > **Thanks for response**
> >
> > Thanks for the response. I've read all the reviewers' comments and the authors' response. The answers address my most of my concerns. So I upgrade my rating.

---

> > > ### Author Response · Authors · 2021-08-25
> > > **Response to Reviewer wb6d**
> > >
> > > We are sincerely happy that our response helps you better understand and addresses your main concerns. Many thanks to you and your insightful comments again!!

---

### Official Review · Reviewer_uTpF · 2021-07-17

**Rating:** 7
**Confidence:** 4

**Summary:**

The paper argues that the one-hot label for instance discriminative contrastive loss will impair the performance in downstream tasks. By assuming ground-truth soft labels, the authors prove that the generalization of the network depends on the discrepancy between ground-truth soft labels and refined labels. Therefore, two approaches, SLR and MM are proposed to address the issue. Empirically, the proposed approach outperforms the baselines by non-trivial margins.

**Limitations And Societal Impact:**

The authors adequately addressed the limitations and potential negative societal impact.

**Main Review:**

Over all, the paper is well written and easy to follow. The motivation is clear and supported theoretical analysis. The proposed approach seems to be simple yet effective. It is good to see paper include theoretical analysis beyond intuitive motivation, Although I found theorem 2 a bit hard to follow. The experiment results are impressive, achieving state-of-the-art in various benchmarks including transfer learning.

Here are a few questions and suggestions:

1. In Theorem 1, shall we add an assumption that the loss function $\ell$ is smooth? As for non-smooth function, the norm of Jacobian might be ill-defined or not be bounded by the Lipschitz constant (the proof in Appendix).

2. In equation (5), what is the main reason to keep both $p$ and $q$? As line 190 states, $p$ could be very concentrated, which makes itself similar to the role of one-hot label $y$. It is actually interesting to see including both $p$ and $q$ improves the performance in Table 4. I try to go over the proofs of Theorem 2 in appendix to investigate the role of $p$, would you mind briefly explain the role of $p$ in theoretical analysis?

3. Theorem 1 bounds the difference between risks with different label distributions. Nevertheless, it is a bit hard to tell which label distribution benefits the downstream tasks. For instance, [1] shows that supervised loss is upper bounded by the contrastive loss under specific assumptions. If one can show the risk with ground-truth smooth labels is a tighter bound or a better estimator of the unknown supervised risk, it would make the big picture more clear.

4. Other contrastive frameworks such as SimCLR also leverage negative samples to learn representations. Can SANE be extended to SimCLR? It seems that the only difference is that there could be more dynamics for the ket set $B$ which makes it hard to be theoretically analyzed. But I hypothesize that SANE can also improve the performance of other contrastive learning frameworks that base on info-NCE loss.

Overall, I think this is a good paper with both theoretical and empirical analysis. I did not observe obvious flaw in the paper.



[1] Arora et al., A Theoretical Analysis of Contrastive Unsupervised Representation Learning, 2019.


**Time Spent Reviewing:**

4

---

> ### Author Response · Authors · 2021-08-09
> **Response to Reviewer uTpF**
>
> Thank you for the insightful review and positive feedback!
>
> (1)  On smoothness of the loss, Theorem 1  does not involve the Jacobian in the proof and thus does not require to impose this smoothness assumption. Theorem 2 substantially uses Jacobian and needs the smoothness assumptions in Assumption 1.
>
>
> (2) For $q$ in our analysis,  we set the weight $\beta_t =0$ of $q$ for simplicity, as (i) performing nonlinear operations on network output  (removing the positive and then compute the similarity) greatly increases analysis difficulty; (ii) as shown in Theorem 2, our refinery (5) is still provably sufficient to refine labels when $\beta_t=0$. This is mentioned in line 224.  As you noted, we verify the effectiveness of $q$ through the empirical experiments in Table 4.
>
>
> (3) Ground-truth smooth labels can provide a tighter bound (better estimator) of the unknown supervised risk. Specifically,  Arora et al. [1] showed that the supervised loss $L_{\text{sup}}(f)$ can be upper bounded by the contrastive loss $L_{\text{un}}(f)$ as $L_{\text{sup}}(f) \leq \alpha L_{\text{un}} (f) + C$ where $\alpha$ and $C$ are two constants. See their Theorem 4.1. Then in Eqn. (8) of their Sec. 4, they showed that the contrastive loss $L_{\text{un}}(f)$ can be divided into two components: $L_{\text{un}} (f) =\tau L_{ \neq}(f) + (1-\tau) L_{=} (f)$, where $\tau>0$ is a constant,  $L_{\neq}(f)$ denotes the loss when the query and its negatives come from different classes, $L_{=}(f)$ is the loss when a query and some of its negatives come from the same class. For details, please refer to their Sec. 4. Therefore, ground-truth smooth labels can directly reduce the second term $L_{=}(f)$, since when query and its some negatives come from same class, then the entries of the ground-truth soft labels that correspond to the negatives from same class as query are larger than other entries which correspond to real negatives (from different classes), which  means the soft labels will be more accurate and thus can reduce the loss $L_{=}(f)$. As a result, ground-truth smooth labels can provide a  tighter bound than the one using the hard-label  in conventional  contrastive loss.
>
>
> (4) We believe that besides MoCo, SANE can be extended to some other contrastive learning frameworks, e.g. SimCLR.  SANE  aims to  obtain a more accurate soft label of a query over its positive and negatives for instance discrimination, and is sufficiently general to be applied to other frameworks which use info-NCE loss to bring a query close to its positive and push it far from its negatives. This is left as our future work to thoroughly test.

---

### Official Review · Reviewer_xAFA · 2021-07-30

**Rating:** 6
**Confidence:** 4

**Summary:**

This paper proves inaccurate label assignment heavily  impairs the generalization of contrastive learning, and then proposes self-labeling refinery (SLR) to  generate accurate labels and momentum mixup (MM) to enhance similarity between query and its positive.

**Ethical Concerns:**

N.A.

**Limitations And Societal Impact:**

Yes.

**Main Review:**

This paper proves inaccurate label assignment heavily impairs the generalization of contrastive learning, and then proposes self-labeling refinery (SLR) to  generate accurate labels and momentum mixup (MM) to enhance similarity between query and its positive. I didn't check the proof, but the idea and the methodology looks good for me.
The major limitations in my opinion:
1) The issue of inaccurate label assignment exists but the severity of this issue is arguable. The proportion of negative samples with the same semantic label as query is very small, given the large batch size.
2) Experimentally the proposed method outperforms CLSA (Moco with strong augmentation) marginally (0.2 - 0.4 %) in Table 2.
3) I don't understand why p and q are sum-to-one vectors. It is possible that there're multiple samples with the same label as query.


**Time Spent Reviewing:**

2

---

> ### Author Response · Authors · 2021-08-09
> **Response to Reviewer xAFA**
>
>
> Thank you for the insightful and encouraging comments!  We hope your main concerns can be addressed by the following clarification.
>
> (1) On label assignment, the one-hot label in MoCo only requires different augmentations of the same sample to be close. So MoCo only considers the relations of augmentations of an individual sample, and does not consider the global semantic structure of data, e.g.  the semantically similar samples (e.g. samples from the same class) being close to each other. In this way, it could lead to the unsatisfactory phenomenon: the learnt features of augmentations of the same sample are close, but the features of samples from same class may be apart from each other.  As a result, these learned features cannot perform well in downstream tasks, e.g. classification. So learning semantical similarity between samples and then bring those similar samples together is very important.
> In this work, to alleviate the label issue in MoCo, we use the current training model to estimate the semantical similarity among one query and its positive and negatives, and iteratively combines this estimated similarity with the prior label, i.e. one-hot label, together to generate more accurate and informative labels.  This mechanism could better push the semantically similar samples together, and its effectiveness is verified by both theory and experimental results.
>
> The proportion of negative samples with the same semantic label as query could be small. But (1) as aforementioned, one-hot label is not informative and cannot bring semantically similar samples closer; (2) samples from different classes  also have some similarity. For example, cat is more similar to dog than car. Learning from those similarity among samples is also important and can improve the performance. This point is also supported by knowledge distillation / self-training, where it uses a teacher model / current training model to predict semantic similarity of a sample on different classes and uses the learnt informative label to supervise model training, achieving better performance.
>
>
> (2) On performance, SANE sets new state-of-the-arts for all settings and almost does not  increase computational cost (see line 347).   (a)  For 800 epochs in Table 2, as self-supervised baselines (e.g., 76.2\% by CLSA-Multi) are already close to supervised accuracy 76.5\%, it is really hard to make substantial improvement over those baselines. Yet, SANE achieves 76.4\% accuracy and still improves 0.2\% over
> SANE-Multi. Our accuracy is very close to supervised baseline 76.5\%, and sets a new state-of-the-art. (b) Table 2 shows that under 200 epochs, SANE often makes big improvement (weak aug.: SANE-Single 70.6\% v.s. MixCo 68.4\%, SANE-Multi 73.5\% v.s. SWAV-Multi 72.7\%; strong aug.: SANE-Single 70.1\% v.s. CLSA-Single 69.4\%, SANE-Multi 73.7\% v.s. SWAV-Multi 73.3\%), and makes 1\% average improvement for these settings.
>
>
> (3) The estimated soft-labels $p$ and $q$ are sum-to-one vectors, since the entries in them measures the relative similarity between a query and its positives and negatives.  This is the same to the predicted similarity in MoCo, where each query  computes its normalized  similarity (compute cosine similarity and then perform normalization) with its positive and negatives and then uses one-hot label to supervise the normalized similarity.   If there are $k$ samples that share the same label with a query,  the  entries in $p$ and $q$ corresponding to these $k$ samples are larger than 0, e.g. $1/k$, denoting these $k$ samples are more similar to the query compared with other negatives which is similar to multi-label classification.  Moreover, we need to combine $p$, $q$ and one-hot labels together in Eqn. (5) which requires them to be the same order.

---

> > ### Comment · Reviewer_xAFA · 2021-08-13
> > **Thanks for the response**
> >
> > Thanks for the response. I've read all the reviewers' comments and the authors' response.
> > The answers address my concerns.
> > So I upgrade my rating.

---

> > > ### Author Response · Authors · 2021-08-15
> > > **Response to Reviewer xAFA**
> > >
> > > We are very glad that our response can address your concerns. Thanks for your insightful comments and positive feedback again!

---

### Decision · Program_Chairs · 2021-09-27

**Decision:**

Accept (Spotlight)

**Comment:**

The reviewers are unanimously positive about the paper. Relevant question, good analysis and well-justified remedy.